# From Offline to Online Memory-Free and Task-Free Continual Learning via Fine-Grained Hypergradients

**Nicolas Michel**                                          *nicolas@cvm.t.u-tokyo.ac.jp*
*The University of Tokyo*
*Japanese-French Laboratory for Informatics*

**Maorong Wang**                                               *maorong@nii.ac.jp*
*National Institute of Informatics*

**Jiangpeng He**                                                    *jhe2@iu.edu*
*Indiana University Bloomington*

**Toshihiko Yamasaki**                                    *yamasaki@cvm.t.u-tokyo.ac.jp*
*The University of Tokyo*

Reviewed on OpenReview: *https://openreview.net/forum?id=Tu9rHiczLm*

## Abstract

Continual Learning (CL) aims to learn from a non-stationary data stream where the underlying distribution changes over time. While recent advances have produced efficient memory-free methods in the offline CL (offCL) setting, online CL (onCL) remains dominated by memory-based approaches. The transition from offCL to onCL is challenging, as many offline methods rely on (1) prior knowledge of task boundaries and (2) sophisticated scheduling or optimization schemes, both of which are unavailable when data arrives sequentially and can be seen only once. In this paper, we investigate the adaptation of state-of-the-art memory-free offCL methods to the online setting. We first show that augmenting these methods with lightweight prototypes significantly improves performance, albeit at the cost of increased Gradient Imbalance, resulting in a biased learning towards earlier tasks. To address this issue, we formulate Fine-Grained Hypergradients as an online mechanism for rebalancing gradient updates during training. Our experiments demonstrate that the synergy between prototype memory and hypergradient reweighting substantially improves the performance of memory-free methods in onCL. The code for this work are publicly available at: `https://github.com/Nicolas1203/fgh`.

## 1 Introduction

Continual Learning (CL) has gained significant popularity over the past decade (Kirkpatrick et al. (2017); Rao et al. (2019); Zhou et al. (2024a)). The core idea is to learn from a sequence of data rather than a fixed dataset. As a result, the data distribution may change, and new classes can emerge, often leading to the well-known problem of Catastrophic Forgetting (French (1999)). In this paper, we focus specifically on the Class Incremental Learning problem (Hsu et al. (2018)).

CL scenarios are typically divided into two categories: *offline* Continual Learning (offCL) (Tiwari et al. (2022)) and *online* Continual Learning (onCL) (Mai et al. (2022)). The former, which is the more widely studied setting, assumes that the data sequence is clearly segmented into discrete tasks and that training within each task is analogous to conventional learning. Specifically, data within each task are assumed to be i.i.d., and the model can be trained over multiple epochs before transitioning to the next task. In contrast, onCL assumes a stream-like data arrival, where each sample is observed only once, requiring rapid adaptation. To further align with real-world conditions, several recent studies consider scenarios with *unclear* or *blurry* task boundaries (Koh et al. (2023); Bang et al. (2022)), removing

access to task identity altogether. These differences make offCL methods poorly transferable to onCL, as many rely on multiple epochs and task boundary information. Representation-based methods such as RANPAC (McDonnell et al. (2024)) and EASE (Zhou et al. (2024b)) are prominent examples: they depend on task boundaries to compute task-specific representations, rendering them incompatible with onCL. In this paper, we aim to explore *how offCL research can contribute to the onCL Task-Free and Memory-Free scenario*.

In Online Task-Free Continual Learning (Aljundi et al. (2019); Koh et al. (2023)), state-of-the-art approaches heavily rely on memory buffers (Michel et al. (2024); Wei et al. (2023); Guo et al. (2022); Gu et al. (2022); Wei et al. (2025); Ye & Bors (2024)). Indeed, memory-based methods are well designed for onCL as they can be used in Task-Free scenarios and naturally tackle online difficulties by allowing data stored in memory to be seen multiple times. However, practically, the use of memory can be limited by hardware or privacy constraints. Conceptually, relying on memory does not solve the Continual Learning problem, but rather avoids it. Therefore, memory-free methods (Wang et al. (2022b); Smith et al. (2023); Roy et al. (2024); Wang et al. (2022a)) are a key step towards solving Continual Learning problems fundamentally, and their adaptation online makes them suitable for more realistic scenarios. Building upon prior works that leverage prototypes (De Lange & Tuytelaars (2021); Wei et al. (2023); McDonnell et al. (2024); Zhou et al. (2024b)), we show that a simple yet effective way to adapt memory-free offCL methods to the online setting is to use prototypes as a simple memory buffer for the last Fully Connected (FC) layer only. While this approach improves accuracy, it also introduces an undesirable side effect: increased Gradient Imbalance (GI) (He (2024); Guo et al. (2023); Dong et al. (2023)), leading to a biased towards earlier tasks.

Another major challenge in onCL is tuning the Learning Rate (LR). While most offCL methods rely on advanced LR optimization schemes, a common practice in onCL is to use the *same* fixed LR and optimizer for all methods (Gu et al. (2022); Mai et al. (2021); Moon et al. (2023); Lin et al. (2023)), typically Stochastic Gradient Descent (SGD) with a fixed LR of $0.1$. However, this design choice is overly restrictive, as the optimal LR varies significantly across methods and datasets. It is well known that a poorly chosen LR can critically hinder final performance. An alternative strategy is to tune the LR on one dataset and transfer it to others (Michel et al. (2024)). While more realistic, this approach provides no guarantee of generalization across datasets.

In this paper, we analyze how Gradient Imbalance and learning-rate sensitivity emerge in online continual learning, and show that *Fine-Grained Hypergradients* (FGH) provide an effective formulation to regulate these dynamics. The core idea is to extend hypergradient theory (Baydin et al. (2018)) to learn low-level gradient weights instead of high-level LR. FGH not only mitigates gradient imbalance but also improves accuracy under suboptimal LR settings. To demonstrate its effectiveness, we introduce a novel evaluation strategy that assesses performance across a range of initial LR values. Our contributions are as follows:

- We provide an empirical analysis of GI in online, task-free continual learning, and show how lightweight prototype-based adaptations exacerbate this issue.

- Based on this analysis, we formulate Fine-Grained Hypergradients as a parameter-wise gradient reweighting mechanism that regulates gradient stability under non-stationary data streams.

- We propose a multi-LR evaluation protocol that reflects realistic online continual learning settings and shows consistent robustness across a wide range of learning rates.

## 2 Related Work

### 2.1 Continual Learning with Blurry Boundaries

Continual Learning (CL) is generally framed as training a model $f_\theta(\cdot)$, parameterized by $\theta$, on a sequence of $K$ tasks. Each task, indexed by $k \in 1, \cdots, K$, is associated with a dataset $\mathcal{D}_k$, which may be drawn from a distinct distribution. In Class Incremental Learning (Hsu et al. (2018)), each dataset is composed of data-label pairs, $\mathcal{D}_k = (\mathcal{X}_k, \mathcal{Y}_k)$. In online CL (onCL), data arrive in a stream and can typically be observed only once (He et al. (2020)), making access to *clear* task boundaries unlikely. Consequently, several studies propose working under boundary-free scenarios (Buzzega et al. (2020)), where task changes are unknown. However, when task changes are *clear*, they may still be inferred. To better model intermediate cases, the *blurry boundary* setting has been introduced (Koh et al. (2023); Bang et al. (2022); Michel et al. (2024)). Of particular interest is the *Si-Blurry* setting (Moon et al. (2023)), in which task boundaries are

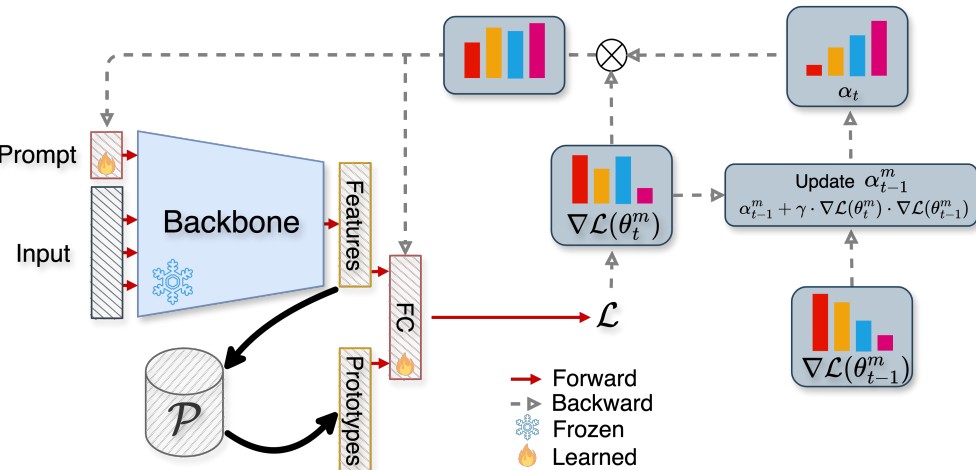

Figure 1: Overall training procedure of FGH when combined with prompt-tuning strategies.

not only blurry but also allow classes to appear or disappear across multiple tasks. This setup is more reflective of real-world scenarios while also presenting additional challenges for continual learning algorithms.

## 2.2 Continual Learning with Memory Buffer

Memory buffers remain among the most practical and effective strategies for mitigating forgetting in onCL (Raghavan et al. (2024b;a); Guo et al. (2022); Su et al. (2025); He & Zhu (2022); Caccia et al. (2022); Ye & Bors (2024); Wang et al. (2024b;a); Buzzega et al. (2021)). Some works have even shown that memory alone can yield competitive performance (Prabhu et al. (2020); Michel et al. (2022)), highlighting its importance in the online setting. As a result, memory buffering is considered a core component of many onCL methods. In contrast, offCL has recently seen a shift toward memory-free approaches (He et al. (2025); Liang & Li (2024); McDonnell et al. (2024); Wang et al. (2022a)). While some memory-free methods have been adapted to the online setting, their performance typically lags behind memory-based approaches (Moon et al. (2023); Wei et al. (2025)). In this work, we aim to bridge this gap by leveraging memory-free offCL methods in the onCL setting.

## 2.3 Hypergradients and Gradient Re-weighting

Hypergradients (Baydin et al. (2018); Almeida et al. (1999)) address the challenge of optimizing learning rates in standard training setups. The key idea is to derive a gradient descent algorithm that updates the learning rate itself. Notably, it is demonstrated that computing the dot product of consecutive gradients, $\nabla \mathcal{L}(\theta_t) \cdot \nabla \mathcal{L}(\theta_{t-1})$, is sufficient to perform one update step for the learning rate. Here, $t$ is the current step index, $\theta$ denotes the model parameters, and $\mathcal{L}$ is the loss function. However, such techniques have traditionally been developed for offline training and applied at a global scale. In the context of CL, gradient re-weighting strategies have been explored primarily in replay-based methods, often focusing on the last layer. For example, previous work has proposed *manually* re-weighting the gradient at the loss level to reduce its accumulation during training, addressing the issue of Gradient Imbalance (Guo et al. (2023); He (2024)). In this work, we extend this idea by introducing Fine-Grained Hypergradients, which enable *learned* gradient re-weighting across all trainable parameters, not just the last layer. This approach allows for more precise control of gradient dynamics during training in onCL scenarios.

## 3 Methodology

To bridge offline and online CL research, this work proposes to adapt and improve existing offCL memory-free methods to the onCL, memory-free, and task-free problem. Firstly, we present the online adaptation and challenges induced by the onCL context. Secondly, we propose to leverage simple prototypes as an efficient way to counter forgetting, without storing input data. Finally, to address the challenges regarding Learning Rate selection and Gradient Imbalance,

we formulate Fine-Grained Hypergradients as an online, adaptive gradient reweighting mechanism. Our goal is not to optimize learning rates per se, but to regulate gradient stability across time in non-stationary, class-incremental streams.

## 3.1 From Offline to Online

Adapting offline methods to the online setting is non-trivial. We highlight key components of offCL methods and outline the modifications necessary to make them applicable in the online scenario.

**Removing Task Boundary Information.**    Typically, most offline methods take advantage of the knowledge of task boundaries (Zhou et al. (2024b); McDonnell et al. (2024); Liang & Li (2024); Smith et al. (2023); Wang et al. (2022b;a); Roy et al. (2024)). While representation-based methods cannot be adapted online as the exact task change is required to recompute representations, most prompt-based methods happen to be more flexible as the task information is used solely to freeze certain prompts in the prompt pool (Smith et al. (2023); Wang et al. (2022b)). Such a prompt-freezing strategy tackles prompt forgetting during training in offCL. Therefore, if learned prompts are never frozen, prompt-based approaches can easily be trained in task-free onCL. More details regarding the parameters used are given in the Appendix.

**Learning Rate Selection.**    When training offCL methods, the choice of the LR as well as the use of an LR scheduler is particularly impactful. In general, LR selection remains a difficult topic in Continual Learning, as, in theory, future datasets are unknown and hyperparameter search is unavailable (Cha & Cho (2024)). This problem is even more pronounced in the online setting, as not even a learning rate scheduler can be used, since the length and boundaries of tasks are considered unknown. More importantly, naively transferring LR values used in offCL to onCL often leads to unsatisfactory performance. Therefore, we evaluate every online method with various fixed learning rate values and report the results in Section 4.3. Additional information regarding the evaluation procedure is provided in Section 4.1.

**Gradient Imbalance.**    Gradient Imbalance (He (2024); Guo et al. (2023); Dong et al. (2023)) in Continual Learning occurs when the model suffers from larger gradients toward specific samples or classes during training. An example of such an imbalance with larger gradients for earlier classes is given in Figure 2. The main consequence is that the model will give stronger updates with regard to specific classes. While this problem can similarly be observed offline, it is most severe in onCL as (1) each data point is seen only once, so the training cannot be adapted from task to task, (2) the usage of memory increases such imbalance (He (2024)), and as discussed above, memory is essential in onCL. When adapting offCL methods in onCL, we not only observe GI, but see an increase in such imbalance when introducing prototypes in Section 3.2.

## 3.2 Prototypes as a Proxy for Memory

As discussed above, memory is at the core of most state-of-the-art onCL methods. In this study, we propose leveraging online prototypes to act as a memory buffer for the last layer only. In this context, we compute prototypes $\mathcal{P} = \{p_{k_1}^1, p_{k_2}^2, \cdots, p_{k_c}^c\}$ for each class during training. Let us consider a model $f_\theta$ parameterized by $\theta$ such that for an input $x \in \mathbb{R}^d$, with $d$ being the dimension of the input space, we have $f_\theta(x) = h_w(x)^T \cdot W$, where $W \in \mathbb{R}^{l,c}$, $c$ is the number of classes, $l$ is the dimension of the output of $h_w$, and $\theta = \{w, W\}$. In this context, $h_w$ would typically be a pre-trained model, and $W$ is the weight of the final FC layer (including the bias). For a given class $j$, the class prototype $p_{k_j}^j$ computed over $k_j$ samples is updated when encountering a new sample $x_{k_j+1}^j$. For simplicity, we omit the $j$ index in $k_j$ going forward. Therefore, we leverage a simple prototype update rule:

$$p_{k+1}^j = \frac{k \cdot p_k^j + h_w(x_{k+1}^j)}{k+1}, \tag{1}$$

where $x_{k+1}^j$ is the $k+1^{th}$ encountered sample of class $j$. For all classes, prototypes are initialized such that $p_0^j = 0$. Prototypes are then used to recalibrate the final FC layer, analogous to replaying the average of past data representations during training. In this sense, we define the prototype-based loss term as:

$$\mathcal{L}_P = -\frac{1}{c} \sum_{j \in \mathcal{C}_{old}} \log \left( \text{softmax} \left( (p^j)^T \cdot W^j \right) \right) \tag{2}$$

where $\mathcal{C}_{old} = \{j \in \{1, \cdots, c\} \mid p_k^j \neq 0\}$. $\mathcal{L}_P$ is the cross-entropy loss with respect to prototypes of encountered classes. As discussed in section 5, while using prototypes as a memory buffer can significantly improve the performance of the considered methods, it also increases the GI in the final layer of continually trained models (He (2024)).

### 3.3 Fine-Grained Hypergradients

In order to give the model the capacity to adapt its LR at a local and global level, we introduce Fine-Grained Hypergradients. FGH introduces independent weights for each trainable parameter, allowing fine-grained adaptation of individual gradients during the learning process, rather than only high-level learning rate adaptation. Formally, let us consider the update rule for an individual parameter $\theta^m \in \theta$ induced by gradient-based optimization algorithms over parameters $\theta$, given a learning rate $\eta$:

$$\theta_{t+1}^m = \theta_t^m - \eta \nabla \mathcal{L}(\theta_t^m), \tag{3}$$

where $t$ is the iteration index and $1 \leq m \leq D$ with $D \in \mathbb{R}^+$ the number of trainable parameters. To reweight the learned gradient, we introduce step-dependent weighting coefficients, leading to the following update rule:

$$\theta_{t+1}^m = \theta_t^m - \alpha_{t+1}^m \eta \nabla \mathcal{L}(\theta_t^m), \tag{4}$$

where $\alpha_t^m \in \mathbb{R}^{+\star}$ is the parameter-dependent gradient weighting coefficient at step $t$. While such gradient weighting strategies were previously limited to the last layer and computed with hand-crafted rules (He (2024)), we propose learning them during training. In particular, we aim to construct a higher-level update for $\alpha_t^m$ such that:

$$\alpha_{t+1}^m = \alpha_t^m - \beta \frac{\partial \mathcal{L}(\theta_t^m)}{\partial \alpha_t^m}, \tag{5}$$

with $\beta \in \mathbb{R}^+$ the hypergradient learning rate. To compute the partial derivative, we apply the chain rule and make use of the fact that $\theta_t^m = \theta_{t-1}^m - \alpha_t^m \eta \nabla \mathcal{L}(\theta_{t-1}^m)$, such that:

$$\frac{\partial \mathcal{L}(\theta_t^m)}{\partial \alpha_t^m} = \nabla \mathcal{L}(\theta_t^m) \cdot \frac{\partial \theta_t^m}{\partial \alpha_t^m} = -\eta \nabla \mathcal{L}(\theta_t^m) \cdot \nabla \mathcal{L}(\theta_{t-1}^m). \tag{6}$$

The resulting Fine-Grained Hypergradients update becomes, for any $1 \leq m \leq D$:

$$\alpha_{t+1}^m = \alpha_t^m + \gamma \cdot \nabla \mathcal{L}(\theta_t^m) \cdot \nabla \mathcal{L}(\theta_{t-1}^m), \tag{7}$$

where $\gamma = \beta \eta$. Our FGH module gives the model the capacity to modify the LR locally, potentially mitigating GI, as well as globally, potentially tackling the problem of unknown LR. Naturally, this introduces an additional hyperparameter. We discuss this limitation in Section 5. For clarity, the relation presented in equation 7 relies on an SGD update. In practice, we favor a momentum-based update. Its detailed implementation is provided in the Appendix.

### 3.4 Overall Training Procedure

Considering a baseline memory-free offCL method trained by minimizing a baseline loss $\mathcal{L}_{base}$, we can adapt it to onCL by introducing prototypes and FGH in the training procedure. We simply add the extra loss term $\mathcal{L}_p$, which amounts to minimizing the overall loss $\mathcal{L} = \mathcal{L}_{base} + \mathcal{L}_p$. Additionally, we modify the gradient update to adjust the gradient weights as defined in Section 3.3. Furthermore, we leverage batch-wise masking to consider the logits of classes that are only present in the current batch. An overview of the training procedure is given in Figure 1. Similarly, a Pytorch-like implementation in the case of SGD is presented in Algorithm 1.

## 4 Experiments

### 4.1 Evaluation Procedure

**Metric.** We follow previous work and define the Average Performance (AP) as the average of the accuracies computed after each task during training (Zhou et al. (2024a)). More details in Appendix.

**Algorithm 1** PyTorch-like pseudo-code of integrating prototypes as memory and FGH with baselines.

```python
gamma, grad_weight, old_grad = 1, {}, {}
for x, y in dataloader:
  h, y_hat = network(x) # features and logits
  loss_baseline = criterion_baseline(y_hat, y) # Baseline loss
  proto, labels = get_prototypes() # Prototypes as memory
  loss_p = cross_entropy(network.fc(proto), labels) # Eq. 2
  loss = loss_baseline + loss_p
  loss.backward() # compute gradients

  # Fine-Grained Hypergradient update
  for i, param in enumerate(network.parameters()):
    curr_grad = param.grad
    if curr_grad is not None:
      if i in grad_weight.keys():
        grad_weight[i] = grad_weight[i] + gamma * curr_grad * old_grad[i] #Eq. 7
        param.grad = grad_weight[i] * param.grad
      else:
        grad_weight[i] = 1.0
      old_grad[i] = curr_grad
  optim.step()
  update_proto(h, y) # Eq. 1
```

**Multi-Learning-Rate Evaluation.** Since finding the optimal LR in onCL is an especially hard task, we introduce a new evaluation setting based on a multi-LR evaluation. Indeed, we propose to give the performances of the compared methods with various LR values. In particular, each method is evaluated given three cases: (1) Using a low LR value, (2) Using a high LR value, (3) Using the best LR value found after conducting a small search for $\gamma$ and the LR on VTAB (Zhai et al. (2019)). Specifically, we experiment for LR values in $\{5 \times 10^{-5}, 5 \times 10^{-3}\}$. The intuition behind such values is that we hypothesize that the optimal LR is likely to fall into that range, and such values are often used in the literature. Such a metric should emphasize the validity of the approach when the optimal LR is unknown, leading to a fairer comparison than using the same LR blindly for every approach.

## 4.2 Experimental Setting

**Baselines and Datasets.** In order to demonstrate the benefits of our approach, we integrate it with several state-of-the-art methods in offCL, when adapting them to onCL. Notably, **L2P** (Wang et al. (2022b)), **DualPrompt** (Wang et al. (2022a)), **CODA** (Smith et al. (2023)), **ConvPrompt** (Roy et al. (2024)). These methods are not naturally suited for the online case, so they had to be adapted, as described in Section 3.1. Additionally, we compare adapted methods to state-of-the-art memory-free onCL methods **MVP** (Moon et al. (2023)) and **Online LORA (oLoRA)** (Wei et al. (2025)). Eventually, we experimented with **Experience Replay (ER)** (Rolnick et al. (2019)) to compare with a traditional memory-based approach, as well as **fine-tuning** and **linear probe** baselines. We evaluate our method on **CUB** (Wah et al. (2011)), **ImageNet-R** (Hendrycks et al. (2021)) and **CIFAR100** (Krizhevsky (2012)). As introduced above, we conduct a small hyperparameter search regarding the LR on VTAB (Zhai et al. (2019)), which is referred to as the *best* columns in Tables 1 and 2. More details in the Appendix.

*Clear* and *Blurry* Boundaries. We experiment in *clear* boundaries settings, for continuity with previous work, despite its lack of realism for onCL. In that sense, we consider an initial count of 10 classes for the first task, with an increment of 10 classes per task. This results in 10 tasks with 10 classes per task for CIFAR100, as well as 20 tasks with 10 classes per task for CUB and ImageNet-R. However, to evaluate our method in more realistic scenarios, we reckon the *Si-Blurry* (Moon et al. (2023)) setting to be the most relevant to our study case. Specifically, we use their implementation of Stochastic incremental Blurry boundaries (*Si-Blurry*). We use the same number of tasks as for the *clear* setting. In this case, some classes can appear or disappear during training, and the transitions are not necessarily clear. More details on this setting can be found in the Appendix.

Table 1: Average performance (%) of all considered baselines in the *Si-Blurry* setting. + *ours* refers to combining baselines with prototypes and FGH. Best HP refers to the best set of LR and $\gamma$ found on VTAB. In some cases, the best HP coincides with one of the default HP values.

| Dataset | CIFAR100 | | | CUB | | | ImageNet-R | | |
|---|---|---|---|---|---|---|---|---|---|
| Learning Rate | $5 \times 10^{-5}$ | $5 \times 10^{-3}$ | Best HP | $5 \times 10^{-5}$ | $5 \times 10^{-3}$ | Best HP | $5 \times 10^{-5}$ | $5 \times 10^{-3}$ | Best HP |
| Fine-tuning | $29.54_{\pm6.44}$ | $2.46_{\pm0.30}$ | $2.42_{\pm0.26}$ | $6.13_{\pm1.87}$ | $1.42_{\pm0.27}$ | $1.38_{\pm0.24}$ | $3.96_{\pm0.70}$ | $1.38_{\pm0.17}$ | $1.49_{\pm0.26}$ |
| Linear probe | $22.15_{\pm3.89}$ | $35.59_{\pm4.14}$ | $35.59_{\pm4.14}$ | $2.24_{\pm0.49}$ | $49.37_{\pm2.75}$ | $49.37_{\pm2.75}$ | $3.83_{\pm0.41}$ | $34.53_{\pm1.56}$ | $34.53_{\pm1.56}$ |
| ER | $\mathbf{81.33_{\pm3.04}}$ | $3.14_{\pm0.63}$ | $81.33_{\pm3.04}$ | $\mathbf{52.45_{\pm3.02}}$ | $1.56_{\pm0.33}$ | $52.45_{\pm3.02}$ | $\mathbf{55.06_{\pm1.92}}$ | $2.00_{\pm0.43}$ | $55.06_{\pm1.92}$ |
| ER + Linear probe | $34.69_{\pm5.44}$ | $79.97_{\pm2.24}$ | $79.74_{\pm2.45}$ | $4.34_{\pm0.92}$ | $64.20_{\pm1.37}$ | $64.07_{\pm1.45}$ | $7.70_{\pm0.92}$ | $54.52_{\pm1.19}$ | $53.98_{\pm1.11}$ |
| MVP | $21.57_{\pm2.27}$ | $41.42_{\pm6.00}$ | $36.88_{\pm1.97}$ | $2.73_{\pm0.65}$ | $47.11_{\pm2.62}$ | $39.12_{\pm2.82}$ | $4.19_{\pm0.55}$ | $31.35_{\pm2.29}$ | $28.48_{\pm1.18}$ |
| oLoRA | $36.27_{\pm4.01}$ | $27.04_{\pm7.18}$ | $34.67_{\pm7.51}$ | $5.04_{\pm1.56}$ | $49.04_{\pm2.24}$ | $47.37_{\pm1.51}$ | $8.82_{\pm1.59}$ | $33.08_{\pm3.67}$ | $39.29_{\pm5.71}$ |
| CODA | $15.14_{\pm3.78}$ | $71.12_{\pm4.47}$ | $56.03_{\pm2.10}$ | $0.83_{\pm0.35}$ | $53.17_{\pm1.96}$ | $35.90_{\pm6.33}$ | $1.92_{\pm0.62}$ | $47.65_{\pm1.40}$ | $32.93_{\pm1.85}$ |
| $\hookrightarrow$ + ours | $44.21_{\pm8.04}$ | $79.47_{\pm2.23}$ | $69.04_{\pm2.56}$ | $4.50_{\pm0.63}$ | $68.64_{\pm3.19}$ | $47.49_{\pm5.25}$ | $9.95_{\pm1.79}$ | $57.16_{\pm1.17}$ | $41.68_{\pm2.32}$ |
| L2P | $10.80_{\pm4.39}$ | $58.20_{\pm6.59}$ | $58.20_{\pm6.59}$ | $0.46_{\pm0.24}$ | $30.57_{\pm3.85}$ | $30.57_{\pm3.85}$ | $1.05_{\pm0.29}$ | $27.17_{\pm4.61}$ | $27.17_{\pm4.61}$ |
| $\hookrightarrow$ + ours | $33.05_{\pm8.01}$ | $79.22_{\pm3.02}$ | $79.22_{\pm3.02}$ | $2.00_{\pm0.98}$ | $68.68_{\pm2.29}$ | $68.68_{\pm2.29}$ | $5.80_{\pm1.47}$ | $59.89_{\pm2.05}$ | $59.89_{\pm2.05}$ |
| DualPrompt | $15.68_{\pm3.53}$ | $66.90_{\pm5.04}$ | $53.39_{\pm5.35}$ | $0.97_{\pm0.42}$ | $52.32_{\pm2.40}$ | $43.76_{\pm3.94}$ | $1.80_{\pm0.39}$ | $46.05_{\pm1.74}$ | $35.27_{\pm2.61}$ |
| $\hookrightarrow$ + ours | $42.12_{\pm6.34}$ | $75.23_{\pm3.21}$ | $70.34_{\pm1.44}$ | $5.43_{\pm0.98}$ | $74.89_{\pm1.51}$ | $67.38_{\pm3.13}$ | $10.11_{\pm1.38}$ | $57.68_{\pm1.70}$ | $51.86_{\pm1.15}$ |
| ConvPrompt | $24.55_{\pm3.80}$ | $75.01_{\pm5.16}$ | $75.01_{\pm5.16}$ | $0.64_{\pm0.23}$ | $56.27_{\pm0.84}$ | $56.27_{\pm0.84}$ | $1.18_{\pm0.02}$ | $46.75_{\pm1.80}$ | $46.75_{\pm1.80}$ |
| $\hookrightarrow$ + ours | $44.23_{\pm3.29}$ | $\mathbf{86.34_{\pm3.59}}$ | $\mathbf{86.34_{\pm3.59}}$ | $4.43_{\pm1.13}$ | $\mathbf{73.88_{\pm0.87}}$ | $\mathbf{73.88_{\pm0.87}}$ | $3.78_{\pm0.22}$ | $\mathbf{62.62_{\pm0.11}}$ | $\mathbf{62.62_{\pm0.11}}$ |

Table 2: Average Performances (%) of all considered baselines, in the *clear* setting. + *ours* refers to combining the baselines with prototypes and FGH. Best HP refer to the best set of LR and $\gamma$ found on VTAB. In some cases, the best HP is the same as one of the default HP values.

| Dataset | CIFAR100 | | | CUB | | | Imagenet-R | | |
|---|---|---|---|---|---|---|---|---|---|
| Learning Rate | $5 \times 10^{-5}$ | $5 \times 10^{-3}$ | Best HP | $5 \times 10^{-5}$ | $5 \times 10^{-3}$ | Best HP | $5 \times 10^{-5}$ | $5 \times 10^{-3}$ | Best HP |
| Fine-tuning | $29.83_{\pm0.56}$ | $2.93_{\pm0.00}$ | $2.93_{\pm0.00}$ | $5.89_{\pm0.91}$ | $1.70_{\pm0.15}$ | $1.77_{\pm0.14}$ | $9.14_{\pm0.94}$ | $2.16_{\pm0.33}$ | $1.92_{\pm0.15}$ |
| Linear probe | $12.50_{\pm1.45}$ | $30.37_{\pm0.63}$ | $30.37_{\pm0.63}$ | $0.80_{\pm0.22}$ | $53.40_{\pm1.62}$ | $53.40_{\pm1.62}$ | $1.87_{\pm0.31}$ | $35.54_{\pm1.04}$ | $35.54_{\pm1.04}$ |
| ER | $\mathbf{81.73_{\pm0.60}}$ | $2.95_{\pm0.06}$ | $81.73_{\pm0.60}$ | $\mathbf{42.92_{\pm3.21}}$ | $1.76_{\pm0.20}$ | $42.92_{\pm3.21}$ | $\mathbf{53.43_{\pm1.19}}$ | $2.17_{\pm0.35}$ | $53.43_{\pm1.19}$ |
| ER + Linear probe | $33.69_{\pm1.47}$ | $81.79_{\pm1.21}$ | $81.13_{\pm1.21}$ | $2.35_{\pm0.24}$ | $63.03_{\pm1.08}$ | $62.81_{\pm0.93}$ | $6.04_{\pm0.83}$ | $51.62_{\pm1.06}$ | $50.25_{\pm1.02}$ |
| MVP | $21.60_{\pm1.58}$ | $33.10_{\pm0.75}$ | $24.97_{\pm1.24}$ | $2.85_{\pm0.76}$ | $57.17_{\pm1.36}$ | $51.08_{\pm2.06}$ | $4.20_{\pm0.73}$ | $35.53_{\pm1.31}$ | $34.53_{\pm1.84}$ |
| oLoRA | $35.35_{\pm5.23}$ | $22.99_{\pm1.77}$ | $29.08_{\pm1.39}$ | $3.88_{\pm1.40}$ | $53.15_{\pm3.92}$ | $43.44_{\pm2.96}$ | $7.01_{\pm0.58}$ | $36.91_{\pm1.79}$ | $48.90_{\pm1.73}$ |
| CODA | $24.71_{\pm2.62}$ | $71.62_{\pm2.35}$ | $66.66_{\pm3.08}$ | $2.54_{\pm0.68}$ | $61.04_{\pm2.98}$ | $49.13_{\pm3.05}$ | $3.64_{\pm0.87}$ | $62.33_{\pm1.97}$ | $53.63_{\pm2.05}$ |
| $\hookrightarrow$ + ours | $58.76_{\pm2.28}$ | $78.50_{\pm1.43}$ | $71.40_{\pm3.86}$ | $5.84_{\pm1.13}$ | $70.32_{\pm2.70}$ | $56.63_{\pm3.97}$ | $13.02_{\pm1.38}$ | $64.41_{\pm1.25}$ | $58.33_{\pm2.09}$ |
| L2P | $20.95_{\pm4.49}$ | $64.86_{\pm3.78}$ | $64.86_{\pm3.78}$ | $2.03_{\pm0.75}$ | $35.67_{\pm3.36}$ | $35.67_{\pm3.36}$ | $3.50_{\pm1.16}$ | $43.15_{\pm2.81}$ | $43.15_{\pm2.81}$ |
| $\hookrightarrow$ + ours | $52.95_{\pm2.94}$ | $82.26_{\pm0.76}$ | $82.26_{\pm0.76}$ | $3.94_{\pm1.12}$ | $72.60_{\pm1.16}$ | $72.60_{\pm1.16}$ | $11.82_{\pm1.42}$ | $66.96_{\pm0.80}$ | $66.96_{\pm0.80}$ |
| DualPrompt | $23.24_{\pm1.59}$ | $69.17_{\pm2.27}$ | $64.77_{\pm2.52}$ | $2.62_{\pm0.69}$ | $61.26_{\pm2.38}$ | $55.62_{\pm2.06}$ | $3.64_{\pm0.46}$ | $59.55_{\pm1.23}$ | $54.23_{\pm0.95}$ |
| $\hookrightarrow$ + ours | $52.84_{\pm2.19}$ | $75.01_{\pm1.32}$ | $72.74_{\pm1.02}$ | $6.09_{\pm1.08}$ | $\mathbf{78.56_{\pm0.87}}$ | $73.30_{\pm0.80}$ | $13.50_{\pm1.02}$ | $63.74_{\pm0.51}$ | $62.47_{\pm1.05}$ |
| ConvPrompt | $33.80_{\pm0.71}$ | $73.88_{\pm3.15}$ | $73.88_{\pm3.15}$ | $2.14_{\pm0.54}$ | $65.96_{\pm2.78}$ | $65.96_{\pm2.78}$ | $3.07_{\pm0.37}$ | $59.60_{\pm0.29}$ | $59.60_{\pm0.29}$ |
| $\hookrightarrow$ + ours | $60.07_{\pm1.37}$ | $\mathbf{87.65_{\pm0.37}}$ | $\mathbf{87.65_{\pm0.37}}$ | $5.54_{\pm1.10}$ | $75.73_{\pm0.12}$ | $\mathbf{75.73_{\pm0.12}}$ | $6.89_{\pm0.32}$ | $\mathbf{69.76_{\pm1.38}}$ | $\mathbf{69.76_{\pm1.38}}$ |

**Implementation Details.** Every method is evaluated in the onCL context, where only one pass over the data is allowed. The batch size is fixed at 100 to simulate small data increments with a low storage budget in the context of fast adaptation. The backbone used for all compared approaches is a ViT-base (Dosovitskiy et al. (2021)), pre-trained on ImageNet 21k. Each experiment was conducted over 10 runs, and the average and standard deviation are reported, except for ConvPrompt and oLoRA, where only 3 runs were used due to their intensive computation requirements. The memory size of memory-based methods is set to 1000. Each run was conducted with a different seed, which also impacted the task generation process. For all experiments, we use $\gamma = 1$ as the default value. More details on the selection of $\gamma$ can be found in Section 5.2. More details are given in the Appendix.

Table 3: Average Performances (%) of all considered baselines with and without prototypes as memory and FGH, in the *Si-Blurry* setting. Results over 10 runs are displayed, and $\gamma = 1$.

| Dataset | CIFAR100 | | CUB | | Imagenet-R | |
|---|---|---|---|---|---|---|
| Learning Rate | $5 \times 10^{-5}$ | $5 \times 10^{-3}$ | $5 \times 10^{-5}$ | $5 \times 10^{-3}$ | $5 \times 10^{-5}$ | $5 \times 10^{-3}$ |
| CODA | $15.14_{\pm3.78}$ | $71.12_{\pm4.47}$ | $0.83_{\pm0.35}$ | $53.17_{\pm1.96}$ | $1.92_{\pm0.62}$ | $47.65_{\pm1.4}$ |
| $\hookrightarrow + P$ | $31.27_{\pm6.95}$ | $78.18_{\pm3.3}$ | $1.69_{\pm0.39}$ | $62.93_{\pm4.78}$ | $4.36_{\pm1.11}$ | $55.92_{\pm1.84}$ |
| $\hookrightarrow + FGH$ | $22.27_{\pm5.88}$ | $69.66_{\pm3.24}$ | $0.84_{\pm0.34}$ | $50.45_{\pm2.43}$ | $2.14_{\pm0.71}$ | $44.31_{\pm3.01}$ |
| $\hookrightarrow + P + FGH$ | $\mathbf{44.21_{\pm8.04}}$ | $\mathbf{79.47_{\pm2.23}}$ | $\mathbf{4.5_{\pm0.63}}$ | $\mathbf{68.64_{\pm3.19}}$ | $\mathbf{9.95_{\pm1.79}}$ | $\mathbf{57.16_{\pm1.17}}$ |
| L2P | $10.8_{\pm4.39}$ | $58.2_{\pm6.59}$ | $0.46_{\pm0.24}$ | $30.57_{\pm3.85}$ | $1.05_{\pm0.29}$ | $27.17_{\pm4.61}$ |
| $\hookrightarrow + P$ | $22.81_{\pm6.61}$ | $78.13_{\pm3.15}$ | $0.82_{\pm0.39}$ | $64.18_{\pm3.26}$ | $2.39_{\pm0.68}$ | $57.83_{\pm2.09}$ |
| $\hookrightarrow + FGH$ | $15.44_{\pm5.76}$ | $55.66_{\pm4.25}$ | $0.46_{\pm0.23}$ | $27.68_{\pm3.64}$ | $1.15_{\pm0.33}$ | $24.15_{\pm5.8}$ |
| $\hookrightarrow + P + FGH$ | $\mathbf{33.05_{\pm8.01}}$ | $\mathbf{79.22_{\pm3.02}}$ | $\mathbf{2.0_{\pm0.98}}$ | $\mathbf{68.68_{\pm2.29}}$ | $\mathbf{5.8_{\pm1.47}}$ | $\mathbf{59.89_{\pm2.05}}$ |
| DualPrompt | $15.68_{\pm3.53}$ | $66.9_{\pm5.04}$ | $0.97_{\pm0.42}$ | $52.32_{\pm2.4}$ | $1.8_{\pm0.39}$ | $46.05_{\pm1.74}$ |
| $\hookrightarrow + P$ | $30.12_{\pm5.66}$ | $74.22_{\pm3.93}$ | $2.07_{\pm0.67}$ | $71.96_{\pm1.6}$ | $4.43_{\pm0.84}$ | $\mathbf{58.37_{\pm1.93}}$ |
| $\hookrightarrow + FGH$ | $22.26_{\pm5.49}$ | $63.93_{\pm3.76}$ | $0.96_{\pm0.43}$ | $50.2_{\pm2.57}$ | $2.09_{\pm0.53}$ | $40.02_{\pm2.42}$ |
| $\hookrightarrow + P + FGH$ | $\mathbf{42.12_{\pm6.34}}$ | $\mathbf{75.23_{\pm3.21}}$ | $\mathbf{5.43_{\pm0.98}}$ | $\mathbf{74.89_{\pm1.51}}$ | $\mathbf{10.11_{\pm1.38}}$ | $57.68_{\pm1.7}$ |
| ConvPrompt | $24.55_{\pm3.8}$ | $75.01_{\pm5.16}$ | $0.64_{\pm0.23}$ | $56.27_{\pm0.84}$ | $1.18_{\pm0.02}$ | $46.75_{\pm1.8}$ |
| $\hookrightarrow + P$ | $42.3_{\pm3.72}$ | $84.14_{\pm3.07}$ | $1.9_{\pm0.63}$ | $70.81_{\pm0.86}$ | $2.41_{\pm0.26}$ | $57.42_{\pm2.58}$ |
| $\hookrightarrow + FGH$ | $28.64_{\pm2.04}$ | $75.99_{\pm7.1}$ | $0.83_{\pm0.15}$ | $55.96_{\pm2.7}$ | $1.19_{\pm0.04}$ | $49.39_{\pm0.69}$ |
| $\hookrightarrow + P + FGH$ | $\mathbf{44.23_{\pm3.29}}$ | $\mathbf{86.34_{\pm3.59}}$ | $\mathbf{4.43_{\pm1.13}}$ | $\mathbf{73.88_{\pm0.87}}$ | $\mathbf{3.78_{\pm0.22}}$ | $\mathbf{62.62_{\pm0.11}}$ |

## 4.3 Experimental Results

**Improvement over suboptimal LR.** As shown in Table 1 and Table 2, augmenting the considered baselines with our proposed strategy consistently yields performance improvements. Moreover, the strongest memory-free methods are always achieved when our strategy is employed. Notably, the relative gain is most pronounced when starting from a suboptimal LR. For instance, on CIFAR100 with ConvPrompt (Table 2), using an initial LR of $5 \times 10^{-5}$ results in a performance improvement of $26.3\%$, whereas with a higher LR of $5 \times 10^{-3}$ the improvement is $13.8\%$. A similar trend can be observed in Table 3, where the benefit of incorporating FGH over prototypes becomes more significant at lower LR values.

**Methods for offCL are Powerful onCL Learners.** We evaluate in both *clear* and *blurry* settings, reporting Average Performance in Table 2 and Table 1. For offCL methods (without + *ours*), we apply only the adaptation described in Section 3.1. Interestingly, these methods prove highly effective in the online setting, often outperforming MVP and oLoRA, despite being originally designed for offline learning. A likely explanation is that prior work typically applied offline hyperparameters directly to the online problem, leading to suboptimal results. In some cases, such as CIFAR100 with an LR of $5 \times 10^{-5}$, oLoRA achieves the strongest performance among memory-free baselines. However, under the *Best HP* setting, offCL methods consistently achieve substantially better results. It is also worth noting that MVP and oLoRA depend on several additional hyperparameters, which may not generalize across scenarios. Together, these observations highlight the central role of learning rate and hyperparameter selection in Continual Learning.

**Ablation Study.** To clarify the contribution of each component of our method, we include the performance of the original baselines, followed by the performance of these baselines combined with Prototypes only (+ *P*), and the performance of these baselines combined with FGH (+ *FGH*). These results are included in Tables 3 for the blurry scenario. While it is clear that the use of prototypes is largely beneficial, in some situations, the addition of *FGH* can lead to a drop in performance. One explanation for this observation is the reverse GI induced by the usage of FGH, as presented in Section 5.1 and Figure 2. Larger gradients on newer tasks induce faster learning of newly introduced classes, with the risk of increased forgetting on earlier classes. Even though this imbalance might be favorable, leveraging FGH without any stability-focused measures can lead to lower performance. Nonetheless, the combination of both strategies largely leads to the best performance. We give more details on this stability-plasticity trade-off in the Appendix.

## 5 Discussions

### 5.1 Prototypes and FGH Synergy

**FGH impact on Gradient Imbalance.**     To analyze the inner workings between prototypes and FGH, let us consider the last classification layer $W$ as defined in Section 3.2. Each column $W^j$, with $j \in \{1, c\}$ as a class index, corresponds to the class-specific weights of the last layer. Therefore, at a training step $t$, we can define the class-specific gradient $g_t^j = \nabla \mathcal{L}(W_t^j)$. We are interested in the average gradient norm throughout training, which is $g^j = \frac{1}{t_{max}} \sum_{t=1}^{t_{max}} ||g_t^j||$, with $t_{max}$ being the maximum number of training steps. Similarly, we define the *task-specific* gradient norm at the end of training for a task $k$ as $G^k = \frac{1}{|C_k|} \sum_{j \in C_k} g^j$, with $C_k$ being the classes present in task $k$. We define:

$$G_n^k = \frac{G^k}{\max_{1 \leq l \leq T} G^l} \tag{8}$$

as the normalized average gradient norm corresponding to a task $k$ at the end of training. We show the values of $G_n^k$ at the end of training for CODA on CIFAR100 in the clear setting and an LR of $5 \times 10^{-3}$ in Figure 2. Several observations can be made: (1) When training in onCL, a strong GI occurs, favoring stronger gradients for earlier classes than for later classes. (2) When introducing prototypes (+ *P*), despite a gain in performance, such an imbalance is increased. This behavior is expected as the prototype induces an additional gradient corresponding to older classes when training on the current task. (3) FGH reverses and reduces the imbalance when compared with using prototypes, leading to larger gradient values for the later classes. We argue that this imbalance is favorable because a larger LR usually implies rapid adaptability of the model, which is desired for newer classes, while older classes typically require lower gradients for more stability. Coefficients of variation are given in the appendix for a more detailed analysis of this behavior.

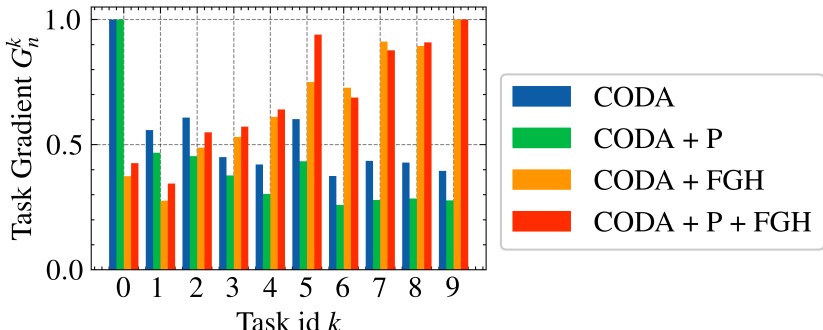

Figure 2: Values of the average normalized gradients per task $G_n^k$ for CODA on CIFAR100, 10 tasks. When including FGH, we display the resulting gradient *after* multiplying by the coefficients.

| Task | L2P | L2P+P | L2P+FGH | L2P+P+FGH |
|------|-------|--------|----------|------------|
| 1 | 4.51 | 44.50 | 6.46 | **65.77** |
| 2 | 6.01 | 53.14 | 9.47 | **70.37** |
| 3 | 12.44 | 52.87 | 16.32 | **67.65** |
| 4 | 16.15 | 55.65 | 23.01 | **67.59** |
| 5 | 21.37 | 49.55 | 30.11 | **61.33** |
| 6 | 28.82 | 51.25 | 39.19 | **61.40** |
| 7 | 25.11 | 44.49 | 36.20 | **51.86** |
| 8 | 34.34 | 46.98 | 46.26 | **53.13** |
| 9 | 32.75 | 37.89 | **42.71** | 41.60 |
| 10 | 30.16 | 29.04 | **39.34** | 29.14 |

Table 4: Task-wise performance of L2P on CIFAR100 with an initial LR of $5 \times 10^{-5}$.

**Stability-Plasticity Trade-Off**   To better understand the fine-grained dynamics of our approach, we provide a task-wise performance breakdown in Table 4. While the aggregate metrics in Tables 1 and 2 demonstrate our overall superiority, these per-task results reveal the individual contributions of the prototype (P) and FGH components. Specifically, we observe that the addition of prototypes (*L2P+P*) significantly boosts performance on earlier tasks compared to the baseline, though it still faces challenges as the sequence progresses. Conversely, the inclusion of FGH (*L2P+FGH*) provides a more consistent stabilization of the gradient updates, particularly as task complexity increases in the later stages. The synergy is most evident in the *L2P+P+FGH* configuration, which achieves the highest accuracy floor across nearly the entire stream. This confirms that while prototypes help anchor class representations, FGH provides the necessary gradient alignment to prevent new updates from interfering with that stored knowledge. This granular view demonstrates that our method's success is rooted in balancing immediate plasticity with long-term stability across all observed tasks.

**Underlying Intuition.**   To illustrate how FGH mitigates GI, Figure 3 shows task-specific gradient values with and without FGH. We observe that gradients are amplified more strongly for later tasks than for earlier ones, a trend confirmed across methods (see Appendix). According to the update rule in equation 7, gradients for early tasks change direction frequently, leading to smaller coefficient growth, whereas later tasks produce more stable gradients and thus larger coefficients. Intuitively, this mechanism down-weights unstable, high-magnitude gradients from early tasks while emphasizing the smaller, steadier gradients of later tasks, thereby correcting the imbalance.

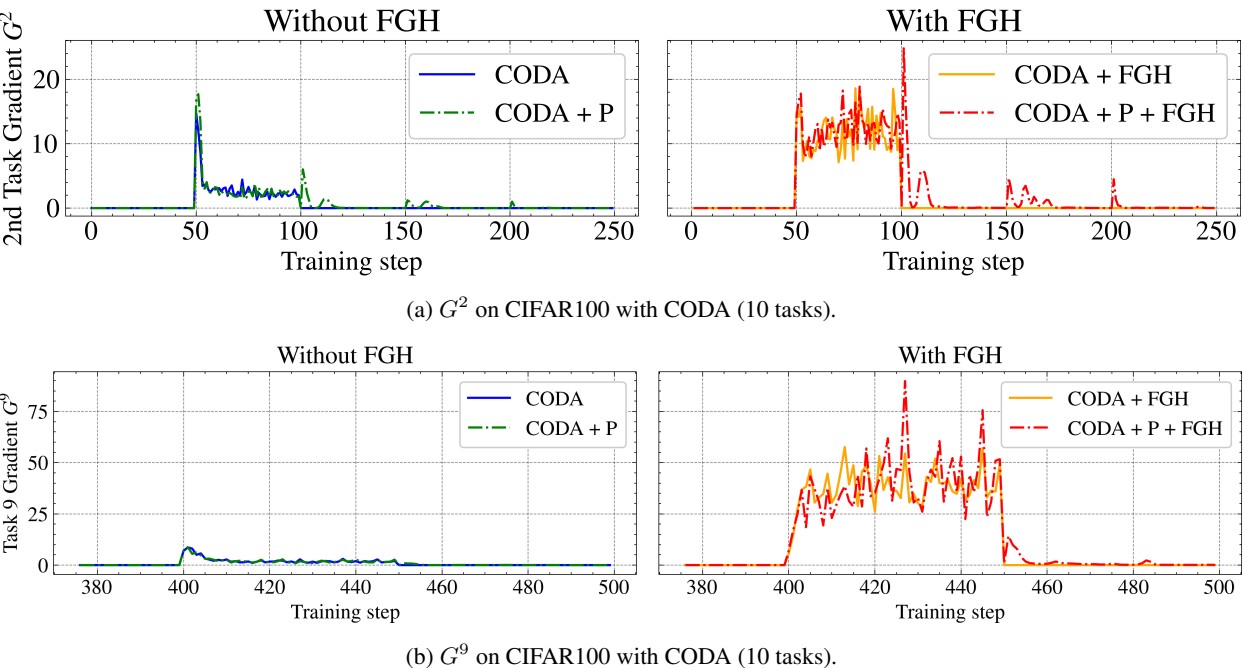

(a) $G^2$ on CIFAR100 with CODA (10 tasks).

(b) $G^9$ on CIFAR100 with CODA (10 tasks).

Figure 3: Values of gradients for CODA on CIFAR100 with 10 tasks, with and without prototypes and FGH. When including FGH, we show the resulting gradient *after* multiplication by the coefficients.

## 5.2   Selecting $\gamma$

The main drawback of leveraging *FGH* is the addition of an extra hyperparameter $\gamma$. To provide some additional insight into the impact of $\gamma$ on the final performance, we experiment with $\gamma \in \{10^{-6}, 10^{-5}, \cdots, 1, 10\}$ and show the results in Figure 4. It is important to note that $\gamma = 0$ is equivalent to disabling the FGH mechanism. Therefore, it can be observed that for all methods, on both datasets, larger values of $\gamma$ lead to substantial improvement over the baselines. Nonetheless, higher values of $\gamma$ may lead to unstable training due to high gradients. Therefore, we set $\gamma = 1$ for all experiments by default. Even though FGH introduces an additional hyperparameter, its impact is positive in all cases when combined with prototypes.

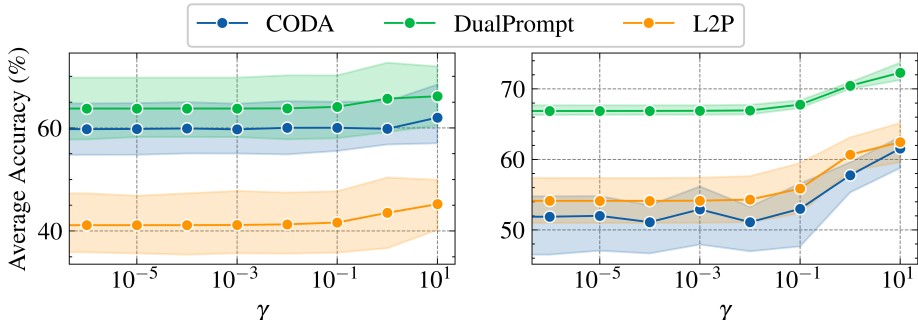

Figure 4: Average Accuracy (%) on VTAB (left) and CUB (right), in the *Si-blurry* setting, with an class increment of 5 classes per task, an LR of $5 \times 10^{-5}$, for CODA, DualPrompt, and L2P combined with prototypes and FGH, for varying values of $\gamma$.

## 6 Conclusion

In this paper, we tackled the problem of Online Memory-Free Task-Free Continual Learning, a particularly realistic problem. In that sense, we propose to narrow the gap between offCL and onCL research fields by adapting state-of-the-art offCL methods to the onCL problem by leveraging prototypes as a simple memory replacement. However, such a strategy increases gradient imbalance towards earlier classes and results in biased training. Moreover, limitations regarding the choice of the optimal LR remain unaddressed. Therefore, we leveraged Fine-Grained Hypergradients (FGH) as a mechanism for regulating Gradient Imbalance and LR sensitivity in online continual learning. Our method consistently outperforms existing memory-free onCL approaches, such as MVP and oLoRA, across a wide range of experimental settings. The synergy between these components enables more efficient and balanced learning throughout the training process. Overall, our results demonstrate significant performance improvements, encouraging further connections between offCL and onCL research. Eventually, this approach offers a promising path towards scalable and efficient online learning solutions.

## Acknowledgments

This work was partially financially supported by JST ASPIRE Program, Japan, Grant Number JPMJAP2303. This work was partially supported by the JSPS Postdoctoral Fellowship for Research in Japan (Fellowship ID: P24752).

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

# A  Implementation and algorithm

## A.1  Implementation

For our implementation, we rely on the LAMDA-PILOT repository (Sun et al. (2025)), available at `https://github.com/LAMDA-CL/LAMDA-PILOT`. The implementation of existing methods was adapted to an online scenario.

## A.2  Algorithm

The implementation that we used for our experiments is based on an Adam update. For the sake of clarity, we present our method with SGD and Adam. We omitted the bias, logits mask, and coefficient clamping from the pseudo-code. Therefore, we give the full details of the procedure in Algorithms 2 and 1, in a pseudo-code Pytorch-like implementation.

## A.3  Backbone

We leverage a ViT-base (Dosovitskiy et al. (2021)), pre-trained on ImageNet-21k. Precisely, we use the implementation of the *timm* library, available at `https://huggingface.co/timm`, with model name "vit_base_patch16_224".

## A.4  Batch Wise Logits Mask

Another key component when training offline is the usage of a logits mask. Let $\mathbf{z} \in \mathbb{R}^c$ denote the logits output of the trained model. In the offline case, the logits mask $\mathbf{m}$ is defined such that

$$\mathbf{m}_j = \begin{cases} 0, & \text{if } j \in \mathcal{Y}, \\ -\infty, & \text{otherwise.} \end{cases}$$

With $\mathcal{Y}$, the ensemble of classes that the model has been exposed to at the current time of training. The masked logits are then computed as

$$\tilde{\mathbf{z}} = \mathbf{z} + \mathbf{m}.$$

In the blurry boundaries setting, classes can appear and disappear several times during training and across tasks. In that sense, we adopt a more flexible version of the logits mask where $\mathcal{Y} = \mathcal{Y}_{batch}$. With $\mathcal{Y}_{batch}$, the set of all classes present in the current batch.

## A.5  Impact of LR on the Stability-Plasticity trade-off.

It is clear that selecting an appropriate learning rate is essential for optimal performance. In standard scenarios, the impact of its choice on loss minimization and convergence speed has been extensively studied (Ruder (2016)). For offCL, previous studies have considered to impact of the LR on forgetting (Mirzadeh et al. (2020)). Notably, a higher LR would increase forgetting, and vice versa. Intuitively, the learning rate gives direct control on the plasticity-stability tradeoff (Wang et al. (2024b)). To confirm such behavior in onCL, we experiment with larger and smaller LR values. As can be seen in Figure A5, when trained with a higher learning rate ($5 \times 10^{-2}$), the model tends to obtain higher performances on the latest tasks while exhibiting especially low performances on earlier tasks. When trained with a lower LR ($5 \times 10^{-5}$), the model tends to achieve better performance on earlier tasks compared to training with a higher LR. In other words, a high LR value induces more plasticity and less stability, and vice versa.

## A.6  Hyperparameters Grid Searched on VTAB

In the presented results, we display a *Best HP* column, which corresponds to the results obtained for the best hyperparameters obtained on VTAB. The objective is to simulate a realistic scenario where the online continual learning datasets are not available for hyperparameter search. Therefore, a realistic solution is to conduct a grid search on an available dataset and hopefully successfully transfer the found hyperparameters to the new datasets. In this work, we search only for the value of the learning rate and $\gamma$ when combining with FGH. The hyperparameters explored for all methods are presented in Table B5.

---

**Algorithm 2** PyTorch-like pseudo-code of our Adam-based method integration with other baselines. Extra details are given in this version regarding bias consideration and batch-wise masking.

```python
# Adam parameters
m, v, beta1, beta2, step = 0, 0, 0.9, 0.999, 0

# Hypergrad parameters
gamma = 1e-3
grad_weight = torch.ones(n_classes)
prev_grad = None
for x, y in dataloader:
    # Baseline loss
    h, logits_base = network(x) # features and logits
    # Batch-wise masking
    mask = [i for i in range(logits_b.shape[-1]) if i not in y.unique()]
    logits_b[:, mask] = float('-inf')
    loss_baseline = criterion_baseline(logits_b, y)

    # FC recalibration
    proto, labels = get_prototypes()
    logits = network.fc(proto)
    # Batch-wise masking
    mask = [i for i in range(logits.shape[-1]) if i not in labels.unique()]
    logits[:, mask] = float('-inf')
    loss_op = cross_entropy(logits, labels)

    loss = loss_baseline + loss_op

    optim.zero_grad()
    loss.backward()

    # Class-Wise Hypergradient update
    curr_W = network.fc.weight.grad
    curr_B = network.fc.bias.grad
    curr_grad = torch.cat([curr_W, curr_B.unsqueeze(1)], dim=1)
    if prev_grad is not None:
        # Adam update
        m = beta1 * m + (1 - beta1) * curr_grad
        v = beta2 * v + (1 - beta2) * (curr_grad ** 2)
        m_hat = m / (1 - beta1 ** step)
        v_hat = v / (1 - beta2 ** step)
        curr_grad = m_hat / (torch.sqrt(v_hat) + 1e-8)

        grad_weight += gamma * (curr_grad @ prev_grad.T).diag() #Eq. 7
        for i in range(n_classes):
            network.fc.weight.grad[i, :] = network.fc.weight.grad[i, :] * grad_weight[i]
            network.fc.bias.grad[i] = network.fc.bias.grad[i] * grad_weight[i]
    prev_grad = curr_grad
    optim.step()

    update_proto(h, y) # Eq. 1
```

---

# B   Datasets and Baselines

## B.1   Datasets

The backbone used for all our experiments has been pre-trained on ImageNet-21k, making it unfair to experiment on such datasets. Following previous work (Sun et al. (2025)), we showcase the performance of our approach and experiment with the following datasets:

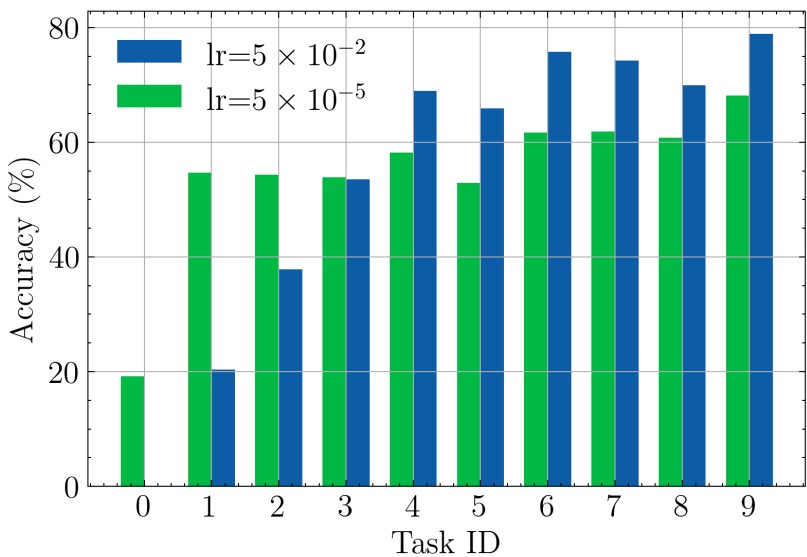

Figure A5: Task-wise accuracy (%) of DualPrompt at the end of training on CIFAR100, split in 10 tasks, for LR values in $\{5 \times 10^{-5}, 5 \times 10^{-2}\}$, with a batch size of 10.

Table B5: Hyperparameters tested on VTAB, clear setting, an increment of 5 classes per task. Hyperparameters used for *Best HP* as written in bold.

| Method | Learning Rate | $\gamma$ |
|---|---|---|
| Fine-tuning | [0.001, 0.005, **0.01**, 0.05, 0.1] | N/A |
| Linear probe | [0.001, **0.005**, 0.01, 0.05, 0.1] | N/A |
| ER | [0.00001, **0.00005**, 0.0001, 0.0005, 0.001, 0.005, 0.01, 0.05, 0.1] | N/A |
| ER + Linear probe | [0.001, 0.005, **0.01**, 0.05, 0.1] | N/A |
| MVP | [0.0001, 0.0005, 0.001, **0.005**, 0.01, 0.05, 0.1] | N/A |
| oLoRA | [0.001, **0.005**, 0.01, 0.05, 0.1] | N/A |
| CODA | [0.0001, 0.0005, 0.001, 0.005, **0.01**, 0.05, 0.1] | [0.00001, 0.0001, 0.001, 0.01, **0.1**, 1, 10] |
| ConvPrompt | [0.0001, 0.0005, 0.001, **0.005**, 0.01, 0.05, 0.1] | [0.00001, 0.0001, 0.001, 0.01, 0.1, **1**, 10] |
| DualPrompt | [0.0001, 0.0005, 0.001, 0.005, **0.01**, 0.05, 0.1] | [0.00001, 0.0001, 0.001, 0.01, 0.1, **1**, 10] |
| L2P | [0.0001, 0.0005, 0.001, 0.005, **0.01**, 0.05, 0.1] | [0.00001, 0.0001, 0.001, 0.01, 0.1, **1**, 10] |

- **CUB** (Wah et al. (2011)): The CUB dataset (Caltech-UCSD Birds-200) contains 200 bird species with 11,788 images, annotated with attributes and part locations for fine-grained classification. We use an increment of 10 classes per task, resulting in 20 tasks (with 10 classes per task).

- **ImageNet-R** (Hendrycks et al. (2021)): ImageNet-R is a set of images labeled with ImageNet label renditions. It contains 30,000 images spanning 200 classes, focusing on robustness with images in various artistic styles. We use an increment of 10 classes per task, resulting in 20 tasks (with 10 classes per task).

- **CIFAR-100** (Krizhevsky (2012)): CIFAR-100 consists of 60,000 $32 \times 32$ color images across 100 classes, with 500 images per class, split into 500 training and 100 test samples per class. We use an increment of 10 classes per task, resulting in 10 tasks (with 10 classes per task).

- **Visual Task Adaptation Benchmark (VTAB)** (Zhai et al. (2019)): VTAB contains the following 19 tasks that are derived from several public datasets. We use an increment of 5 classes per task, resulting in 10 tasks (with 5 classes per task).

## B.2 Baselines

**Offline methods adapted to Online** Prompt learning-based methods (Zhou et al. (2024a)) are particularly suited for being combined with our approach in onCL as they all capitalize on a final FC layer for classification. Therefore, we consider the following.

- **L2P** (Wang et al. (2022b)): Learning to Prompt (L2P) is the foundation of prompt learning methods in Continual Learning. The main idea is to learn how to append a fixed-sized prompt to the input of the ViT (Dosovitskiy et al. (2021)). The ViT stays frozen; only the appended prompt as well as the FC layer are trained.

- **DualPrompt** (Wang et al. (2022a)): DualPrompt follows closely the work of L2P by addressing forgetting in the prompt level with task-specific prompts as well as higher-level long-term prompts.

- **CODA** (Smith et al. (2023)): CODA-prompt improves prompt learning by computing the prompt on the fly, leveraging a component pool and an attention mechanism. Therefore, CODA benefits from a single gradient flow.

- **ConvPrompt** (Roy et al. (2024)): ConvPrompt leverages convolutional prompts and dynamic task-specific embeddings while incorporating text information from large language models.

**Online memory-free and task-free methods**

- **MVP** (Moon et al. (2023)): MVP uses learned instance-wise logit masking, contrastive visual prompt tuning for Continual Learning in the *Si-Blurry* context.

- **Online LoRA (oLoRA)** (Wei et al. (2025)): Trains a LoRA (Hu et al. (2022)) module for each task in the online task-free setting by detecting task-change by estimating the convergence of the model.

**Mainstream baselines** Additionally, we considered traditional baselines when working with continual learning methods:

- **Fine-tuning**: Straightforward fine-tuning where the backbone is fine-tuned on new tasks by training all the present weights without any specific constraint

- **Linear probe**: Fine-tuning training where only the last fully connected (FC) layer is trained. All other weights are frozen.

- **Experience Replay (ER)** (Rolnick et al. (2019)): A memory-based approach that reuses the experience of previous tasks to train the model on the new task. In our experiments, we limit the memory size to 1000 samples, and retrieve 100 samples at each training iteration.

- **ER + Linear probe**: This method consists of training a Linear probe (Alain & Bengio (2016)) method and incorporating an ER mechanism. In our experiments, we limit the memory size to 1000 samples, and retrieve 100 samples at each training iteration.

## C Adaptation of Methods to Our Setup

Since most methods compared here were originally designed for offCL, they had to be specifically adapted to the onCL scenario. In that sense, some parameters have been chosen arbitrarily, based on their offCL values, without additional hyperparameter search. Such a situation is similar to one that would be observed in real-world cases where an offCL model tries to be adapted to an onCL problem. For all methods, we use a learning rate, no scheduler, and Adam optimizer. Of course, we disabled an operation that was operated at task change. Additionally, even though MVP was indeed designed for online cases, we found several differences between their training procedure and ours, which we discuss below.

**Adaptation of CODA.** In their original paper and implementation (Smith et al. (2023)), the authors require freezing components after each task, therefore having task-specific components. Typically, they show that performances tend to plateau for more than 100 components, and for a 10-task sequence, they would reserve 10 components per task. In our implementation, we decided to similarly use 100 components for the entire training. However, we train all components together at all times during training since we cannot know when the correct time to freeze or unfreeze them. For other parameters, we followed the original implementation. Code adapted from `https://github.com/LAMDA-CL/LAMDA-PILOT`

**Adaptation of ConvPrompt.** ConvPrompt (Roy et al. (2024)) is a method that heavily relies on task boundaries in its original implementation, notably by incorporating five new prompts per task. Contrary to CODA, allocating the maximum number of prompt generators at all times, without a freeze, would induce an important training time constraint. Therefore, we only use five prompt generators at all times. Despite this reduction in overall parameters, ConvPrompt still achieves competitive results in the *clear* setting. However, its performances drastically fall off in the *Si-Blurry* case. Further, an in-depth adaptation of ConvPrompt in the online context could potentially improve its performance; however, such a study is not covered in this work. Code adapted from `https://github.com/CVIR/convprompt`.

**Adaptation of DualPrompt.** Similar to CODA, but on a prompt level, DualPrompt (Wang et al. (2022a)) requires freezing prompts at task change. For adapting it to onCL, we chose to use all prompts at all times without freezing the prompt pool. The E-Prompt pool size is set to 10 and the G-Prompt pool size is set to 5. Code adapted from `https://github.com/LAMDA-CL/LAMDA-PILOT`.

**Adaptation of L2P.** The same logic as the one described for CODA and DualPrompt applies to L2P (Wang et al. (2022b)). In that sense, we use the entire prompt pool at all times without freezing. The prompt pool size is set to 10. Code adapted from `https://github.com/LAMDA-CL/LAMDA-PILOT`.

**Adaptation and Performances of MVP** Even though MVP (Moon et al. (2023)) is designed for the online case, its original training setup differs slightly. Firstly, the batch size is set to 32 (we use 100), and they similarly consider that each batch can be used for 3 separate gradient steps. In that sense, the performances reported in the original paper might be higher as they trained on a slightly more constrained setup. Secondly, the authors use the same learning rate and optimizer for each compared method, which, as we argued in this work, might lead to different results, relatively speaking, compared to other methods. Such experimental differences might lead to the performances obtained in our experiments, which are, in most cases, surprisingly low. The code was adapted from `https://github.com/KU-VGI/Si-Blurry`.

**Adaptation and Performances of oLoRA** Even though oLoRA (Wei et al. (2025)) is designed for online problems, it relies on several hyperparameters. Notably, it requires computing a moving average of the current loss, which, depending on the batch size and task size, can lead to significantly different results. For example, on the CUB dataset, a task consists of 400 images. In our setup, the batch size is 100, so the default window size of 5 would span over multiple tasks. Such behavior makes the working mechanism of oLoRA very sensitive to the setup. Other hyperparameters include variance and mean loss threshold for triggering loss change detection. Similarly, this is very dependent on the dataset. Lastly, a loss weighting term must be grid-searched for optimal results. Code adapted from `https://github.com/christina200/online-lora-official`.

# D  Additional Evaluation Metrics

Here, we report additional metrics in the clear and blurry boundary contexts for all methods for additional insights into the performance.

## D.1  Average Performances

We follow previous work and define the Average Performance (AP) as the average of the accuracies computed after each task during training (Zhou et al. (2024a)). Formally, when training on $\{\mathcal{D}_1, \cdots, \mathcal{D}_T\}$, we define $\mathcal{A}_k = \frac{1}{k}\sum_{l=1}^{k} a_{l,k}$ as the Average Accuracy (AA), with $a_{l,k}$ being the accuracy on task $l$ after training on $\mathcal{D}_k$. Building on this, we define the

Table B6: Time and Spatial complexity of compared methods on CUB in the *clear* setting, with a batch size of 100.

| Method | Time (min) | Memory Footprint (MB) |
|---|---|---|
| Fine-tuning | 3m26s | 17,089 |
| Linear probe | 2m40s | 2,566 |
| ER | 4m54s | 34,481 |
| ER + Linear probe | 3m12s | 4,647 |
| MVP | 5m22s | 12,722 |
| oLoRA | 5m20s | 56,357 |
| CODA | 5m37s | 16,923 |
| $\hookrightarrow$ + P | 5m47s | 16,921 |
| $\hookrightarrow$ + FGH | 5m54s | 18,287 |
| $\hookrightarrow$ + ours | 5m55s | 18,288 |
| L2P | 5m32s | 14,090 |
| $\hookrightarrow$ + P | 5m35s | 14,092 |
| $\hookrightarrow$ + FGH | 5m43s | 14,090 |
| $\hookrightarrow$ + ours | 5m43s | 14,092 |
| DualPrompt | 5m12s | 11,827 |
| $\hookrightarrow$ + P | 5m14s | 11,829 |
| $\hookrightarrow$ + FGH | 5m23s | 11,828 |
| $\hookrightarrow$ + ours | 5m18s | 11,829 |
| ConvPrompt | 1h12m24s | 11,708 |
| $\hookrightarrow$ + P | 1h12m33s | 11,709 |
| $\hookrightarrow$ + FGH | 1h12m22s | 11,708 |
| $\hookrightarrow$ + ours | 1h12m40s | 11,709 |

Average Performance (AP) as:

$$\mathcal{P} = \frac{1}{T} \sum_{k=1}^{T} \mathcal{A}_k. \tag{9}$$

## D.2 Final Average Accuracy

We report the final average accuracy $\mathcal{A}_T$ as per the definition given in the main draft. Such results are presented in Tables B10 and B11.

## D.3 Performances on Previous Tasks

We report the accuracy at the end of training on previous tasks when training in the *clear* setting. Notably, show the accuracy for each method on the first 10 tasks in Table B12. It can be observed that for earlier tasks, leveraging FGH and Prototypes (+ *ours*) leads to the best performances on older tasks, see for example the performances of CODA on CIFAR-100 on the first task, presented in Tables B12 to B21.

## D.4 Time Complexity

Experiments were run on various machines, including Quadro RTX 8000 50Go GPU, Tesla V100 16Go GPU, and A100 40Go GPU. In this section, we report the times of execution of each method. To do so, we run all methods (except oLoRA) on a single Quadro RTX 8000 50Go GPU, for the CUB dataset, clear setting, with a batch size of 100. Since oLoRA requires a large amount of GPU memory, we have to evaluate its training time and memory consumption on two Quadro RTX 8000 50Go GPUs. The results are presented in Table B6. It can be observed that the time consumption overhead of including *prototypes* and *FGH* is minimal.

### D.5  Spatial Complexity

**Fine-Grained Hypergradients.**  The usage of FGH requires storing one float per trainable parameter $D$ as well as previous gradient values of those parameters. This amounts to a total of $2 \times D$ additional floats to store. We show memory footprint on GPU in Table B6 using a Quadro RTX 8000 50Go GPU, on the CUB dataset, clear setting, with a batch size of 100.

**Prototypes.**  Storing prototypes only requires one vector of dimension $l$ per class, with $l = 768$ in the case of ViT-base. Additionally, an extra integer per class must be stored to keep track of the index of the update of each class-dependent prototype. If the index is stored as a float, the additional amount of floating points to store is $c \times (l + 1)$, with $c$ the number of classes, and $l$ the output dimension of the backbone. We show memory footprint on GPU in Table B6 using a Quadro RTX 8000 50Go GPU, on CUB dataset, clear setting, with a batch size of 100.

## E  Additional Experiments

### E.1  Details on Gradient Imbalance

In the following, we give additional insights into the results displayed in Figure 2 regarding GI. In this regard, we computed the coefficient of variation across normalized class-wise gradients. Namely, we compute the $std/mean$ ratio on the data presented in said Figure. The results are presented in Table B7.

| Method | Coefficient of Variation | Comments | Perfs |
|---|---|---|---|
| CODA | 0.3353 | Baseline | 71.12 |
| CODA + P | 0.5069 | GI increased | 78.18 |
| CODA + FGH | 0.3488 | GI reversed and slightly increased | 69.66 |
| CODA + P + FGH | 0.3107 | GI reversed and decreased compared to P | 79.47 |

Table B7: Comparison of Methods with Coefficient of Variation and Performance

### E.2  Longer Task Sequence

We conducted brief experiments regarding the performance of CODA + ours on Imagenet-R, non-blurry, with an increment of 2 classes per task, for various values of $\gamma$, with and without prototypes. The results are displayed in Table B8.

| $\gamma$ | $+P$ | **Average Performances** |
|---|---|---|
| 0 | 0 | 32.7 |
| 0.5 | 0 | 30.35 |
| 1 | 0 | 32.7 |
| 0 | 1 | 38.9 |
| 0.5 | 1 | **44.4** |
| 1 | 1 | 38.1 |

Table B8: Average Performances for Different Values of $\gamma$ and $+P$

### E.3  Additional Memory Sizes

In the main paper, all memory sizes are limited to 1000 for ER-based methods. In the following, we show additional performances for ER with larger memory sizes on CIFAR100 and Imagenet-R for an initial LR of $5 \times 10^{-5}$. The results are presented in Table B9.

Table B9: Performance of ER on CIFAR100 and ImageNet-R with varying memory sizes and a learning rate of $5 \times 10^{-5}$.

| Dataset | 1000 | 5000 | 10000 |
|---|---|---|---|
| CIFAR100 (ER) | 81.20 | 85.88 | 86.94 |
| ImageNet-R (ER) | 53.35 | 58.30 | 58.63 |

## F   Details on the Si-Blurry Setting

We followed the procedure and code made available by the authors of MVP (Moon et al. (2023)) in order to generate the *Si-Blurry* versions of the datasets. Notably, we use $M = 10$ and $N = 50$, following the original work. The number of tasks is the same as in the *clear* setting. We show the number of images per class appearance during training for a subset of classes to give a better overview of this training environment in Figure A6.

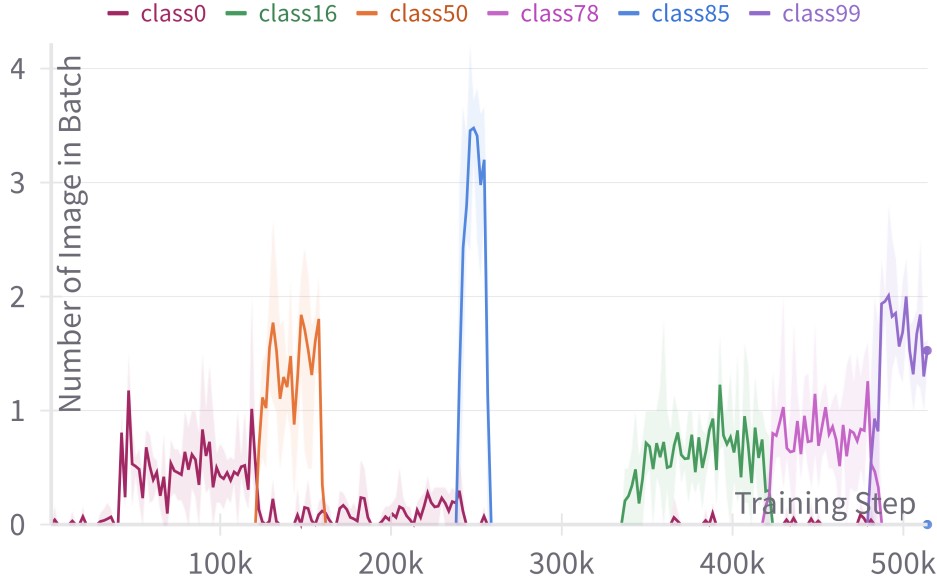

Figure A6: Example of class appearance during training in the *Si-Blurry* setting on CIFAR100. The y-axis represents the average number of images of a given class present in the current batch size of 10.

## G   Additional Gradient Values

Following the analysis on the interactions between FGH and prototypes with regard to past gradients, we include the gradients norm of previous task for more tasks and methods in Figure A7 to Figure A36.

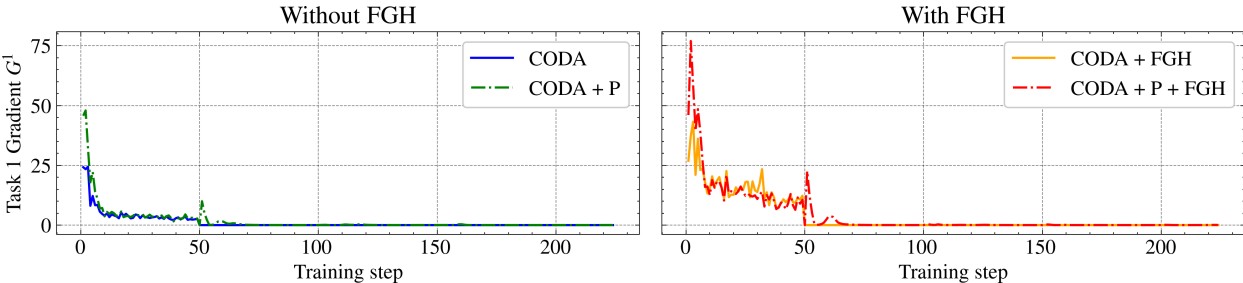

Figure A7: Values of $G^1$ for CODA on CIFAR100, 10 tasks, with and without prototypes and FGH. When including FGH, we display the resulting gradient *after* multiplying by the coefficients. Task changes every 50 steps. Only 250 steps are displayed for readability.

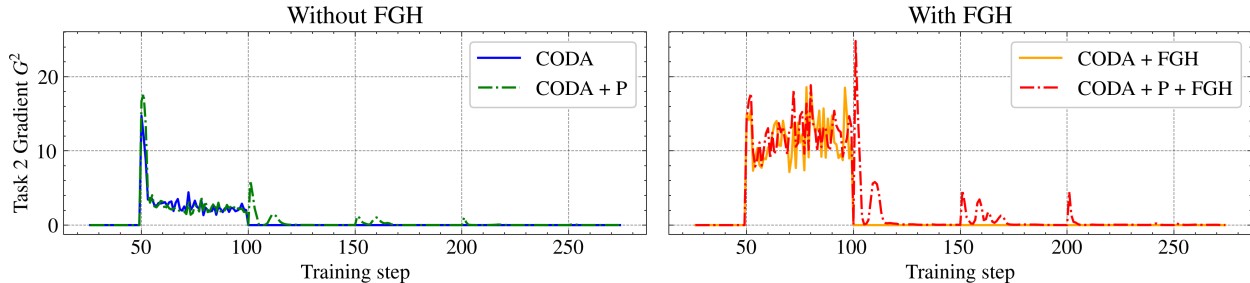

Figure A8: Values of $G^2$ for CODA on CIFAR100, 10 tasks, with and without prototypes and FGH. When including FGH, we display the resulting gradient *after* multiplying by the coefficients. Task changes every 50 steps. Only 250 steps are displayed for readability.

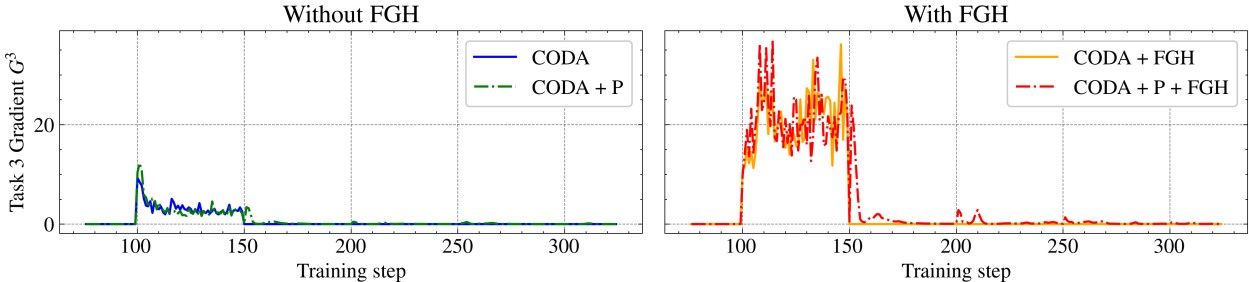

Figure A9: Values of $G^3$ for CODA on CIFAR100, 10 tasks, with and without prototypes and FGH. When including FGH, we display the resulting gradient *after* multiplying by the coefficients. Task changes every 50 steps. Only 250 steps are displayed for readability.

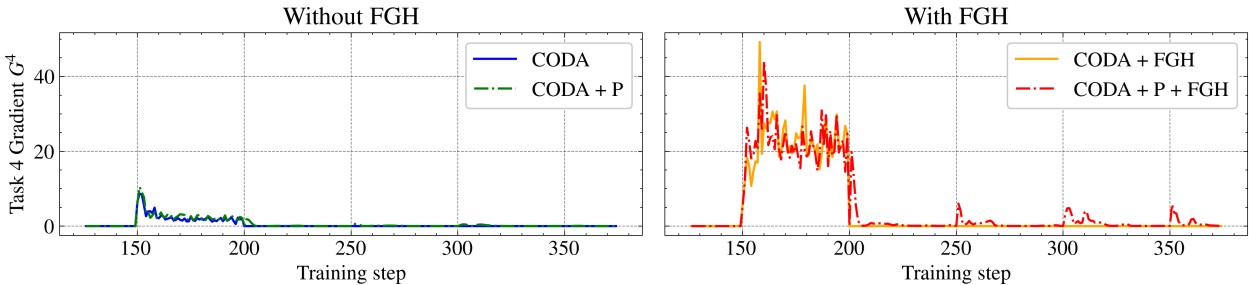

Figure A10: Values of $G^4$ for CODA on CIFAR100, 10 tasks, with and without prototypes and FGH. When including FGH, we display the resulting gradient *after* multiplying by the coefficients. Task changes every 50 steps. Only 250 steps are displayed for readability.

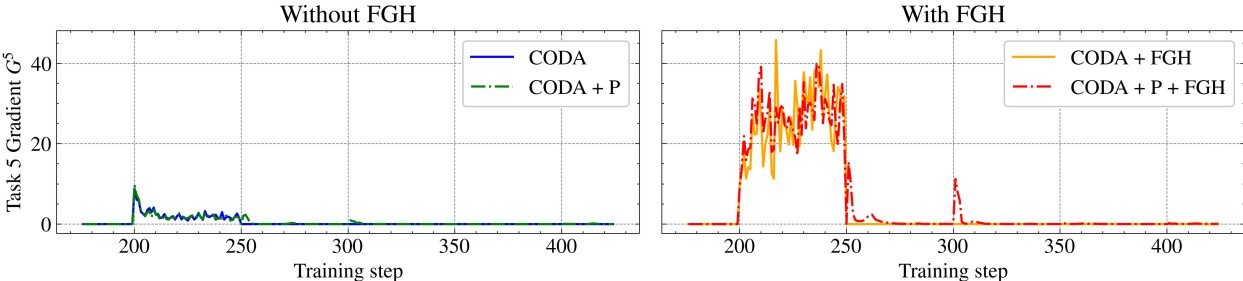

Figure A11: Values of $G^5$ for CODA on CIFAR100, 10 tasks, with and without prototypes and FGH. When including FGH, we display the resulting gradient *after* multiplying by the coefficients. Task changes every 50 steps. Only 250 steps are displayed for readability.

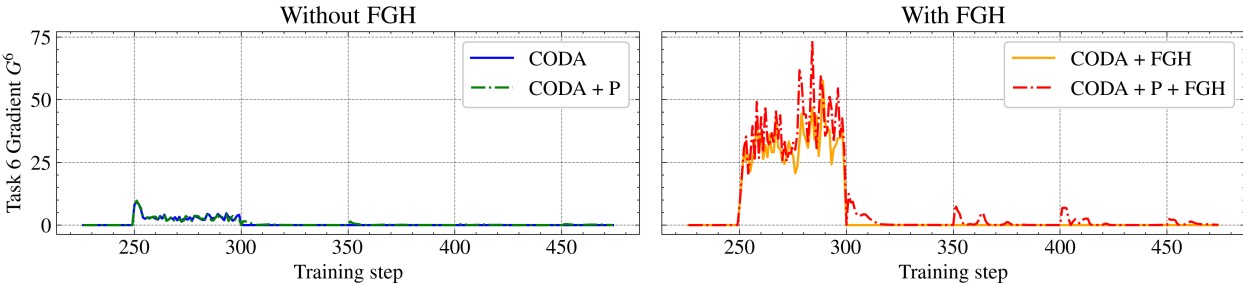

Figure A12: Values of $G^6$ for CODA on CIFAR100, 10 tasks, with and without prototypes and FGH. When including FGH, we display the resulting gradient *after* multiplying by the coefficients. Task changes every 50 steps. Only 250 steps are displayed for readability.

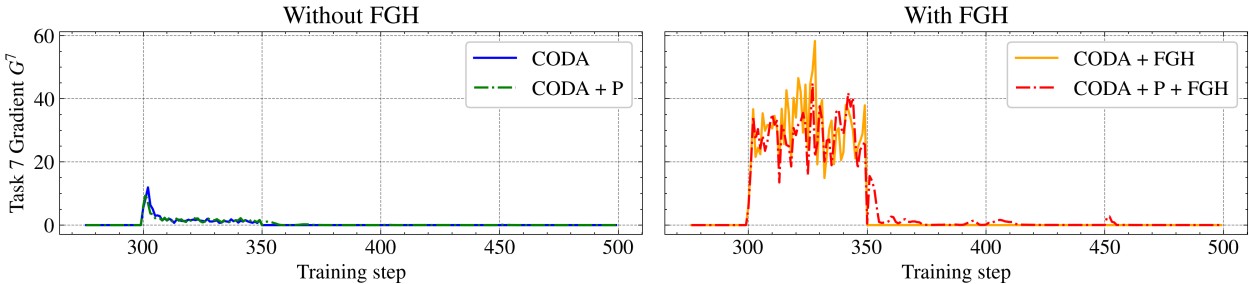

Figure A13: Values of $G^7$ for CODA on CIFAR100, 10 tasks, with and without prototypes and FGH. When including FGH, we display the resulting gradient *after* multiplying by the coefficients. Task changes every 50 steps. Only 250 steps are displayed for readability.

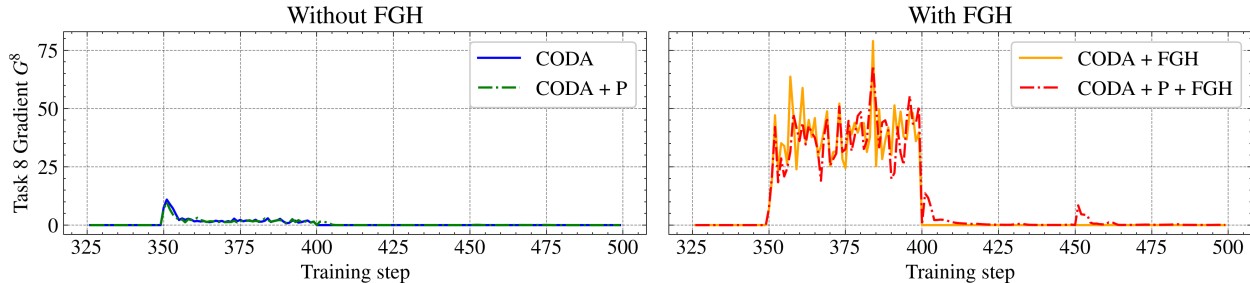

Figure A14: Values of $G^8$ for CODA on CIFAR100, 10 tasks, with and without prototypes and FGH. When including FGH, we display the resulting gradient *after* multiplying by the coefficients. Task changes every 50 steps. Only 250 steps are displayed for readability.

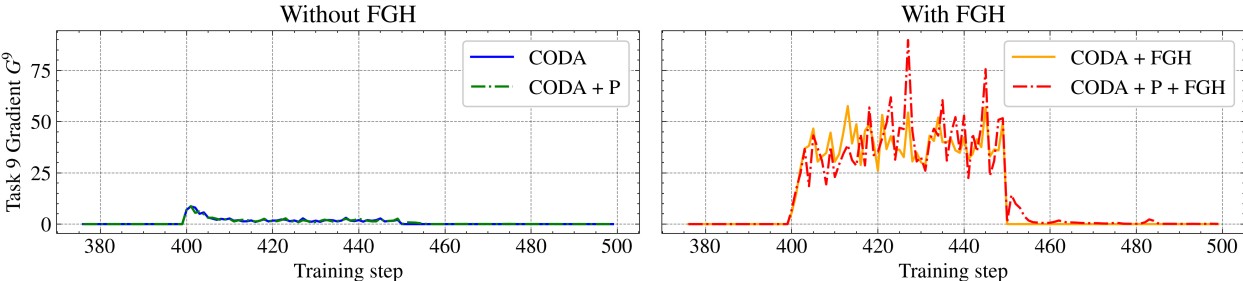

Figure A15: Values of $G^9$ for CODA on CIFAR100, 10 tasks, with and without prototypes and FGH. When including FGH, we display the resulting gradient *after* multiplying by the coefficients. Task changes every 50 steps. Only 250 steps are displayed for readability.

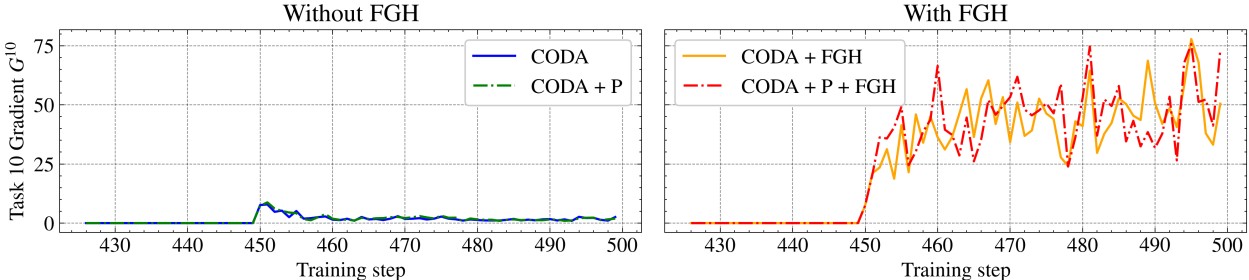

Figure A16: Values of $G^{10}$ for CODA on CIFAR100, 10 tasks, with and without prototypes and FGH. When including FGH, we display the resulting gradient *after* multiplying by the coefficients. Task changes every 50 steps. Only 250 steps are displayed for readability.

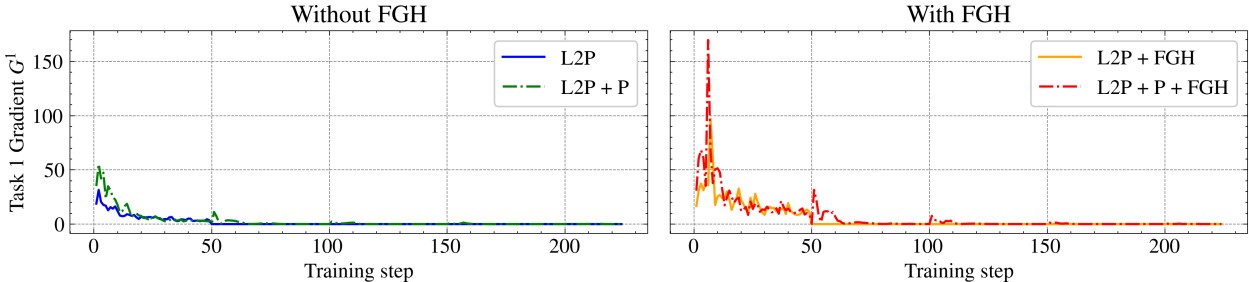

Figure A17: Values of $G^1$ for L2P on CIFAR100, 10 tasks, with and without prototypes and FGH. When including FGH, we display the resulting gradient *after* multiplying by the coefficients. Task changes every 50 steps. Only 250 steps are displayed for readability.

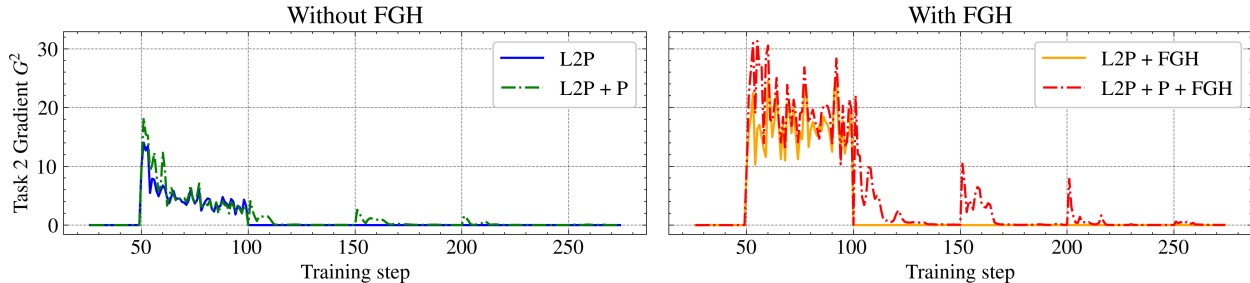

Figure A18: Values of $G^2$ for L2P on CIFAR100, 10 tasks, with and without prototypes and FGH. When including FGH, we display the resulting gradient *after* multiplying by the coefficients. Task changes every 50 steps. Only 250 steps are displayed for readability.

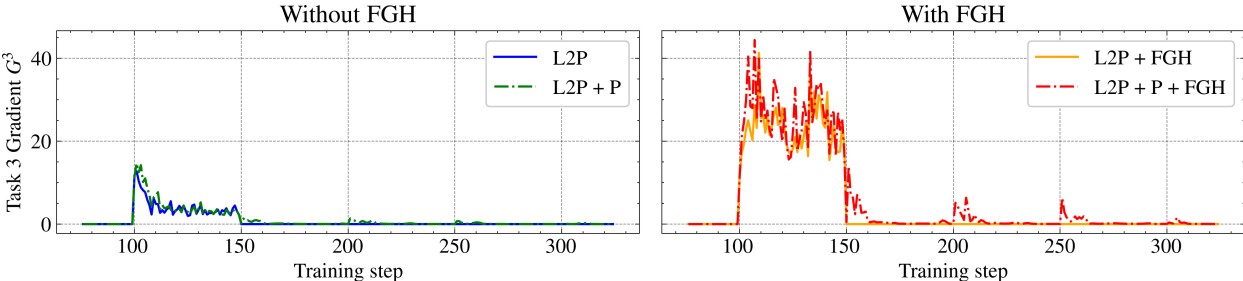

Figure A19: Values of $G^3$ for L2P on CIFAR100, 10 tasks, with and without prototypes and FGH. When including FGH, we display the resulting gradient *after* multiplying by the coefficients. Task changes every 50 steps. Only 250 steps are displayed for readability.

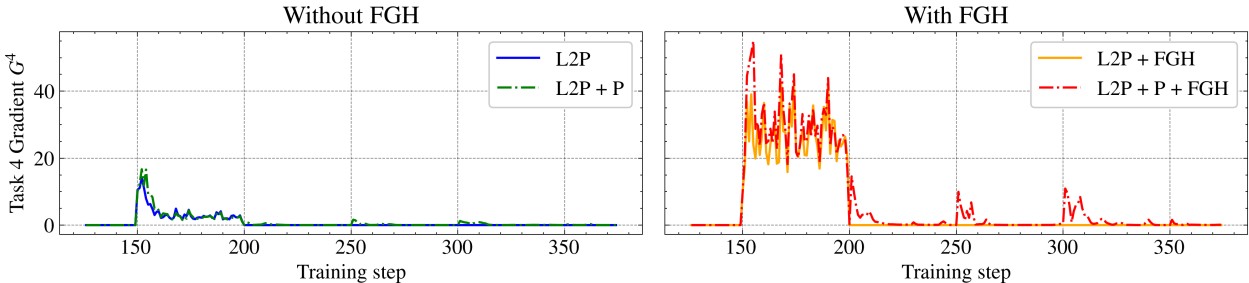

Figure A20: Values of $G^4$ for L2P on CIFAR100, 10 tasks, with and without prototypes and FGH. When including FGH, we display the resulting gradient *after* multiplying by the coefficients. Task changes every 50 steps. Only 250 steps are displayed for readability.

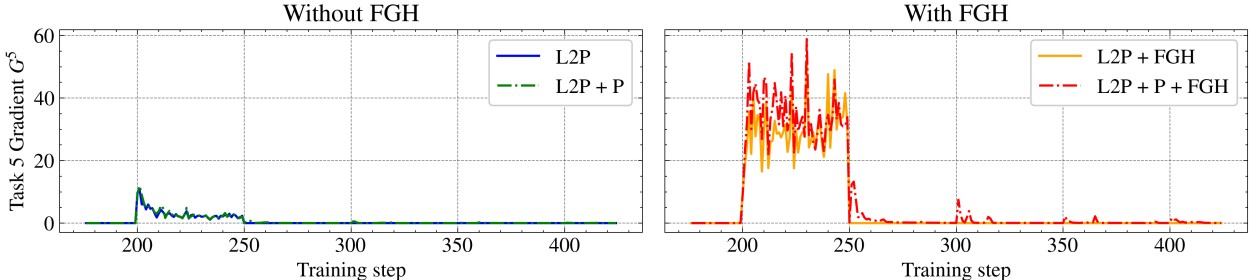

Figure A21: Values of $G^5$ for L2P on CIFAR100, 10 tasks, with and without prototypes and FGH. When including FGH, we display the resulting gradient *after* multiplying by the coefficients. Task changes every 50 steps. Only 250 steps are displayed for readability.

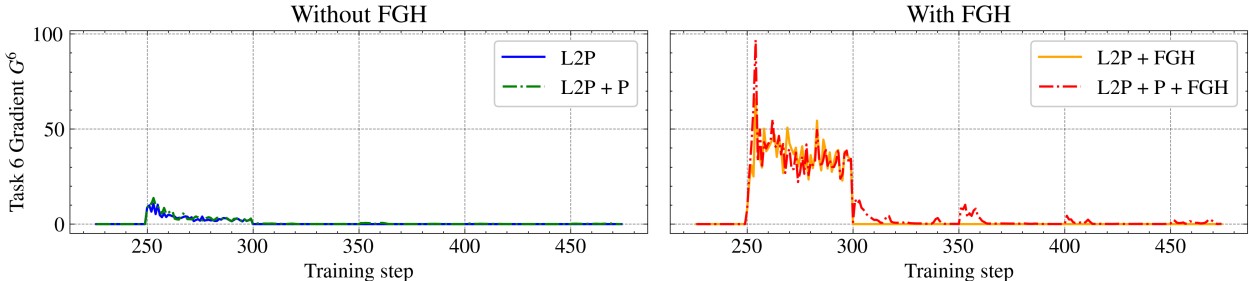

Figure A22: Values of $G^6$ for L2P on CIFAR100, 10 tasks, with and without prototypes and FGH. When including FGH, we display the resulting gradient *after* multiplying by the coefficients. Task changes every 50 steps. Only 250 steps are displayed for readability.

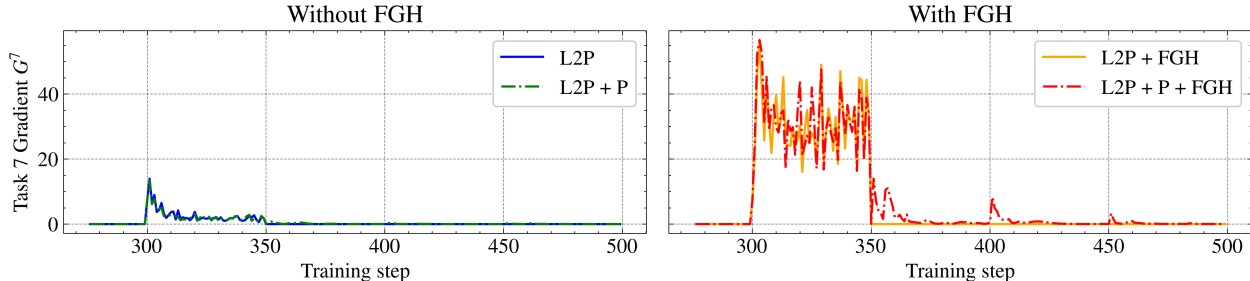

Figure A23: Values of $G^7$ for L2P on CIFAR100, 10 tasks, with and without prototypes and FGH. When including FGH, we display the resulting gradient *after* multiplying by the coefficients. Task changes every 50 steps. Only 250 steps are displayed for readability.

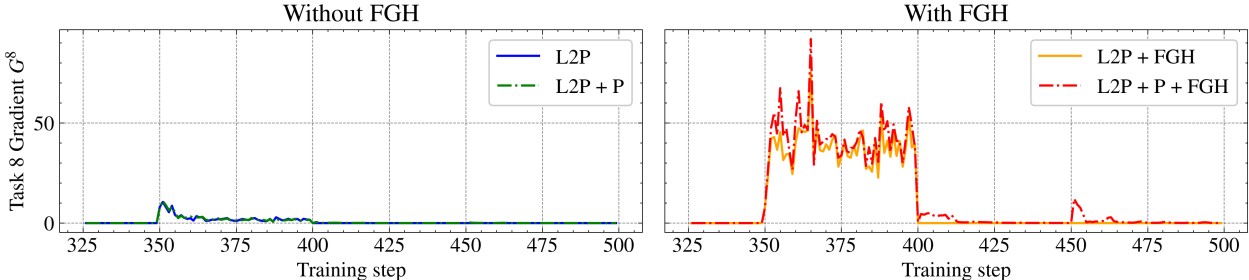

Figure A24: Values of $G^8$ for L2P on CIFAR100, 10 tasks, with and without prototypes and FGH. When including FGH, we display the resulting gradient *after* multiplying by the coefficients. Task changes every 50 steps. Only 250 steps are displayed for readability.

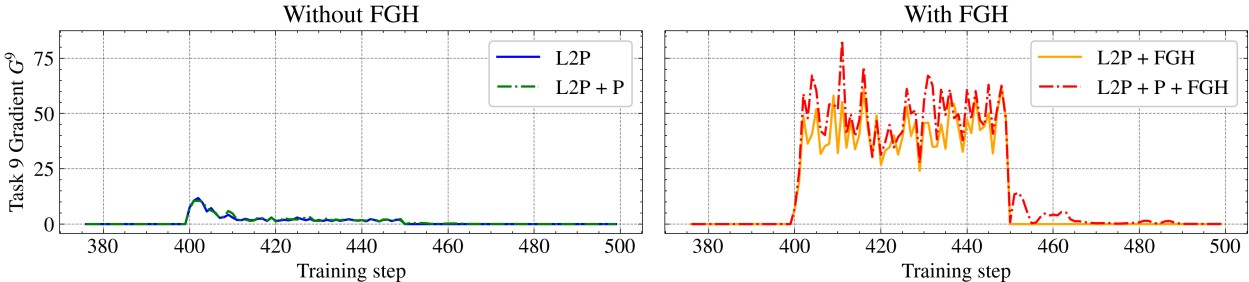

Figure A25: Values of $G^9$ for L2P on CIFAR100, 10 tasks, with and without prototypes and FGH. When including FGH, we display the resulting gradient *after* multiplying by the coefficients. Task changes every 50 steps. Only 250 steps are displayed for readability.

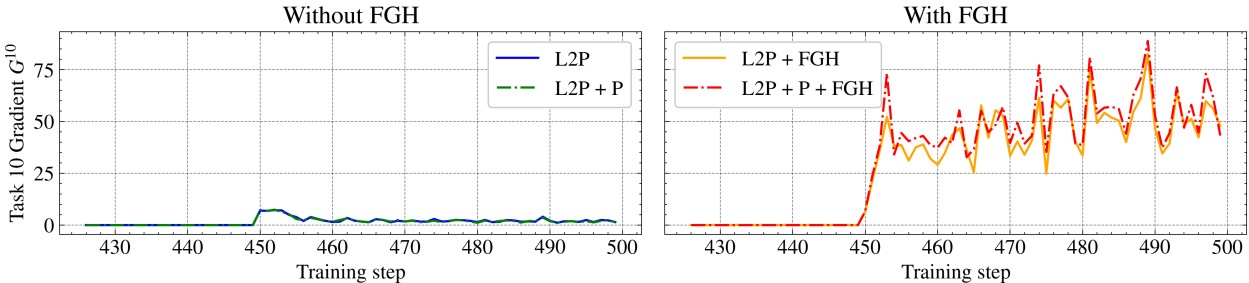

Figure A26: Values of $G^{10}$ for L2P on CIFAR100, 10 tasks, with and without prototypes and FGH. When including FGH, we display the resulting gradient *after* multiplying by the coefficients. Task changes every 50 steps. Only 250 steps are displayed for readability.

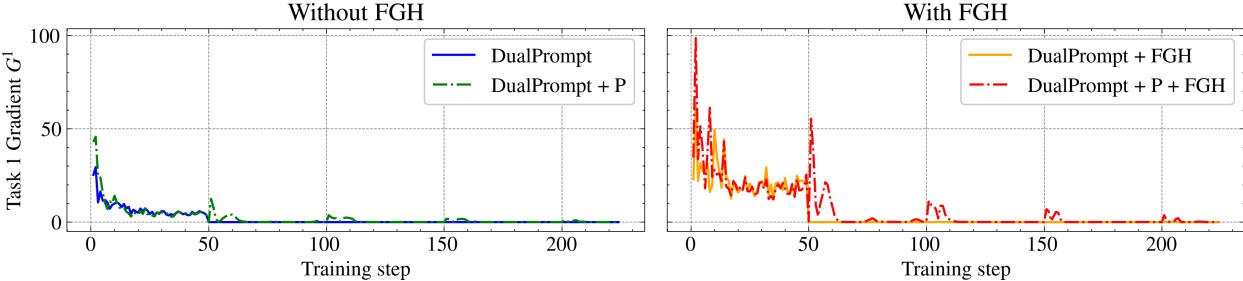

Figure A27: Values of $G^1$ for DualPrompt on CIFAR100, 10 tasks, with and without prototypes and FGH. When including FGH, we display the resulting gradient *after* multiplying by the coefficients. Task changes every 50 steps. Only 250 steps are displayed for readability.

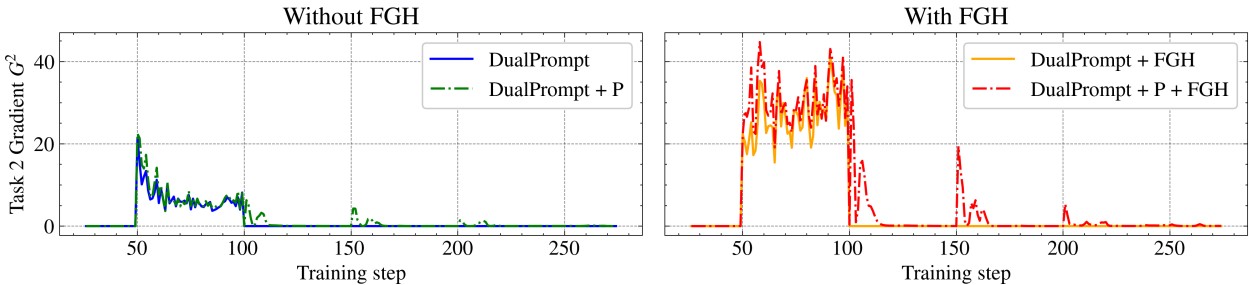

Figure A28: Values of $G^2$ for DualPrompt on CIFAR100, 10 tasks, with and without prototypes and FGH. When including FGH, we display the resulting gradient *after* multiplying by the coefficients. Task changes every 50 steps. Only 250 steps are displayed for readability.

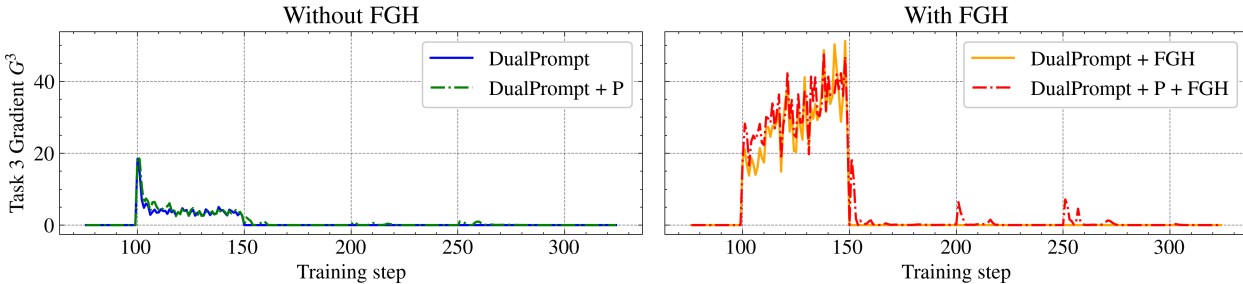

Figure A29: Values of $G^3$ for DualPrompt on CIFAR100, 10 tasks, with and without prototypes and FGH. When including FGH, we display the resulting gradient *after* multiplying by the coefficients. Task changes every 50 steps. Only 250 steps are displayed for readability.

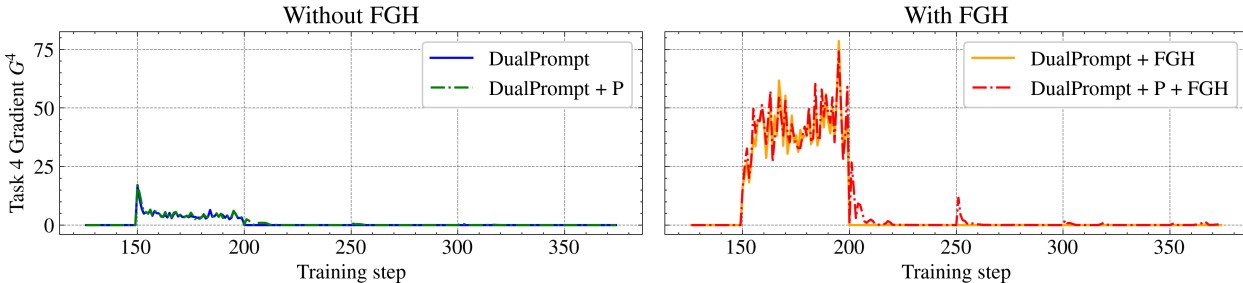

Figure A30: Values of $G^4$ for DualPrompt on CIFAR100, 10 tasks, with and without prototypes and FGH. When including FGH, we display the resulting gradient *after* multiplying by the coefficients. Task changes every 50 steps. Only 250 steps are displayed for readability.

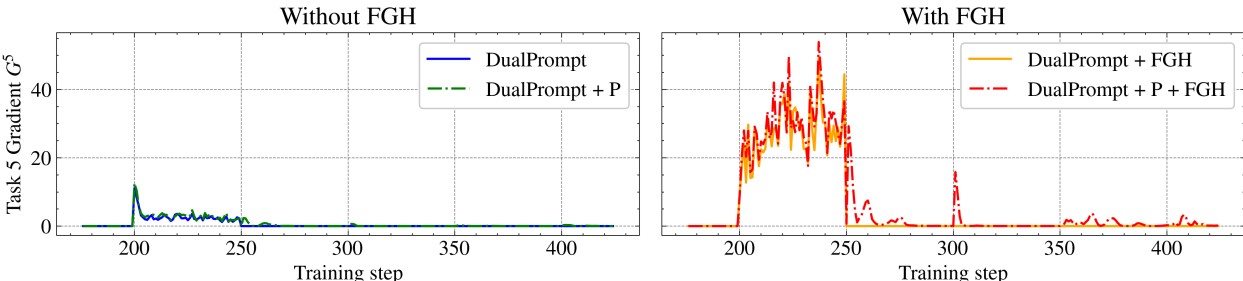

Figure A31: Values of $G^5$ for DualPrompt on CIFAR100, 10 tasks, with and without prototypes and FGH. When including FGH, we display the resulting gradient *after* multiplying by the coefficients. Task changes every 50 steps. Only 250 steps are displayed for readability.

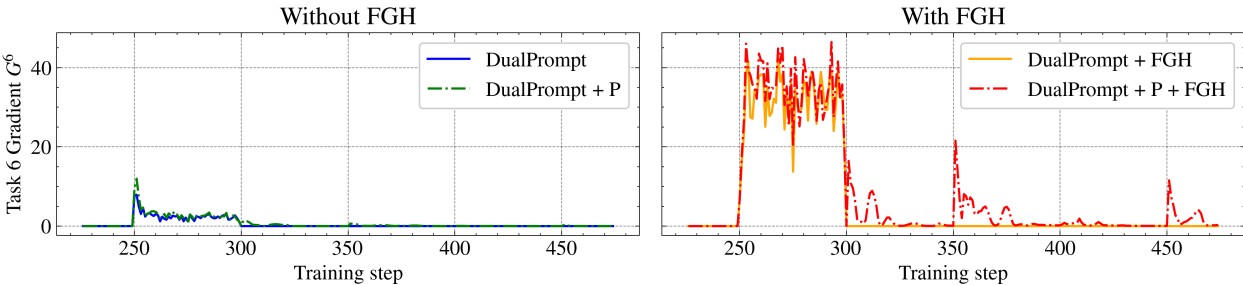

Figure A32: Values of $G^6$ for DualPrompt on CIFAR100, 10 tasks, with and without prototypes and FGH. When including FGH, we display the resulting gradient *after* multiplying by the coefficients. Task changes every 50 steps. Only 250 steps are displayed for readability.

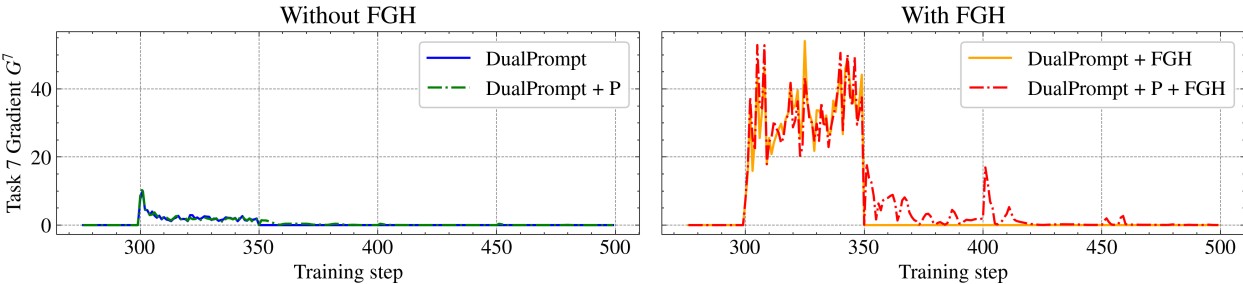

Figure A33: Values of $G^7$ for DualPrompt on CIFAR100, 10 tasks, with and without prototypes and FGH. When including FGH, we display the resulting gradient *after* multiplying by the coefficients. Task changes every 50 steps. Only 250 steps are displayed for readability.

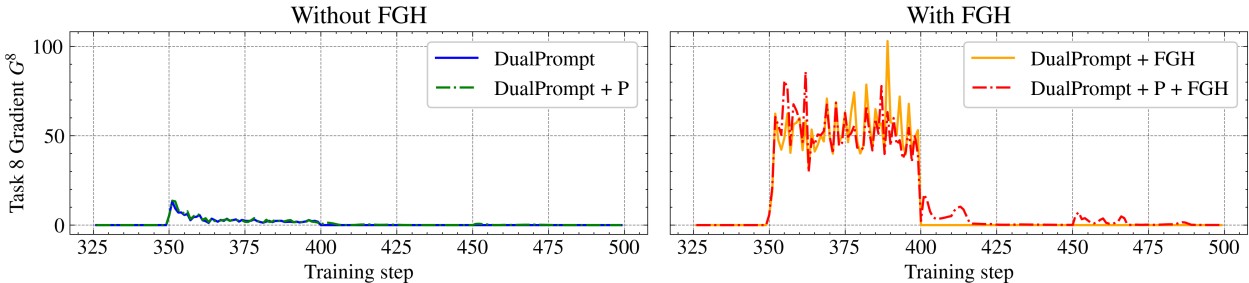

Figure A34: Values of $G^8$ for DualPrompt on CIFAR100, 10 tasks, with and without prototypes and FGH. When including FGH, we display the resulting gradient *after* multiplying by the coefficients. Task changes every 50 steps. Only 250 steps are displayed for readability.

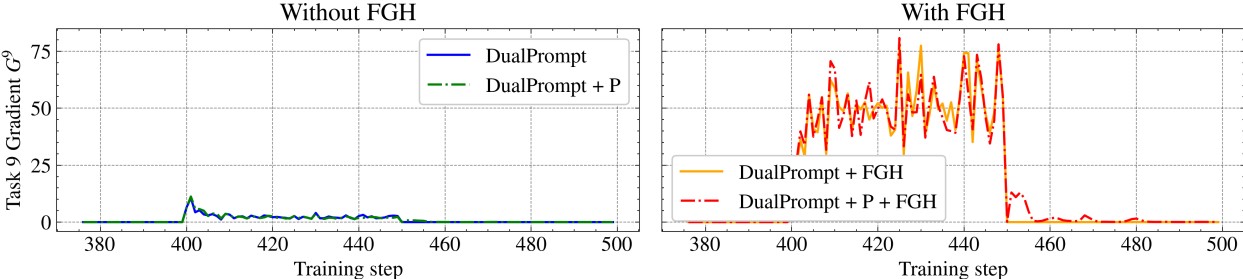

Figure A35: Values of $G^9$ for DualPrompt on CIFAR100, 10 tasks, with and without prototypes and FGH. When including FGH, we display the resulting gradient *after* multiplying by the coefficients. Task changes every 50 steps. Only 250 steps are displayed for readability.

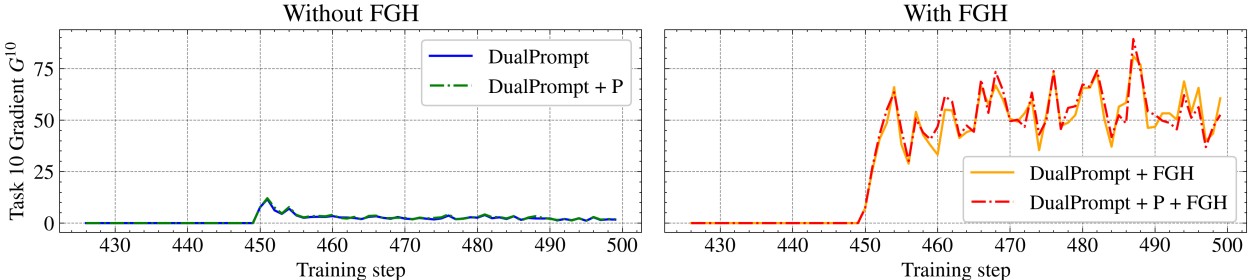

Figure A36: Values of $G^{10}$ for DualPrompt on CIFAR100, 10 tasks, with and without prototypes and FGH. When including FGH, we display the resulting gradient *after* multiplying by the coefficients. Task changes every 50 steps. Only 250 steps are displayed for readability.

Table B10: Final performances $\mathcal{A}_T$ (%) of all considered baselines, in the *clear* setting. + *ours* refers to combining the baselines with prototypes and FGH. Best HP refers to the best set of LR and $\gamma$ found on VTAB. In some cases, the best HP is the same as one of the default HP values.

| Dataset | CIFAR100 | | | CUB | | | Imagenet-R | | |
|---|---|---|---|---|---|---|---|---|---|
| Learning Rate | $5 \times 10^{-5}$ | $5 \times 10^{-3}$ | Best HP | $5 \times 10^{-5}$ | $5 \times 10^{-3}$ | Best HP | $5 \times 10^{-5}$ | $5 \times 10^{-3}$ | Best HP |
| Fine-tuning | $9.12_{\pm2.82}$ | $1.0_{\pm0.0}$ | $1.0_{\pm0.0}$ | $0.48_{\pm0.12}$ | $0.53_{\pm0.09}$ | $0.43_{\pm0.15}$ | $0.76_{\pm0.22}$ | $0.74_{\pm0.31}$ | $0.63_{\pm0.36}$ |
| Linear probe | $17.56_{\pm1.51}$ | $14.51_{\pm1.27}$ | $14.51_{\pm1.27}$ | $1.41_{\pm0.37}$ | $34.96_{\pm1.46}$ | $34.96_{\pm1.46}$ | $2.96_{\pm0.64}$ | $27.33_{\pm1.35}$ | $27.33_{\pm1.35}$ |
| ER | $\mathbf{69.7_{\pm0.98}}$ | $1.03_{\pm0.11}$ | $69.7_{\pm0.98}$ | $\mathbf{35.91_{\pm3.98}}$ | $0.37_{\pm0.17}$ | $35.91_{\pm3.98}$ | $\mathbf{39.99_{\pm2.15}}$ | $0.74_{\pm0.21}$ | $39.99_{\pm2.15}$ |
| ER + Linear probe | $41.78_{\pm1.37}$ | $72.25_{\pm1.35}$ | $71.08_{\pm0.89}$ | $3.32_{\pm0.49}$ | $48.07_{\pm2.0}$ | $47.65_{\pm1.99}$ | $8.56_{\pm0.68}$ | $38.33_{\pm2.6}$ | $37.56_{\pm2.79}$ |
| MVP | $19.76_{\pm2.35}$ | $17.42_{\pm4.28}$ | $20.18_{\pm4.27}$ | $1.54_{\pm0.52}$ | $34.18_{\pm1.94}$ | $16.3_{\pm1.83}$ | $2.94_{\pm0.39}$ | $23.48_{\pm10.91}$ | $21.8_{\pm2.68}$ |
| oLoRA | $40.07_{\pm9.51}$ | $3.9_{\pm2.28}$ | $5.0_{\pm1.45}$ | $4.37_{\pm0.78}$ | $31.69_{\pm2.72}$ | $26.59_{\pm5.33}$ | $5.74_{\pm1.0}$ | $18.96_{\pm3.16}$ | $37.09_{\pm11.54}$ |
| CODA | $26.02_{\pm1.99}$ | $59.34_{\pm6.28}$ | $55.56_{\pm4.43}$ | $1.05_{\pm0.28}$ | $48.09_{\pm2.82}$ | $35.19_{\pm4.61}$ | $2.2_{\pm0.49}$ | $47.38_{\pm3.45}$ | $40.13_{\pm4.45}$ |
| $\hookrightarrow$ + ours | $68.12_{\pm1.68}$ | $72.26_{\pm2.4}$ | $66.71_{\pm3.53}$ | $9.52_{\pm1.38}$ | $58.46_{\pm5.04}$ | $42.88_{\pm4.22}$ | $20.07_{\pm0.96}$ | $51.8_{\pm2.35}$ | $44.99_{\pm4.45}$ |
| L2P | $21.17_{\pm3.45}$ | $49.26_{\pm3.88}$ | $49.26_{\pm3.88}$ | $0.65_{\pm0.36}$ | $33.2_{\pm1.96}$ | $33.2_{\pm1.96}$ | $1.91_{\pm0.44}$ | $36.22_{\pm3.29}$ | $36.22_{\pm3.29}$ |
| $\hookrightarrow$ + ours | $56.98_{\pm2.11}$ | $73.59_{\pm1.83}$ | $73.59_{\pm1.83}$ | $5.15_{\pm2.52}$ | $66.74_{\pm1.44}$ | $66.74_{\pm1.44}$ | $18.0_{\pm1.17}$ | $60.23_{\pm0.87}$ | $60.23_{\pm0.87}$ |
| DualPrompt | $23.28_{\pm1.69}$ | $50.68_{\pm3.38}$ | $53.77_{\pm3.48}$ | $0.9_{\pm0.24}$ | $52.03_{\pm1.96}$ | $48.51_{\pm2.73}$ | $2.32_{\pm0.23}$ | $47.89_{\pm1.8}$ | $46.34_{\pm1.95}$ |
| $\hookrightarrow$ + ours | $58.96_{\pm1.43}$ | $62.63_{\pm3.33}$ | $66.09_{\pm2.41}$ | $9.87_{\pm1.86}$ | $\mathbf{72.25_{\pm0.88}}$ | $65.47_{\pm1.8}$ | $20.85_{\pm1.05}$ | $55.87_{\pm1.41}$ | $54.15_{\pm1.64}$ |
| ConvPrompt | $34.48_{\pm2.51}$ | $59.87_{\pm6.26}$ | $59.87_{\pm6.26}$ | $0.65_{\pm0.14}$ | $54.96_{\pm2.82}$ | $54.96_{\pm2.82}$ | $1.71_{\pm0.08}$ | $46.92_{\pm2.9}$ | $46.92_{\pm2.9}$ |
| $\hookrightarrow$ + ours | $66.21_{\pm1.54}$ | $\mathbf{82.49_{\pm0.62}}$ | $\mathbf{82.49_{\pm0.62}}$ | $9.98_{\pm0.96}$ | $69.78_{\pm0.03}$ | $\mathbf{69.78_{\pm0.03}}$ | $5.78_{\pm2.71}$ | $\mathbf{63.68_{\pm1.2}}$ | $\mathbf{63.68_{\pm1.2}}$ |

Table B11: Final performances $\mathcal{A}_T$ (%) of all considered baselines, in the *Si-Blurry* setting. + *ours* refers to combining the baselines with prototypes and FGH. Best HP refer to the best set of LR and $\gamma$ found on VTAB. In some cases, the best HP is the same as one of the default HP values.

| Dataset | CIFAR100 | | | CUB | | | Imagenet-R | | |
|---|---|---|---|---|---|---|---|---|---|
| Learning Rate | $5\times10^{-5}$ | $5\times10^{-3}$ | Best HP | $5\times10^{-5}$ | $5\times10^{-3}$ | Best HP | $5\times10^{-5}$ | $5\times10^{-3}$ | Best HP |
| Fine-tuning | 9.85±6.88 | 1.03±0.1 | 1.0±0.0 | 0.46±0.14 | 0.58±0.13 | 0.49±0.06 | 0.84±0.37 | 0.64±0.34 | 0.55±0.23 |
| Linear probe | 28.61±2.74 | 18.48±4.2 | 18.48±4.2 | 2.99±0.68 | 32.82±2.42 | 32.82±2.42 | 5.77±0.77 | 23.99±3.58 | 23.99±3.58 |
| ER | **71.42±2.96** | 0.89±1.03 | 71.42±2.96 | **50.37±2.36** | 0.23±0.35 | 50.37±2.36 | **49.24±7.13** | 1.03±0.63 | 49.24±7.13 |
| ER + Linear probe | 47.39±4.38 | 71.72±2.43 | 70.72±2.5 | 6.96±1.87 | 51.09±4.49 | 51.85±3.87 | 13.53±1.57 | 45.87±5.73 | 45.79±6.3 |
| MVP | 23.31±3.02 | 20.2±7.9 | 25.44±5.23 | 2.12±0.57 | 24.57±3.46 | 17.23±4.38 | 4.72±0.65 | 24.96±4.06 | 25.14±4.6 |
| oLoRA | 49.35±1.05 | 4.82±1.74 | 4.94±1.89 | 7.24±1.75 | 34.35±9.53 | 30.52±9.55 | 12.1±2.38 | 14.74±7.94 | 35.38±1.72 |
| CODA | 32.14±3.11 | 60.88±8.31 | 54.18±6.45 | 1.46±0.34 | 43.19±5.91 | 25.37±5.35 | 3.48±0.59 | 37.42±5.75 | 31.46±3.66 |
| ↪ + ours | 70.53±1.9 | 76.11±2.94 | 70.21±4.21 | 15.3±1.49 | 64.97±3.97 | 42.92±9.1 | 23.35±1.42 | 51.86±3.52 | 40.15±10.91 |
| L2P | 25.14±4.21 | 52.54±6.04 | 52.54±6.04 | 0.95±0.35 | 25.14±5.06 | 25.14±5.06 | 2.71±0.47 | 28.38±7.13 | 28.38±7.13 |
| ↪ + ours | 59.49±1.92 | 75.39±2.31 | 75.39±2.31 | 10.37±2.86 | 66.55±2.84 | 66.55±2.84 | 20.35±1.44 | 59.26±1.91 | 59.26±1.91 |
| DualPrompt | 29.15±2.87 | 52.4±10.47 | 45.96±8.47 | 1.34±0.47 | 46.61±3.34 | 33.99±5.95 | 3.72±0.96 | 43.83±1.94 | 35.19±3.1 |
| ↪ + ours | 61.71±2.11 | 67.74±3.19 | 68.56±3.87 | 16.87±1.35 | 73.13±0.73 | 63.73±7.43 | 24.19±1.64 | 55.16±1.58 | 52.02±2.08 |
| ConvPrompt | 38.13±5.9 | 63.8±1.98 | 63.8±1.98 | 1.36±0.32 | 42.78±2.83 | 42.78±2.83 | 1.88±0.74 | 44.13±3.43 | 44.13±3.43 |
| ↪ + ours | 62.52±2.96 | **81.71±1.64** | **81.71±1.64** | 15.06±0.96 | **73.99±3.9** | **73.99±3.9** | 12.61±0.83 | **61.89±4.2** | **61.89±4.2** |

Table B12: Accuracy on the **first task** at the end of training, in the *clear* setting. + *ours* refers to combining the baselines with prototypes and FGH. Best HP refers to the best set of LR and $\gamma$ found on VTAB. In some cases, the best HP is the same as one of the default HP values.

| Dataset | CIFAR100 | | | CUB | | | Imagenet-R | | |
|---|---|---|---|---|---|---|---|---|---|
| Learning Rate | $5\times10^{-5}$ | $5\times10^{-3}$ | Best HP | $5\times10^{-5}$ | $5\times10^{-3}$ | Best HP | $5\times10^{-5}$ | $5\times10^{-3}$ | Best HP |
| Fine-tuning | 0.0±0.0 | 0.0±0.0 | 0.0±0.0 | 0.0±0.0 | 0.0±0.0 | 0.0±0.0 | 0.0±0.0 | 0.0±0.0 | 0.0±0.0 |
| Linear probe | 0.41±0.29 | 0.0±0.0 | 0.0±0.0 | 0.08±0.27 | 1.38±3.23 | 1.38±3.23 | 0.1±0.22 | 0.0±0.0 | 0.0±0.0 |
| ER | 62.43±2.97 | 0.0±0.0 | 62.43±2.97 | **64.67±13.04** | 0.0±0.0 | **64.67±13.04** | 38.18±7.77 | 0.0±0.0 | 38.18±7.77 |
| ER + Linear probe | 38.96±5.37 | 64.99±6.09 | 61.42±7.66 | 13.17±5.13 | 45.09±11.47 | 46.07±10.57 | 18.92±7.04 | 26.8±7.04 | 24.2±8.42 |
| MVP | 0.01±0.03 | 0.05±0.16 | 0.01±0.03 | 0.42±0.6 | 7.53±5.74 | 0.0±0.0 | 0.03±0.08 | 1.29±1.32 | 1.14±1.06 |
| oLoRA | 40.07±9.51 | 3.9±2.28 | 5.0±1.45 | 4.37±0.78 | 31.69±2.72 | 26.59±5.33 | 5.74±1.0 | 18.96±3.16 | 37.09±11.54 |
| CODA | 5.22±3.27 | 2.98±2.79 | 3.85±2.02 | 0.77±1.27 | 33.14±14.08 | 34.34±12.6 | 0.85±1.1 | 16.97±7.85 | 13.93±5.53 |
| ↪ + P | 49.35±5.74 | 15.74±6.12 | 24.29±8.71 | 2.87±2.62 | 36.17±10.26 | 18.73±9.61 | 7.6±4.47 | 35.64±11.15 | 28.21±7.93 |
| ↪ + FGH | 6.6±3.9 | 2.48±2.39 | – | 0.68±1.03 | 32.46±11.4 | – | – | – | – |
| ↪ + ours | 64.86±4.68 | 28.92±7.37 | 33.08±11.74 | 13.02±6.27 | 44.95±18.49 | 22.29±10.48 | 31.44±10.41 | 47.5±10.46 | 28.58±8.82 |
| L2P | 4.51±5.77 | 2.93±2.61 | 2.93±2.61 | 1.24±1.71 | 26.1±10.59 | 26.1±10.59 | 1.26±1.39 | 7.85±6.01 | 7.85±6.01 |
| ↪ + P | 44.5±10.62 | 36.37±7.39 | 36.37±7.39 | 2.99±4.06 | 38.8±11.83 | 38.8±11.83 | 7.08±2.79 | 38.28±4.39 | 38.28±4.39 |
| ↪ + FGH | 6.46±6.62 | 2.98±1.95 | 2.98±1.95 | 1.34±1.83 | 29.58±13.42 | 29.58±13.42 | – | – | – |
| ↪ + ours | **65.77±7.03** | 42.58±8.65 | 42.58±8.65 | 15.09±9.39 | 39.96±10.48 | 39.96±10.48 | 27.16±9.5 | 48.4±4.65 | 48.4±4.65 |
| DualPrompt | 4.75±3.64 | 4.99±2.78 | 3.13±2.85 | 0.85±1.3 | 43.81±13.29 | 48.9±16.35 | 1.17±0.98 | 18.52±7.71 | 16.38±6.74 |
| ↪ + P | 45.65±7.79 | 18.56±6.36 | 18.24±6.02 | 4.19±4.34 | 68.22±8.01 | 59.78±15.56 | 10.48±6.13 | 46.8±5.62 | 37.83±5.94 |
| ↪ + FGH | 6.51±4.1 | 0.69±1.24 | – | 0.77±1.08 | 47.78±12.73 | – | – | – | – |
| ↪ + ours | 64.83±5.93 | 28.03±8.27 | 25.14±4.54 | 19.8±8.87 | **76.92±6.5** | 61.09±12.27 | 32.31±7.76 | **53.72±5.67** | 43.84±4.97 |
| ConvPrompt | 5.1±4.78 | 11.37±7.29 | 11.37±7.29 | 0.32±0.56 | 54.18±12.29 | 54.18±12.29 | 0.0±0.0 | 18.29±6.08 | 18.29±6.08 |
| ↪ + P | 38.3±10.64 | 46.57±13.77 | 46.57±13.77 | 3.19±3.16 | 53.99±13.23 | 53.99±13.23 | 0.43±0.74 | 39.76±3.45 | 39.76±3.45 |
| ↪ + ours | 38.37±6.1 | **66.67±2.22** | **66.67±2.22** | 18.26±7.14 | 61.51±1.87 | 61.51±1.87 | 0.43±0.74 | 48.04±8.67 | **48.04±8.67** |

Table B13: Accuracy on the **second task** at the end of training, in the *clear* setting. + *ours* refers to combining the baselines with prototypes and FGH. Best HP refers to the best set of LR and $\gamma$ found on VTAB. In some cases, the best HP is the same as one of the default HP values.

| Dataset | CIFAR100 | | | CUB | | | Imagenet-R | | |
|---|---|---|---|---|---|---|---|---|---|
| Learning Rate | $5 \times 10^{-5}$ | $5 \times 10^{-3}$ | Best HP | $5 \times 10^{-5}$ | $5 \times 10^{-3}$ | Best HP | $5 \times 10^{-5}$ | $5 \times 10^{-3}$ | Best HP |
| Fine-tuning | 0.5±1.35 | 0.0±0.0 | 0.0±0.0 | 0.0±0.0 | 0.0±0.0 | 0.0±0.0 | 0.0±0.0 | 0.0±0.0 | 0.0±0.0 |
| Linear probe | 1.76±1.56 | 0.06±0.13 | 0.06±0.13 | 0.08±0.27 | 4.79±4.02 | 4.79±4.02 | 0.32±0.41 | 0.42±0.81 | 0.42±0.81 |
| ER | 62.07±5.34 | 0.0±0.0 | 62.07±5.34 | **57.2±13.2** | 0.0±0.0 | 57.2±13.2 | **40.57±5.46** | 0.0±0.0 | 40.57±5.46 |
| ER + Linear probe | 44.28±5.71 | 67.03±6.96 | 67.22±6.88 | 7.11±3.78 | 42.13±14.69 | 38.89±17.54 | 16.3±4.03 | 27.22±6.59 | 27.28±5.21 |
| MVP | 1.33±1.5 | 0.11±0.23 | – | 0.0±0.0 | 7.35±4.7 | – | 0.37±0.52 | 1.8±1.99 | – |
| oLoRA | 14.9±6.68 | 0.0±0.0 | 0.0±0.0 | 0.63±1.09 | 8.78±7.32 | 2.97±3.76 | 0.84±1.46 | 0.25±0.43 | 11.79±10.63 |
| CODA | 11.4±5.2 | 29.81±7.19 | – | 1.56±2.52 | 28.65±11.51 | – | 1.02±1.2 | 21.98±4.95 | – |
| ↪ + P | 59.05±6.64 | 42.75±7.63 | – | 5.15±4.14 | 35.57±11.26 | – | 8.88±2.94 | 35.44±6.67 | – |
| ↪ + FGH | 16.47±6.98 | 20.89±7.97 | – | 1.56±2.52 | 26.36±12.74 | – | – | – | – |
| ↪ + ours | 68.73±7.1 | 54.2±5.28 | – | 19.64±5.68 | 45.65±12.77 | – | 36.15±6.85 | 46.16±9.51 | – |
| L2P | 6.01±8.11 | 16.23±5.69 | 16.23±5.69 | 1.1±2.13 | 20.67±13.88 | 20.67±13.88 | 0.8±1.11 | 10.34±3.75 | 10.34±3.75 |
| ↪ + P | 53.14±8.38 | 56.89±7.6 | 56.89±7.6 | 2.44±3.86 | 53.0±12.79 | 53.0±12.79 | 5.44±2.29 | 45.3±3.54 | 45.3±3.54 |
| ↪ + FGH | 9.47±9.71 | 10.51±4.1 | 10.51±4.1 | 1.1±2.13 | 19.88±14.5 | 19.88±14.5 | – | – | – |
| ↪ + ours | **70.37±5.79** | 58.16±7.25 | 58.16±7.25 | 10.95±10.12 | 55.79±12.47 | 55.79±12.47 | 32.08±7.12 | 52.57±5.92 | 52.57±5.92 |
| DualPrompt | 13.28±6.22 | 18.57±9.45 | – | 1.32±1.75 | 40.33±10.2 | – | 1.09±1.42 | 25.61±5.38 | – |
| ↪ + P | 56.55±8.69 | 28.6±9.94 | – | 4.48±5.39 | 70.17±11.28 | – | 10.38±4.43 | 50.53±4.11 | – |
| ↪ + FGH | 18.62±7.88 | 4.92±3.61 | – | 1.32±1.75 | 42.12±10.12 | – | – | – | – |
| ↪ + ours | 70.0±6.32 | 35.28±8.88 | – | 18.71±10.63 | **76.09±8.61** | – | 37.32±7.09 | **55.13±6.03** | – |
| ConvPrompt | 19.83±14.51 | 40.1±12.7 | 40.1±12.7 | 0.27±0.46 | 46.95±5.32 | 46.95±5.32 | 0.21±0.37 | 29.64±4.04 | 29.64±4.04 |
| ↪ + P | 35.93±10.27 | 65.83±11.0 | 65.83±11.0 | 3.84±2.51 | 51.34±27.6 | 51.34±27.6 | 1.5±2.59 | 44.97±3.34 | 44.97±3.34 |
| ↪ + ours | 57.3±1.04 | **72.63±2.73** | **72.63±2.73** | 22.31±11.37 | 68.48±4.98 | **68.48±4.98** | 2.35±4.07 | 52.59±3.73 | **52.59±3.73** |

Table B14: Accuracy on the **third task** at the end of training, in the *clear* setting. + *ours* refers to combining the baselines with prototypes and FGH. Best HP refer to the best set of LR and $\gamma$ found on VTAB. In some cases, the best HP is the same as one of the default HP values.

| Dataset | CIFAR100 | | | CUB | | | Imagenet-R | | |
|---|---|---|---|---|---|---|---|---|---|
| Learning Rate | $5 \times 10^{-5}$ | $5 \times 10^{-3}$ | Best HP | $5 \times 10^{-5}$ | $5 \times 10^{-3}$ | Best HP | $5 \times 10^{-5}$ | $5 \times 10^{-3}$ | Best HP |
| Fine-tuning | 0.2±0.42 | 0.0±0.0 | 0.0±0.0 | 0.0±0.0 | 0.0±0.0 | 0.0±0.0 | 0.0±0.0 | 0.0±0.0 | 0.0±0.0 |
| Linear probe | 3.58±2.43 | 0.06±0.19 | 0.06±0.19 | 0.26±0.43 | 7.39±7.84 | 7.39±7.84 | 0.42±0.42 | 1.28±1.59 | 1.28±1.59 |
| ER | 62.01±7.57 | 0.0±0.0 | 62.01±7.57 | **45.23±12.07** | 0.0±0.0 | 45.23±12.07 | **41.22±4.39** | 0.0±0.0 | 41.22±4.39 |
| ER + Linear probe | 44.07±6.01 | 66.83±8.0 | 64.91±8.62 | 5.93±4.84 | 39.05±11.92 | 39.21±10.9 | 14.68±3.14 | 30.15±6.34 | 30.37±6.9 |
| MVP | 2.35±2.41 | 0.73±1.06 | – | 0.08±0.27 | 10.94±9.81 | – | 0.89±1.97 | 3.86±3.97 | – |
| oLoRA | 12.3±6.68 | 0.0±0.0 | 0.0±0.0 | 29.05±26.66 | 8.67±14.33 | 33.17±39.11 | 51.74±9.2 | 0.63±0.84 | 15.85±11.14 |
| CODA | 14.32±10.11 | 46.0±10.51 | – | 0.84±1.34 | 40.51±16.78 | – | 1.93±2.17 | 30.22±11.81 | – |
| ↪ + P | 61.79±5.14 | 62.85±10.83 | – | 4.3±2.77 | 46.09±12.72 | – | 11.6±3.67 | 39.91±10.5 | – |
| ↪ + FGH | 20.74±11.84 | 34.76±6.96 | – | 0.84±1.34 | 39.07±14.24 | – | – | – | – |
| ↪ + ours | 71.38±5.67 | 68.36±7.09 | – | 21.98±7.13 | 51.6±13.32 | – | 39.22±7.7 | 48.57±4.53 | – |
| L2P | 12.44±10.05 | 26.18±10.63 | 26.18±10.63 | 0.65±1.79 | 20.46±12.57 | 20.46±12.57 | 1.21±2.71 | 15.3±7.01 | 15.3±7.01 |
| ↪ + P | 52.87±5.92 | 65.37±8.19 | 65.37±8.19 | 1.14±3.05 | 64.42±10.22 | 64.42±10.22 | 8.59±4.06 | 50.06±8.21 | 50.06±8.21 |
| ↪ + FGH | 16.32±12.08 | 17.25±9.09 | 17.25±9.09 | 0.65±1.79 | 21.08±12.96 | 21.08±12.96 | – | – | – |
| ↪ + ours | **67.65±6.55** | 64.57±8.51 | 64.57±8.51 | 11.03±8.88 | 63.87±10.59 | 63.87±10.59 | 34.31±10.38 | 55.19±6.82 | **55.19±6.82** |
| DualPrompt | 12.34±5.41 | 30.04±9.72 | – | 0.42±0.71 | 49.04±11.44 | – | 1.68±0.94 | 32.92±9.09 | – |
| ↪ + P | 51.77±6.41 | 34.82±7.71 | – | 4.19±2.85 | 73.38±8.12 | – | 12.05±4.16 | 52.07±5.48 | – |
| ↪ + FGH | 18.18±5.57 | 8.89±3.58 | – | 0.42±0.71 | 50.02±11.28 | – | – | – | – |
| ↪ + ours | 66.3±4.77 | 37.81±8.51 | – | 24.14±8.4 | **79.8±5.17** | – | 39.5±7.43 | **57.79±4.49** | – |
| ConvPrompt | 13.6±0.9 | 48.57±13.06 | 48.57±13.06 | 0.84±0.84 | 42.89±17.61 | 42.89±17.61 | 0.85±1.47 | 21.72±9.31 | 21.72±9.31 |
| ↪ + P | 30.07±11.51 | 71.6±6.85 | 71.6±6.85 | 5.28±5.68 | 51.32±6.29 | 51.32±6.29 | 1.86±3.22 | 41.33±2.76 | 41.33±2.76 |
| ↪ + ours | 53.37±5.4 | **79.3±2.95** | **79.3±2.95** | 21.32±9.5 | 69.56±15.89 | **69.56±15.89** | 2.67±4.28 | 52.44±7.75 | 52.44±7.75 |

Table B15: Accuracy on the **fourth task** at the end of training, in the *clear* setting. + *ours* refers to combining the baselines with prototypes and FGH. Best HP refer to the best set of LR and $\gamma$ found on VTAB. In some cases, the best HP is the same as one of the default HP values.

| Dataset | CIFAR100 | | | CUB | | | Imagenet-R | | |
|---|---|---|---|---|---|---|---|---|---|
| Learning Rate | $5 \times 10^{-5}$ | $5 \times 10^{-3}$ | Best HP | $5 \times 10^{-5}$ | $5 \times 10^{-3}$ | Best HP | $5 \times 10^{-5}$ | $5 \times 10^{-3}$ | Best HP |
| Fine-tuning | $0.54_{\pm0.99}$ | $0.0_{\pm0.0}$ | $0.0_{\pm0.0}$ | $0.0_{\pm0.0}$ | $0.0_{\pm0.0}$ | $0.0_{\pm0.0}$ | $0.0_{\pm0.0}$ | $0.0_{\pm0.0}$ | $0.0_{\pm0.0}$ |
| Linear probe | $4.76_{\pm1.79}$ | $0.49_{\pm1.45}$ | $0.49_{\pm1.45}$ | $0.62_{\pm0.92}$ | $5.07_{\pm5.34}$ | $5.07_{\pm5.34}$ | $0.62_{\pm0.97}$ | $2.33_{\pm2.85}$ | $2.33_{\pm2.85}$ |
| ER | $66.35_{\pm4.69}$ | $0.0_{\pm0.0}$ | $66.35_{\pm4.69}$ | $\mathbf{38.98_{\pm10.54}}$ | $0.0_{\pm0.0}$ | $38.98_{\pm10.54}$ | $34.96_{\pm6.97}$ | $0.0_{\pm0.0}$ | $34.96_{\pm6.97}$ |
| ER + Linear probe | $42.95_{\pm6.26}$ | $68.43_{\pm4.89}$ | $69.05_{\pm5.27}$ | $6.57_{\pm3.61}$ | $38.72_{\pm9.06}$ | $37.56_{\pm11.7}$ | $12.47_{\pm6.5}$ | $27.76_{\pm8.92}$ | $28.24_{\pm7.64}$ |
| MVP | $6.75_{\pm4.68}$ | $5.76_{\pm5.72}$ | – | $1.01_{\pm1.8}$ | $7.89_{\pm6.65}$ | – | $0.42_{\pm0.54}$ | $4.25_{\pm4.66}$ | – |
| oLoRA | $25.57_{\pm26.01}$ | $0.0_{\pm0.0}$ | $0.0_{\pm0.0}$ | $1.17_{\pm1.02}$ | $10.24_{\pm9.43}$ | $8.91_{\pm8.71}$ | $0.33_{\pm0.31}$ | $0.0_{\pm0.0}$ | $15.45_{\pm22.49}$ |
| CODA | $20.45_{\pm8.6}$ | $58.94_{\pm13.78}$ | – | $1.44_{\pm3.05}$ | $32.5_{\pm6.88}$ | – | $1.6_{\pm0.72}$ | $29.01_{\pm8.65}$ | – |
| $\hookrightarrow$ + P | $59.24_{\pm4.04}$ | $75.75_{\pm5.02}$ | – | $4.28_{\pm4.08}$ | $44.25_{\pm11.51}$ | – | $10.06_{\pm4.67}$ | $34.45_{\pm9.94}$ | – |
| $\hookrightarrow$ + FGH | $28.89_{\pm10.91}$ | $45.19_{\pm8.85}$ | – | $1.44_{\pm3.05}$ | $33.64_{\pm10.82}$ | – | – | – | – |
| $\hookrightarrow$ + ours | $69.33_{\pm5.02}$ | $76.04_{\pm3.36}$ | – | $22.63_{\pm7.87}$ | $50.18_{\pm11.71}$ | – | $34.7_{\pm7.41}$ | $44.95_{\pm10.76}$ | – |
| L2P | $16.15_{\pm9.74}$ | $41.97_{\pm10.57}$ | $41.97_{\pm10.57}$ | $1.24_{\pm2.54}$ | $21.06_{\pm12.41}$ | $21.06_{\pm12.41}$ | $1.13_{\pm1.92}$ | $17.15_{\pm7.33}$ | $17.15_{\pm7.33}$ |
| $\hookrightarrow$ + P | $55.65_{\pm6.53}$ | $73.84_{\pm6.57}$ | $73.84_{\pm6.57}$ | $2.65_{\pm4.07}$ | $55.37_{\pm14.24}$ | $55.37_{\pm14.24}$ | $9.54_{\pm6.78}$ | $49.76_{\pm8.76}$ | $49.76_{\pm8.76}$ |
| $\hookrightarrow$ + FGH | $23.01_{\pm10.64}$ | $33.76_{\pm8.39}$ | $33.76_{\pm8.39}$ | $1.33_{\pm2.5}$ | $23.23_{\pm15.51}$ | $23.23_{\pm15.51}$ | – | – | – |
| $\hookrightarrow$ + ours | $67.59_{\pm5.03}$ | $73.88_{\pm7.47}$ | $73.88_{\pm7.47}$ | $12.06_{\pm5.9}$ | $53.07_{\pm13.2}$ | $53.07_{\pm13.2}$ | $\mathbf{37.38_{\pm11.45}}$ | $54.17_{\pm7.52}$ | $54.17_{\pm7.52}$ |
| DualPrompt | $14.79_{\pm6.6}$ | $36.67_{\pm7.25}$ | – | $0.47_{\pm0.81}$ | $36.89_{\pm10.78}$ | – | $1.41_{\pm1.13}$ | $31.87_{\pm8.22}$ | – |
| $\hookrightarrow$ + P | $51.78_{\pm10.31}$ | $44.45_{\pm6.67}$ | – | $2.84_{\pm2.99}$ | $65.33_{\pm8.16}$ | – | $9.72_{\pm4.64}$ | $47.43_{\pm8.23}$ | – |
| $\hookrightarrow$ + FGH | $23.06_{\pm8.64}$ | $17.68_{\pm7.97}$ | – | $0.47_{\pm0.81}$ | $38.37_{\pm11.81}$ | – | – | – | – |
| $\hookrightarrow$ + ours | $64.26_{\pm8.1}$ | $49.89_{\pm7.68}$ | – | $17.88_{\pm9.14}$ | $69.35_{\pm7.3}$ | – | $35.98_{\pm10.89}$ | $52.34_{\pm6.25}$ | – |
| ConvPrompt | $19.4_{\pm4.47}$ | $56.37_{\pm11.38}$ | $56.37_{\pm11.38}$ | $1.25_{\pm2.16}$ | $48.53_{\pm3.41}$ | $48.53_{\pm3.41}$ | $0.1_{\pm0.17}$ | $29.75_{\pm9.05}$ | $29.75_{\pm9.05}$ |
| $\hookrightarrow$ + P | $50.77_{\pm13.59}$ | $83.0_{\pm4.71}$ | $83.0_{\pm4.71}$ | $1.87_{\pm3.24}$ | $62.22_{\pm7.28}$ | $62.22_{\pm7.28}$ | $3.26_{\pm5.64}$ | $52.37_{\pm11.98}$ | $52.37_{\pm11.98}$ |
| $\hookrightarrow$ + ours | $\mathbf{70.2_{\pm13.89}}$ | $\mathbf{84.43_{\pm3.2}}$ | $\mathbf{84.43_{\pm3.2}}$ | $21.84_{\pm7.27}$ | $\mathbf{76.98_{\pm0.48}}$ | $\mathbf{76.98_{\pm0.48}}$ | $8.75_{\pm9.03}$ | $\mathbf{61.55_{\pm6.52}}$ | $\mathbf{61.55_{\pm6.52}}$ |

Table B16: Accuracy on the **fifth task** at the end of training, in the *clear* setting. + *ours* refers to combining the baselines with prototypes and FGH. Best HP refer to the best set of LR and $\gamma$ found on VTAB. In some cases, the best HP is the same as one of the default HP values.

| Dataset | CIFAR100 | | | CUB | | | Imagenet-R | | |
|---|---|---|---|---|---|---|---|---|---|
| Learning Rate | $5 \times 10^{-5}$ | $5 \times 10^{-3}$ | Best HP | $5 \times 10^{-5}$ | $5 \times 10^{-3}$ | Best HP | $5 \times 10^{-5}$ | $5 \times 10^{-3}$ | Best HP |
| Fine-tuning | $0.93_{\pm1.52}$ | $0.0_{\pm0.0}$ | $0.0_{\pm0.0}$ | $0.0_{\pm0.0}$ | $0.0_{\pm0.0}$ | $0.0_{\pm0.0}$ | $0.0_{\pm0.0}$ | $0.0_{\pm0.0}$ | $0.0_{\pm0.0}$ |
| Linear probe | $9.81_{\pm5.26}$ | $2.28_{\pm2.32}$ | $2.28_{\pm2.32}$ | $0.56_{\pm0.99}$ | $11.15_{\pm7.31}$ | $11.15_{\pm7.31}$ | $0.81_{\pm0.56}$ | $3.85_{\pm2.62}$ | $3.85_{\pm2.62}$ |
| ER | $63.41_{\pm6.25}$ | $0.0_{\pm0.0}$ | $63.41_{\pm6.25}$ | $\mathbf{34.64_{\pm10.14}}$ | $0.0_{\pm0.0}$ | $34.64_{\pm10.14}$ | $\mathbf{36.64_{\pm3.55}}$ | $0.0_{\pm0.0}$ | $36.64_{\pm3.55}$ |
| ER + Linear probe | $41.63_{\pm7.56}$ | $68.84_{\pm6.22}$ | $64.93_{\pm7.41}$ | $3.4_{\pm3.28}$ | $35.35_{\pm20.24}$ | $34.66_{\pm12.81}$ | $14.98_{\pm4.75}$ | $28.69_{\pm4.92}$ | $28.74_{\pm5.22}$ |
| MVP | $12.56_{\pm7.83}$ | $9.92_{\pm7.88}$ | – | $1.27_{\pm1.93}$ | $12.63_{\pm10.23}$ | – | $0.72_{\pm0.69}$ | $6.29_{\pm4.85}$ | – |
| oLoRA | $34.6_{\pm9.55}$ | $0.0_{\pm0.0}$ | $0.0_{\pm0.0}$ | $2.0_{\pm1.4}$ | $13.13_{\pm6.84}$ | $23.43_{\pm8.08}$ | $0.0_{\pm0.0}$ | $1.69_{\pm1.84}$ | $22.22_{\pm20.04}$ |
| CODA | $24.51_{\pm10.15}$ | $64.12_{\pm13.27}$ | – | $1.48_{\pm1.91}$ | $43.67_{\pm12.95}$ | – | $2.04_{\pm1.19}$ | $31.29_{\pm8.09}$ | – |
| $\hookrightarrow$ + P | $57.74_{\pm9.04}$ | $75.1_{\pm8.49}$ | – | $3.55_{\pm2.97}$ | $48.63_{\pm13.71}$ | – | $9.06_{\pm2.24}$ | $38.62_{\pm9.98}$ | – |
| $\hookrightarrow$ + FGH | $32.28_{\pm10.92}$ | $53.51_{\pm6.12}$ | – | $1.48_{\pm1.91}$ | $44.86_{\pm13.58}$ | – | – | – | – |
| $\hookrightarrow$ + ours | $66.78_{\pm6.12}$ | $74.97_{\pm3.88}$ | – | $21.62_{\pm8.06}$ | $55.87_{\pm9.17}$ | – | $33.0_{\pm8.61}$ | $44.49_{\pm6.31}$ | – |
| L2P | $21.37_{\pm14.86}$ | $55.98_{\pm8.42}$ | $55.98_{\pm8.42}$ | $0.32_{\pm1.0}$ | $27.7_{\pm10.63}$ | $27.7_{\pm10.63}$ | $1.06_{\pm2.36}$ | $21.73_{\pm8.48}$ | $21.73_{\pm8.48}$ |
| $\hookrightarrow$ + P | $49.55_{\pm9.79}$ | $77.87_{\pm5.03}$ | $77.87_{\pm5.03}$ | $0.26_{\pm0.59}$ | $62.67_{\pm11.48}$ | $62.67_{\pm11.48}$ | $8.35_{\pm3.45}$ | $51.25_{\pm4.32}$ | $51.25_{\pm4.32}$ |
| $\hookrightarrow$ + FGH | $30.11_{\pm18.23}$ | $49.08_{\pm8.94}$ | $49.08_{\pm8.94}$ | $0.32_{\pm1.0}$ | $26.16_{\pm10.18}$ | $26.16_{\pm10.18}$ | – | – | – |
| $\hookrightarrow$ + ours | $61.33_{\pm9.61}$ | $77.23_{\pm6.29}$ | $77.23_{\pm6.29}$ | $8.68_{\pm8.36}$ | $60.0_{\pm14.31}$ | $60.0_{\pm14.31}$ | $28.57_{\pm8.45}$ | $55.5_{\pm5.28}$ | $55.5_{\pm5.28}$ |
| DualPrompt | $22.8_{\pm8.35}$ | $48.66_{\pm11.73}$ | – | $1.77_{\pm2.11}$ | $47.92_{\pm11.94}$ | – | $0.98_{\pm0.82}$ | $34.14_{\pm5.94}$ | – |
| $\hookrightarrow$ + P | $52.89_{\pm7.42}$ | $56.22_{\pm8.72}$ | – | $3.39_{\pm2.12}$ | $66.59_{\pm12.65}$ | – | $9.2_{\pm2.77}$ | $50.07_{\pm6.14}$ | – |
| $\hookrightarrow$ + FGH | $32.04_{\pm9.04}$ | $30.79_{\pm11.01}$ | – | $1.77_{\pm2.11}$ | $49.39_{\pm12.19}$ | – | – | – | – |
| $\hookrightarrow$ + ours | $64.6_{\pm6.35}$ | $61.13_{\pm11.36}$ | – | $19.43_{\pm6.96}$ | $\mathbf{68.81_{\pm9.17}}$ | – | $31.61_{\pm5.24}$ | $53.48_{\pm7.11}$ | – |
| ConvPrompt | $44.0_{\pm5.37}$ | $67.1_{\pm6.49}$ | $67.1_{\pm6.49}$ | $0.56_{\pm0.97}$ | $67.53_{\pm3.13}$ | $\mathbf{67.53_{\pm3.13}}$ | $0.22_{\pm0.21}$ | $34.65_{\pm9.55}$ | $34.65_{\pm9.55}$ |
| $\hookrightarrow$ + P | $66.47_{\pm10.37}$ | $83.0_{\pm3.34}$ | $83.0_{\pm3.34}$ | $4.75_{\pm1.01}$ | $66.53_{\pm16.34}$ | $66.53_{\pm16.34}$ | $2.32_{\pm3.74}$ | $48.11_{\pm1.56}$ | $48.11_{\pm1.56}$ |
| $\hookrightarrow$ + ours | $\mathbf{76.9_{\pm7.73}}$ | $\mathbf{86.33_{\pm1.5}}$ | $\mathbf{86.33_{\pm1.5}}$ | $22.61_{\pm1.32}$ | $64.88_{\pm7.12}$ | $64.88_{\pm7.12}$ | $4.09_{\pm5.81}$ | $\mathbf{61.05_{\pm1.22}}$ | $\mathbf{61.05_{\pm1.22}}$ |

Table B17: Accuracy on the **sixth task** at the end of training, in the *clear* setting. *+ ours* refers to combining the baselines with prototypes and FGH. Best HP refer to the best set of LR and $\gamma$ found on VTAB. In some cases, the best HP is the same as one of the default HP values.

| Dataset | CIFAR100 | | | CUB | | | Imagenet-R | | |
|---|---|---|---|---|---|---|---|---|---|
| Learning Rate | $5 \times 10^{-5}$ | $5 \times 10^{-3}$ | Best HP | $5 \times 10^{-5}$ | $5 \times 10^{-3}$ | Best HP | $5 \times 10^{-5}$ | $5 \times 10^{-3}$ | Best HP |
| Fine-tuning | $0.73_{\pm2.07}$ | $0.0_{\pm0.0}$ | $0.0_{\pm0.0}$ | $0.0_{\pm0.0}$ | $0.0_{\pm0.0}$ | $0.0_{\pm0.0}$ | $0.0_{\pm0.0}$ | $0.0_{\pm0.0}$ | $0.0_{\pm0.0}$ |
| Linear probe | $13.07_{\pm3.73}$ | $3.92_{\pm3.06}$ | $3.92_{\pm3.06}$ | $0.42_{\pm0.71}$ | $10.52_{\pm8.85}$ | $10.52_{\pm8.85}$ | $1.09_{\pm1.04}$ | $9.54_{\pm6.91}$ | $9.54_{\pm6.91}$ |
| ER | $64.01_{\pm9.26}$ | $0.07_{\pm0.22}$ | $64.01_{\pm9.26}$ | $31.7_{\pm12.63}$ | $0.0_{\pm0.0}$ | $31.7_{\pm12.63}$ | $34.51_{\pm6.64}$ | $0.0_{\pm0.0}$ | $34.51_{\pm6.64}$ |
| ER + Linear probe | $41.2_{\pm7.94}$ | $68.26_{\pm8.22}$ | $70.19_{\pm8.9}$ | $3.42_{\pm3.23}$ | $33.78_{\pm15.96}$ | $33.24_{\pm12.44}$ | $11.83_{\pm5.3}$ | $33.14_{\pm9.33}$ | $31.96_{\pm9.41}$ |
| MVP | $18.89_{\pm6.0}$ | $10.32_{\pm9.15}$ | – | $0.9_{\pm1.68}$ | $10.67_{\pm9.53}$ | – | $1.21_{\pm1.1}$ | $11.86_{\pm10.9}$ | – |
| oLoRA | $29.2_{\pm17.33}$ | $0.0_{\pm0.0}$ | $0.0_{\pm0.0}$ | $17.48_{\pm23.71}$ | $5.56_{\pm5.52}$ | $23.82_{\pm9.07}$ | $0.56_{\pm0.71}$ | $2.84_{\pm3.68}$ | $22.48_{\pm14.74}$ |
| CODA | $33.59_{\pm13.57}$ | $71.66_{\pm9.87}$ | – | $0.51_{\pm0.58}$ | $46.03_{\pm13.74}$ | – | $2.21_{\pm1.21}$ | $37.22_{\pm10.13}$ | – |
| $\hookrightarrow$ + P | $54.8_{\pm8.31}$ | $80.35_{\pm5.92}$ | – | $1.75_{\pm1.65}$ | $53.25_{\pm11.43}$ | – | $9.67_{\pm2.49}$ | $43.61_{\pm10.07}$ | – |
| $\hookrightarrow$ + FGH | $43.81_{\pm14.33}$ | $59.88_{\pm9.37}$ | – | $0.51_{\pm0.58}$ | $45.81_{\pm10.93}$ | – | – | – | – |
| $\hookrightarrow$ + ours | $\mathbf{66.31_{\pm8.42}}$ | $78.3_{\pm6.8}$ | – | $13.08_{\pm6.36}$ | $54.34_{\pm15.83}$ | – | $33.15_{\pm7.66}$ | $45.94_{\pm8.92}$ | – |
| L2P | $28.82_{\pm15.82}$ | $61.31_{\pm13.86}$ | $61.31_{\pm13.86}$ | $1.01_{\pm2.11}$ | $32.25_{\pm11.31}$ | $32.25_{\pm11.31}$ | $0.98_{\pm1.39}$ | $29.08_{\pm6.32}$ | $29.08_{\pm6.32}$ |
| $\hookrightarrow$ + P | $51.25_{\pm7.26}$ | $80.92_{\pm5.0}$ | $80.92_{\pm5.0}$ | $1.73_{\pm2.29}$ | $65.31_{\pm8.23}$ | $65.31_{\pm8.23}$ | $6.55_{\pm6.26}$ | $53.65_{\pm7.73}$ | $53.65_{\pm7.73}$ |
| $\hookrightarrow$ + FGH | $39.19_{\pm16.86}$ | $49.36_{\pm13.53}$ | $49.36_{\pm13.53}$ | $1.01_{\pm2.11}$ | $30.8_{\pm9.84}$ | $30.8_{\pm9.84}$ | – | – | – |
| $\hookrightarrow$ + ours | $61.4_{\pm7.81}$ | $77.7_{\pm6.73}$ | $77.7_{\pm6.73}$ | $7.16_{\pm7.61}$ | $69.63_{\pm7.38}$ | $\mathbf{69.63_{\pm7.38}}$ | $27.77_{\pm12.23}$ | $58.42_{\pm4.51}$ | $58.42_{\pm4.51}$ |
| DualPrompt | $29.3_{\pm8.28}$ | $62.5_{\pm11.34}$ | – | $0.25_{\pm0.41}$ | $49.04_{\pm12.54}$ | – | $1.08_{\pm1.45}$ | $42.18_{\pm8.15}$ | – |
| $\hookrightarrow$ + P | $52.76_{\pm4.7}$ | $66.65_{\pm12.69}$ | – | $3.0_{\pm2.68}$ | $70.42_{\pm8.72}$ | – | $8.05_{\pm6.19}$ | $53.06_{\pm4.84}$ | – |
| $\hookrightarrow$ + FGH | $39.17_{\pm8.27}$ | $43.89_{\pm15.11}$ | – | $0.25_{\pm0.41}$ | $49.07_{\pm12.41}$ | – | – | – | – |
| $\hookrightarrow$ + ours | $63.12_{\pm4.29}$ | $72.07_{\pm12.75}$ | – | $16.38_{\pm8.14}$ | $72.04_{\pm8.71}$ | – | $32.13_{\pm11.03}$ | $56.05_{\pm5.53}$ | – |
| ConvPrompt | $30.97_{\pm6.3}$ | $58.27_{\pm9.92}$ | $58.27_{\pm9.92}$ | $1.0_{\pm0.87}$ | $62.72_{\pm7.53}$ | $62.72_{\pm7.53}$ | $0.21_{\pm0.37}$ | $46.15_{\pm7.21}$ | $46.15_{\pm7.21}$ |
| $\hookrightarrow$ + P | $55.43_{\pm15.51}$ | $83.03_{\pm3.93}$ | $83.03_{\pm3.93}$ | $4.26_{\pm3.79}$ | $61.88_{\pm20.85}$ | $61.88_{\pm20.85}$ | $1.77_{\pm1.79}$ | $48.14_{\pm6.73}$ | $48.14_{\pm6.73}$ |
| $\hookrightarrow$ + ours | $61.5_{\pm14.34}$ | $\mathbf{83.97_{\pm2.36}}$ | $\mathbf{83.97_{\pm2.36}}$ | $\mathbf{18.93_{\pm12.97}}$ | $39.28_{\pm29.31}$ | $39.28_{\pm29.31}$ | $7.84_{\pm6.81}$ | $\mathbf{63.22_{\pm4.16}}$ | $\mathbf{63.22_{\pm4.16}}$ |

Table B18: Accuracy on the **seventh task** at the end of training, in the *clear* setting. *+ ours* refers to combining the baselines with prototypes and FGH. Best HP refer to the best set of LR and $\gamma$ found on VTAB. In some cases, the best HP is the same as one of the default HP values.

| Dataset | CIFAR100 | | | CUB | | | Imagenet-R | | |
|---|---|---|---|---|---|---|---|---|---|
| Learning Rate | $5 \times 10^{-5}$ | $5 \times 10^{-3}$ | Best HP | $5 \times 10^{-5}$ | $5 \times 10^{-3}$ | Best HP | $5 \times 10^{-5}$ | $5 \times 10^{-3}$ | Best HP |
| Fine-tuning | $0.13_{\pm0.16}$ | $0.0_{\pm0.0}$ | $0.0_{\pm0.0}$ | $0.0_{\pm0.0}$ | $0.0_{\pm0.0}$ | $0.0_{\pm0.0}$ | $0.0_{\pm0.0}$ | $0.0_{\pm0.0}$ | $0.0_{\pm0.0}$ |
| Linear probe | $22.16_{\pm6.73}$ | $6.58_{\pm4.53}$ | $6.58_{\pm4.53}$ | $1.4_{\pm1.7}$ | $13.16_{\pm11.44}$ | $13.16_{\pm11.44}$ | $2.12_{\pm2.2}$ | $15.51_{\pm8.55}$ | $15.51_{\pm8.55}$ |
| ER | $68.97_{\pm6.63}$ | $0.0_{\pm0.0}$ | $68.97_{\pm6.63}$ | $\mathbf{38.09_{\pm8.72}}$ | $0.0_{\pm0.0}$ | $38.09_{\pm8.72}$ | $\mathbf{38.3_{\pm5.45}}$ | $0.0_{\pm0.0}$ | $38.3_{\pm5.45}$ |
| ER + Linear probe | $41.97_{\pm8.36}$ | $70.56_{\pm6.34}$ | $69.34_{\pm6.99}$ | $4.65_{\pm4.13}$ | $36.52_{\pm12.07}$ | $38.45_{\pm11.58}$ | $13.32_{\pm3.15}$ | $33.8_{\pm7.94}$ | $33.4_{\pm8.26}$ |
| MVP | $21.45_{\pm7.62}$ | $13.14_{\pm8.6}$ | – | $0.66_{\pm1.54}$ | $13.35_{\pm11.27}$ | – | $1.15_{\pm1.05}$ | $15.29_{\pm10.45}$ | – |
| oLoRA | $49.03_{\pm23.85}$ | $0.0_{\pm0.0}$ | $0.03_{\pm0.06}$ | $4.18_{\pm3.09}$ | $20.94_{\pm18.88}$ | $20.29_{\pm21.12}$ | $0.52_{\pm0.66}$ | $3.73_{\pm4.57}$ | $14.3_{\pm5.79}$ |
| CODA | $31.2_{\pm7.85}$ | $74.07_{\pm10.53}$ | – | $0.67_{\pm0.88}$ | $46.45_{\pm15.23}$ | – | $1.33_{\pm0.79}$ | $43.0_{\pm16.61}$ | – |
| $\hookrightarrow$ + P | $55.1_{\pm6.79}$ | $81.48_{\pm5.37}$ | – | $2.98_{\pm3.25}$ | $50.1_{\pm18.11}$ | – | $7.23_{\pm2.65}$ | $49.76_{\pm11.35}$ | – |
| $\hookrightarrow$ + FGH | $44.49_{\pm9.48}$ | $66.99_{\pm12.44}$ | – | $0.67_{\pm0.88}$ | $42.74_{\pm9.92}$ | – | – | – | – |
| $\hookrightarrow$ + ours | $68.73_{\pm5.59}$ | $79.73_{\pm8.31}$ | – | $17.57_{\pm7.11}$ | $57.91_{\pm17.99}$ | – | $30.76_{\pm8.49}$ | $48.14_{\pm10.53}$ | – |
| L2P | $25.11_{\pm14.27}$ | $66.08_{\pm4.84}$ | $66.08_{\pm4.84}$ | $0.17_{\pm0.54}$ | $37.17_{\pm17.05}$ | $37.17_{\pm17.05}$ | $1.27_{\pm1.43}$ | $32.57_{\pm9.71}$ | $32.57_{\pm9.71}$ |
| $\hookrightarrow$ + P | $44.49_{\pm9.29}$ | $82.29_{\pm4.67}$ | $82.29_{\pm4.67}$ | $0.99_{\pm1.85}$ | $68.68_{\pm10.22}$ | $68.68_{\pm10.22}$ | $9.52_{\pm5.2}$ | $57.63_{\pm5.72}$ | $57.63_{\pm5.72}$ |
| $\hookrightarrow$ + FGH | $36.2_{\pm14.93}$ | $59.06_{\pm11.95}$ | $59.06_{\pm11.95}$ | $0.17_{\pm0.54}$ | $37.69_{\pm19.45}$ | $37.69_{\pm19.45}$ | – | – | – |
| $\hookrightarrow$ + ours | $51.86_{\pm9.29}$ | $80.26_{\pm4.61}$ | $80.26_{\pm4.61}$ | $4.89_{\pm5.15}$ | $72.35_{\pm9.7}$ | $72.35_{\pm9.7}$ | $29.94_{\pm10.56}$ | $61.92_{\pm5.79}$ | $61.92_{\pm5.79}$ |
| DualPrompt | $30.55_{\pm8.18}$ | $66.76_{\pm12.01}$ | – | $0.76_{\pm1.01}$ | $55.84_{\pm11.41}$ | – | $1.96_{\pm0.73}$ | $47.9_{\pm11.97}$ | – |
| $\hookrightarrow$ + P | $44.59_{\pm9.09}$ | $77.4_{\pm9.19}$ | – | $3.76_{\pm3.5}$ | $67.07_{\pm9.02}$ | – | $9.72_{\pm5.31}$ | $55.74_{\pm7.13}$ | – |
| $\hookrightarrow$ + FGH | $40.53_{\pm9.47}$ | $51.57_{\pm10.52}$ | – | $0.76_{\pm1.01}$ | $56.11_{\pm12.18}$ | – | – | – | – |
| $\hookrightarrow$ + ours | $54.84_{\pm8.22}$ | $81.02_{\pm4.51}$ | – | $16.57_{\pm6.34}$ | $74.98_{\pm8.11}$ | – | $32.2_{\pm10.55}$ | $56.31_{\pm7.76}$ | – |
| ConvPrompt | $55.4_{\pm8.66}$ | $80.73_{\pm10.6}$ | $80.73_{\pm10.6}$ | $0.8_{\pm0.79}$ | $59.11_{\pm25.48}$ | $59.11_{\pm25.48}$ | $0.0_{\pm0.0}$ | $43.15_{\pm11.51}$ | $43.15_{\pm11.51}$ |
| $\hookrightarrow$ + P | $\mathbf{77.07_{\pm6.13}}$ | $85.47_{\pm3.49}$ | $85.47_{\pm3.49}$ | $4.66_{\pm1.35}$ | $62.52_{\pm9.51}$ | $62.52_{\pm9.51}$ | $2.08_{\pm0.76}$ | $52.51_{\pm7.86}$ | $52.51_{\pm7.86}$ |
| $\hookrightarrow$ + ours | $76.87_{\pm5.83}$ | $\mathbf{88.17_{\pm2.16}}$ | $\mathbf{88.17_{\pm2.16}}$ | $18.83_{\pm0.87}$ | $\mathbf{80.46_{\pm4.66}}$ | $\mathbf{80.46_{\pm4.66}}$ | $6.67_{\pm0.49}$ | $\mathbf{63.2_{\pm4.64}}$ | $\mathbf{63.2_{\pm4.64}}$ |

Table B19: Accuracy on the **eighth task** at the end of training, in the *clear* setting. + *ours* refers to combining the baselines with prototypes and FGH. Best HP refer to the best set of LR and $\gamma$ found on VTAB. In some cases, the best HP is the same as one of the default HP values.

| Dataset | CIFAR100 | | | CUB | | | Imagenet-R | | |
|---|---|---|---|---|---|---|---|---|---|
| Learning Rate | $5 \times 10^{-5}$ | $5 \times 10^{-3}$ | Best HP | $5 \times 10^{-5}$ | $5 \times 10^{-3}$ | Best HP | $5 \times 10^{-5}$ | $5 \times 10^{-3}$ | Best HP |
| Fine-tuning | 1.28±1.79 | 0.0±0.0 | 0.0±0.0 | 0.0±0.0 | 0.0±0.0 | 0.0±0.0 | 0.0±0.0 | 0.0±0.0 | 0.0±0.0 |
| Linear probe | 27.21±8.14 | 15.77±7.48 | 15.77±7.48 | 1.62±2.29 | 20.79±15.04 | 20.79±15.04 | 2.07±1.69 | 16.08±5.09 | 16.08±5.09 |
| ER | 73.81±5.46 | 0.0±0.0 | 73.81±5.46 | **30.82±10.29** | 0.0±0.0 | 30.82±10.29 | **37.82±9.88** | 0.0±0.0 | 37.82±9.88 |
| ER + Linear probe | 39.34±9.68 | 77.82±4.76 | 75.47±5.01 | 3.16±3.3 | 42.6±18.18 | 42.98±16.16 | 11.01±5.37 | 34.68±6.62 | 35.18±6.69 |
| MVP | 35.61±9.12 | 17.22±10.87 | – | 1.18±1.69 | 20.2±14.8 | – | 1.71±1.5 | 17.97±12.5 | – |
| oLoRA | 65.87±19.23 | 0.1±0.17 | 0.03±0.06 | 0.24±0.42 | 8.69±8.36 | 18.32±17.59 | 3.32±2.33 | 10.12±1.57 | 44.6±1.66 |
| CODA | 37.34±12.63 | 79.84±7.43 | – | 0.51±1.11 | 55.28±11.95 | – | 2.49±1.67 | 46.88±5.68 | – |
| ↪ + P | 55.88±8.16 | 86.52±4.15 | – | 2.29±3.04 | 59.99±20.43 | – | 8.71±2.86 | 48.02±7.24 | – |
| ↪ + FGH | 52.98±12.62 | 68.37±9.02 | – | 0.51±1.11 | 52.32±13.12 | – | – | – | – |
| ↪ + ours | 73.32±5.49 | 84.08±3.49 | – | 10.94±8.19 | 58.11±17.21 | – | 27.13±5.49 | 46.66±8.8 | – |
| L2P | 34.34±17.96 | 70.62±9.41 | 70.62±9.41 | 1.01±2.23 | 39.43±12.8 | 39.43±12.8 | 2.61±3.36 | 36.12±8.1 | 36.12±8.1 |
| ↪ + P | 46.98±9.76 | **87.1±3.42** | **87.1±3.42** | 1.09±1.76 | **76.13±10.45** | **76.13±10.45** | 8.07±3.48 | 58.0±6.35 | 58.0±6.35 |
| ↪ + FGH | 46.26±18.21 | 59.06±7.8 | 59.06±7.8 | 1.08±2.33 | 38.35±13.37 | 38.35±13.37 | – | – | – |
| ↪ + ours | 53.13±9.61 | 85.19±3.61 | 85.19±3.61 | 7.41±6.41 | 74.01±10.55 | 74.01±10.55 | 26.95±4.41 | 61.55±6.64 | 61.55±6.64 |
| DualPrompt | 38.12±12.65 | 76.59±9.45 | – | 1.66±2.83 | 58.03±8.3 | – | 1.22±0.86 | 47.18±10.63 | – |
| ↪ + P | 47.04±10.12 | 83.64±4.23 | – | 3.34±4.18 | 71.79±9.35 | – | 6.69±3.31 | 53.29±6.83 | – |
| ↪ + FGH | 48.73±12.61 | 60.69±6.95 | – | 1.66±2.83 | 59.67±9.71 | – | – | – | – |
| ↪ + ours | 57.35±9.48 | 83.94±3.24 | – | 12.95±8.2 | 73.56±9.2 | – | 26.6±4.67 | 53.26±6.73 | – |
| ConvPrompt | 58.23±24.93 | 75.9±5.14 | 75.9±5.14 | 0.0±0.0 | 73.07±8.8 | 73.07±8.8 | 0.0±0.0 | 53.12±12.48 | 53.12±12.48 |
| ↪ + P | 78.4±11.57 | 83.67±7.12 | 83.67±7.12 | 1.41±1.3 | 71.02±25.33 | 71.02±25.33 | 1.07±1.1 | 60.95±8.47 | 60.95±8.47 |
| ↪ + ours | **82.83±7.77** | 84.4±2.03 | 84.4±2.03 | 8.3±6.0 | 73.04±15.16 | 73.04±15.16 | 8.65±3.25 | **64.4±5.98** | **64.4±5.98** |

Table B20: Accuracy on the **ninth task** at the end of training, in the *clear* setting. + *ours* refers to combining the baselines with prototypes and FGH. Best HP refer to the best set of LR and $\gamma$ found on VTAB. In some cases, the best HP is the same as one of the default HP values.

| Dataset | CIFAR100 | | | CUB | | | Imagenet-R | | |
|---|---|---|---|---|---|---|---|---|---|
| Learning Rate | $5 \times 10^{-5}$ | $5 \times 10^{-3}$ | Best HP | $5 \times 10^{-5}$ | $5 \times 10^{-3}$ | Best HP | $5 \times 10^{-5}$ | $5 \times 10^{-3}$ | Best HP |
| Fine-tuning | 6.15±9.25 | 0.0±0.0 | 0.0±0.0 | 0.0±0.0 | 0.0±0.0 | 0.0±0.0 | 0.0±0.0 | 0.0±0.0 | 0.0±0.0 |
| Linear probe | 37.66±6.23 | 22.6±8.2 | 22.6±8.2 | 0.72±0.78 | 30.8±14.39 | 30.8±14.39 | 2.08±1.17 | 17.13±7.51 | 17.13±7.51 |
| ER | 78.51±4.8 | 0.0±0.0 | 78.51±4.8 | **37.45±11.88** | 0.0±0.0 | 37.45±11.88 | **35.92±8.73** | 0.0±0.0 | 35.92±8.73 |
| ER + Linear probe | 40.88±6.63 | 80.3±4.91 | 79.28±5.72 | 3.07±2.03 | 52.04±15.48 | 44.13±12.89 | 10.72±5.31 | 36.53±6.67 | 34.89±7.23 |
| MVP | 43.58±6.45 | 23.27±14.93 | – | 0.9±1.23 | 26.76±12.6 | – | 3.0±2.7 | 20.39±13.29 | – |
| oLoRA | **80.87±4.14** | 0.23±0.32 | 0.4±0.61 | 2.14±2.14 | 20.88±5.05 | 19.02±20.12 | 1.67±2.33 | 11.98±11.8 | 32.84±11.87 |
| CODA | 42.42±14.64 | 83.39±4.62 | – | 1.14±2.33 | 60.37±9.36 | – | 3.44±2.28 | 47.25±7.21 | – |
| ↪ + P | 51.25±9.76 | **88.28±2.32** | – | 2.82±4.4 | 65.46±13.16 | – | 8.87±3.63 | 49.62±10.67 | – |
| ↪ + FGH | 59.44±14.55 | 77.23±8.62 | – | 1.14±2.33 | 58.56±11.27 | – | – | – | – |
| ↪ + ours | 72.18±5.34 | 86.14±1.98 | – | 11.26±6.88 | 65.34±14.14 | – | 26.59±6.41 | 47.63±6.6 | – |
| L2P | 32.75±12.29 | 77.24±6.53 | 77.24±6.53 | 0.0±0.0 | 45.99±7.22 | 45.99±7.22 | 1.58±2.11 | 38.1±7.27 | 38.1±7.27 |
| ↪ + P | 37.89±7.12 | 86.99±3.6 | 86.99±3.6 | 1.2±1.5 | 79.37±7.6 | 79.37±7.6 | 5.97±4.42 | 60.38±5.11 | 60.38±5.11 |
| ↪ + FGH | 42.71±13.2 | 67.35±10.43 | 67.35±10.43 | 0.0±0.0 | 44.37±9.34 | 44.37±9.34 | – | – | – |
| ↪ + ours | 41.6±8.64 | 87.25±3.82 | **87.25±3.82** | 5.39±4.14 | 78.19±5.59 | 78.19±5.59 | 20.78±5.93 | 63.4±4.43 | 63.4±4.43 |
| DualPrompt | 34.09±8.35 | 79.19±6.65 | – | 1.06±1.42 | 65.76±7.33 | – | 1.93±1.2 | 50.17±6.1 | – |
| ↪ + P | 40.39±7.06 | 86.63±2.82 | – | 3.35±3.21 | 74.8±6.89 | – | 7.29±4.18 | 56.79±8.28 | – |
| ↪ + FGH | 45.17±8.12 | 63.4±7.34 | – | 1.06±1.42 | 66.85±7.3 | – | – | – | – |
| ↪ + ours | 48.94±6.67 | 86.89±2.7 | – | 12.33±4.95 | 79.27±6.35 | – | 23.57±8.43 | 55.08±7.4 | – |
| ConvPrompt | 39.53±3.8 | 76.8±9.07 | 76.8±9.07 | 1.61±1.4 | 68.49±7.69 | 68.49±7.69 | 2.31±4.0 | 45.3±3.94 | 45.3±3.94 |
| ↪ + P | 74.43±6.01 | 84.93±5.8 | 84.93±5.8 | 4.56±1.66 | 71.92±8.61 | 71.92±8.61 | 3.72±6.21 | 58.85±4.05 | 58.85±4.05 |
| ↪ + ours | 74.53±5.64 | 87.0±2.25 | 87.0±2.25 | 19.21±6.48 | **81.4±3.92** | **81.4±3.92** | 8.54±9.95 | **65.56±2.62** | **65.56±2.62** |

Table B21: Accuracy on the **tenth task** at the end of training, in the *clear* setting. *+ ours* refers to combining the baselines with prototypes and FGH. Best HP refer to the best set of LR and $\gamma$ found on VTAB. In some cases, the best HP is the same as one of the default HP values.

| Dataset | CIFAR100 | | | CUB | | | Imagenet-R | | |
|---|---|---|---|---|---|---|---|---|---|
| Learning Rate | $5 \times 10^{-5}$ | $5 \times 10^{-3}$ | Best HP | $5 \times 10^{-5}$ | $5 \times 10^{-3}$ | Best HP | $5 \times 10^{-5}$ | $5 \times 10^{-3}$ | Best HP |
| Fine-tuning | $80.73_{\pm18.69}$ | $10.0_{\pm0.0}$ | $10.0_{\pm0.0}$ | $0.0_{\pm0.0}$ | $0.0_{\pm0.0}$ | $1.13_{\pm3.58}$ | $0.0_{\pm0.0}$ | $0.0_{\pm0.0}$ | $0.0_{\pm0.0}$ |
| Linear probe | $55.21_{\pm6.6}$ | $\mathbf{93.38_{\pm3.35}}$ | $93.38_{\pm3.35}$ | $1.73_{\pm1.83}$ | $21.96_{\pm9.99}$ | $21.96_{\pm9.99}$ | $2.1_{\pm2.15}$ | $23.77_{\pm6.44}$ | $23.77_{\pm6.44}$ |
| ER | $\mathbf{95.46_{\pm2.8}}$ | $10.26_{\pm0.89}$ | $95.46_{\pm2.8}$ | $\mathbf{25.22_{\pm9.33}}$ | $0.0_{\pm0.0}$ | $25.22_{\pm9.33}$ | $\mathbf{39.17_{\pm8.62}}$ | $0.0_{\pm0.0}$ | $39.17_{\pm8.62}$ |
| ER + Linear probe | $42.54_{\pm7.05}$ | $89.44_{\pm3.86}$ | $89.02_{\pm4.04}$ | $4.68_{\pm4.22}$ | $33.71_{\pm8.96}$ | $36.04_{\pm8.12}$ | $5.86_{\pm3.92}$ | $36.89_{\pm6.33}$ | $35.71_{\pm7.5}$ |
| MVP | $55.04_{\pm3.94}$ | $93.63_{\pm3.2}$ | – | $1.74_{\pm2.36}$ | $19.32_{\pm8.88}$ | – | $2.43_{\pm2.19}$ | $23.85_{\pm13.53}$ | – |
| oLoRA | $86.47_{\pm1.15}$ | $38.7_{\pm22.78}$ | $49.5_{\pm14.2}$ | $5.5_{\pm7.1}$ | $20.2_{\pm5.92}$ | $8.94_{\pm4.5}$ | $1.1_{\pm0.97}$ | $15.08_{\pm2.23}$ | $27.32_{\pm16.16}$ |
| CODA | $39.71_{\pm13.53}$ | $82.54_{\pm7.16}$ | – | $0.46_{\pm0.66}$ | $63.39_{\pm12.83}$ | – | $1.76_{\pm1.51}$ | $53.4_{\pm9.88}$ | – |
| $\hookrightarrow$ + P | $42.32_{\pm11.7}$ | $92.15_{\pm3.14}$ | – | $2.18_{\pm3.5}$ | $59.69_{\pm12.16}$ | – | $6.46_{\pm3.25}$ | $54.98_{\pm10.4}$ | – |
| $\hookrightarrow$ + FGH | $53.67_{\pm13.36}$ | $72.21_{\pm13.67}$ | – | $0.46_{\pm0.66}$ | $58.35_{\pm13.41}$ | – | – | – | – |
| $\hookrightarrow$ + ours | $59.56_{\pm10.73}$ | $91.85_{\pm2.25}$ | – | $9.76_{\pm8.99}$ | $60.87_{\pm14.56}$ | – | $24.92_{\pm4.8}$ | $50.4_{\pm7.78}$ | – |
| L2P | $30.16_{\pm12.98}$ | $74.09_{\pm9.14}$ | $74.09_{\pm9.14}$ | $0.5_{\pm0.78}$ | $39.0_{\pm8.32}$ | $39.0_{\pm8.32}$ | $2.15_{\pm1.9}$ | $46.55_{\pm4.41}$ | $46.55_{\pm4.41}$ |
| $\hookrightarrow$ + P | $29.04_{\pm9.53}$ | $87.51_{\pm3.59}$ | $87.51_{\pm3.59}$ | $1.16_{\pm1.38}$ | $73.72_{\pm5.37}$ | $73.72_{\pm5.37}$ | $6.5_{\pm4.87}$ | $63.23_{\pm5.37}$ | $63.23_{\pm5.37}$ |
| $\hookrightarrow$ + FGH | $39.34_{\pm12.82}$ | $65.74_{\pm14.48}$ | $65.74_{\pm14.48}$ | $0.43_{\pm0.79}$ | $39.64_{\pm11.76}$ | $39.64_{\pm11.76}$ | – | – | – |
| $\hookrightarrow$ + ours | $29.14_{\pm10.07}$ | $89.09_{\pm3.51}$ | $89.09_{\pm3.51}$ | $4.43_{\pm4.34}$ | $\mathbf{80.02_{\pm7.78}}$ | $\mathbf{80.02_{\pm7.78}}$ | $22.96_{\pm6.73}$ | $\mathbf{65.11_{\pm4.63}}$ | $\mathbf{65.11_{\pm4.63}}$ |
| DualPrompt | $32.75_{\pm9.51}$ | $82.79_{\pm4.04}$ | – | $0.89_{\pm0.94}$ | $65.36_{\pm8.6}$ | – | $2.82_{\pm1.62}$ | $57.25_{\pm4.51}$ | – |
| $\hookrightarrow$ + P | $29.56_{\pm7.34}$ | $92.91_{\pm3.07}$ | – | $2.26_{\pm1.6}$ | $71.48_{\pm6.8}$ | – | $8.48_{\pm4.47}$ | $59.15_{\pm3.98}$ | – |
| $\hookrightarrow$ + FGH | $42.1_{\pm10.5}$ | $68.25_{\pm9.35}$ | – | $0.89_{\pm0.94}$ | $64.75_{\pm11.8}$ | – | – | – | – |
| $\hookrightarrow$ + ours | $35.38_{\pm7.66}$ | $90.23_{\pm3.24}$ | – | $8.09_{\pm3.88}$ | $75.21_{\pm7.03}$ | – | $27.55_{\pm7.38}$ | $58.69_{\pm3.97}$ | – |
| ConvPrompt | $58.77_{\pm20.94}$ | $83.53_{\pm8.62}$ | $83.53_{\pm8.62}$ | $0.0_{\pm0.0}$ | $71.3_{\pm6.08}$ | $71.3_{\pm6.08}$ | $1.38_{\pm2.39}$ | $49.93_{\pm2.86}$ | $49.93_{\pm2.86}$ |
| $\hookrightarrow$ + P | $74.03_{\pm17.46}$ | $93.63_{\pm3.1}$ | $93.63_{\pm3.1}$ | $0.35_{\pm0.61}$ | $67.19_{\pm8.49}$ | $67.19_{\pm8.49}$ | $1.68_{\pm2.9}$ | $51.55_{\pm13.8}$ | $51.55_{\pm13.8}$ |
| $\hookrightarrow$ + ours | $70.23_{\pm8.9}$ | $92.0_{\pm4.25}$ | $92.0_{\pm4.25}$ | $1.36_{\pm1.62}$ | $73.94_{\pm4.83}$ | $73.94_{\pm4.83}$ | $6.55_{\pm10.72}$ | $60.49_{\pm8.22}$ | $60.49_{\pm8.22}$ |

