# OpenReview forum: "From Offline to Online Memory-Free and Task-Free Continual Learning via Fine-Grained Hypergradients"
_TMLR — Accepted by TMLR_

### Review · Reviewer_Cqv4 · 2026-03-13

**Summary Of Contributions:**

**Summary:**

This paper studies online, task-free, memory-free continual learning focusing on how strong offline prompt-based CL methods can be adapted to this more realistic setting. The proposed approach has two main components: lightweight class prototypes, which act as representation-level memory for the final classifier, and Fine-Grained Hypergradients (FGH), a parameter-wise gradient reweighting rule motivated by hypergradients and intended to mitigate gradient imbalance and reduce sensitivity to the initial learning rate. The paper also advocates evaluating methods across multiple learning rates, rather than reporting only a single fixed-LR result. Experiments on CIFAR100, CUB, and ImageNet-R, under both clear and Si-Blurry settings, show that prototypes provide most of the raw gains, while FGH is useful when combined with prototypes, especially under suboptimal learning rates.

**Strengths:**

The paper addresses a practically relevant setting: online, task-free, memory-free continual learning.

The method is simple and can be integrated into several prompt-based memory-free baselines.

The empirical section is thorough, with results across multiple datasets and settings, and useful ablations.

The paper makes a useful evaluation point that single-LR reporting can be misleading in online CL, and its multi-LR protocol is a valuable addition.

**Weaknesses:**

The novelty is somewhat incremental: prototypes are a simple design choice, and FGH is a natural parameter-wise extension of hypergradient-style adaptation.

The strongest evidence is for the practical usefulness of the combined method; the mechanistic claim that FGH specifically resolves gradient imbalance is plausible but not fully isolated from more generic adaptive rescaling effects.

FGH introduces an additional hyperparameter, and the "best" setting still relies on VTAB-based hyperparameter transfer.

Replay-based methods remain clearly stronger in several settings, so the method does not close the gap to strong memory-based baselines.

**Audience:**

Yes

**Audience Explanation:**

The paper sits at the intersection of continual learning, online learning, and adaptation of pretrained vision models, all of which are of clear interest to the TMLR audience.

**Broader Impact Concerns:**

I do not see major ethical concerns specific to this work beyond the usual considerations associated with continual learning systems in changing environments.

**Claims And Evidence:**

Yes

**Claims Explanation:**

**Overall, I find the evidence convincing for empirical usefulness, though slightly less definitive for the precise mechanistic explanation.**

The authors evaluate on three standard vision datasets, in two settings, over multiple runs, and compare against several important baselines. They also include ablations that separate prototypes, FGH, and their combination. The results consistently show that prototypes substantially improve the adapted baselines, and that the combination of prototypes and FGH usually yields the strongest memory-free results, particularly when the initial learning rate is not well chosen.

The mechanistic claim that FGH specifically mitigates gradient imbalance is reasonable and supported by the paper’s diagnostics, but the evidence remains somewhat qualitative. In particular, the paper does not fully rule out the possibility that the gains come from a broader adaptive gradient-rescaling effect rather than the specific hypergradient-motivated construction.

**Requested Changes:**

1. Better isolate the contribution of FGH from generic adaptive gradient rescaling. For example, compare against simpler per-parameter or per-layer adaptive weighting baselines that do not rely on the proposed hypergradient derivation. This would make the claimed mechanism more convincing.

2. The computational and memory overhead should be mentioned more prominently in the main paper. Since the paper targets online efficiency, a concise summary of the practical overhead would help readers assess the tradeoff more easily.

3. Clarify the fairness of the hyperparameter protocol. The paper argues against single-LR evaluation, but the results still rely on VTAB-based transfer; it would help to report how sensitive the conclusions are to this choice and whether the qualitative ranking remains similar without transferred tuning.

4. Improve writing clarity in a few places, especially around the practical implementation of FGH with momentum/Adam-style updates and the relation between the simple derivation and the actual optimizer used.

---

> ### Author Response · Authors · 2026-03-31
>
> We thank the reviewer for their comments and suggestions. We respond to specific points below and in the shared reply.
>
> ## Prototypes give most of the gain
>
> Our claim is not that FGH independently provides the largest improvement, but that it plays a *critical complementary role*.
>
> As shown in Table 3, using FGH alone may not consistently improve performance. However:
> - Prototypes introduce strong gradient imbalance toward earlier classes,
> - FGH corrects this imbalance, leading to more stable learning over time.
>
> This interaction is also supported by the gradient analysis in Section 5.1, where FGH reverses the imbalance induced by prototypes.
>
> Therefore, the contribution of FGH is not additive in isolation, but *structural*, enabling prototypes to perform effectively in the online setting.
>
> ## Novelty is incremental
>
> We acknowledge that both prototypes and hypergradient-based adaptation have prior foundations.
>
> However, we would like to clarify that the novelty of this work lies in *how these elements are reformulated and combined in the online, task-free setting*, rather than in proposing entirely new primitives.
>
> Specifically:
> - Existing hypergradient methods operate at a global level (learning rate adaptation), whereas we extend them to *parameter-wise gradient reweighting* in a non-stationary stream.
> - Prior gradient reweighting approaches in continual learning are typically heuristic and limited to the last layer. In contrast, FGH provides a *learned, principled mechanism applied across all parameters*.
> - Most importantly, we show that this formulation addresses two intertwined challenges in onCL: gradient imbalance and learning-rate sensitivity, which are not jointly addressed in prior work.
>
> Therefore, while the individual components are related to prior ideas, their formulation and role in this setting are, to our knowledge, novel.
>
> ## Difference between SGD and generic adaptive rescaling
> We would like to emphasize that FGH differs from standard adaptive methods (e.g., Adam, RPROP) in a key aspect:
>
> - FGH explicitly leverages *temporal gradient alignment* via the inner product $\nabla L_t \cdot \nabla L_{t-1}$, whereas standard methods rely on gradient magnitude or variance.
>
> This distinction leads to different behavior:
> - FGH selectively amplifies gradients that are temporally consistent (typically later tasks),
> - while suppressing unstable gradients (typically earlier tasks).
>
> In contrast, magnitude-based methods cannot distinguish between consistent and oscillatory gradients.
>
> The gradient imbalance analysis (Section 5.1) supports this interpretation, showing that FGH systematically redistributes gradient budget toward later tasks, which aligns with the desired stability-plasticity trade-off.
>
> ## Request 1
>
> We have experimented with RPROP [1] and showed a detailed comparison with our method in this rebuttal. We point the reviewer toward our general reply for more details.
>
> ## Request 2
>
> Spatial and temporal complexity tables are included in the appendix and will be moved to the main paper in the final version of the paper. Importantly, we can note that the introduction of FGH has negligible overhead.
>
> ## Request 3
>
> We would like to respectfully clarify a potential misunderstanding. The method **does not rely on VTAB-based transfer**. The objective is to design a method that is robust to sub-optimal hyperparameters, notably the Learning Rate (LR).  In that sense we conduct evaluation in three scenarios:
>
> - Fixed low LR (0.00005)
> - Fixed "high" LR (0.005)
> - LR found on VTAB
>
> The same procedure is applied to all methods in order to have a comparison that is as fair as possible. The main motivation follows a common trend in continual learning, which is hyper-parameter search, even though this breaks fundamental setup assumptions.
>
> ## Request 4
>
> The explicit details of how FGH is derived with Adam are given in the appendix of the paper, which we are willing to include in the main section for clarity.
>
> ## References
>
> [1] Riedmiller, Martin, and Heinrich Braun. "A direct adaptive method for faster backpropagation learning: The RPROP algorithm." in IEEE International Conference on Neural Networks (ICNN), 1993.

---

### Review · Reviewer_6FFN · 2026-03-16

**Summary Of Contributions:**

Summary:
This manuscript aims at adapting offline continual learning methods to online setting where data arrive sequentially and are seen only once, targeting at class-incremental learning problem specifically. The proposed method has two parts: class prototypes used as lightweight memory proxies for the classifier head, and Fine-Grained Hypergradients (FGH) that learn parameter-wise gradient reweighting to reduce gradient imbalance and sensitivity to learning-rate choice. Across three image classification datasets (CIFAR100, CUB, and ImageNet-R) in both clear-boundary and Si-Blurry settings, adding prototypes plus FGH consistently improves offline CL baselines.

Strengths
- Adapting offline CL methods to a more realistic online setting is very well motivated and practically relevant.
- The method is conceptually straightforward and easy to understand and implement.
- The experiments cover multiple datasets, boundary settings, and continual learning approaches, providing decent comparisons for analysis.

Weaknesses
- The paper claims the setting is memory-free, but the class prototypes is still a form of memory.
- The FGH requires an extra hyperparameter γ, weakening the argument that the method regulates LR sensitivity in realistic online settings.
- The evaluation is limited to ViT-based backbones/prompting methods.

**Audience:**

Yes

**Audience Explanation:**

The continual learning topic is highly relevant to the research interests of TMLR.

**Claims And Evidence:**

Yes

**Claims Explanation:**

In general, the paper provides convincing empirical support through experiments on multiple datasets, ablation studies, gradient analysis, etc.

**Requested Changes:**

The paper would benefit from changes such as a clearer claim of the problem setting, acknowledgement of limitation, and more experiments on different backbones. See aforementioned weaknesses for details.

---

> ### Author Response · Authors · 2026-03-31
>
> We thank the reviewer for their thoughtful review. We respond to specific points below and in the shared response.
>
> ## Memory-Free claim
>
> We agree that, at a high level, prototypes may appear as a form of memory. However, the distinction lies in both *what is stored* and *how it is used*.
>
> Specifically:
> - Prototypes store **class-level sufficient statistics in feature space**, rather than individual samples.
> - The storage complexity is O(C × d), identical to the final classifier layer, and independent of the number of observed samples.
> - No raw data or replayed features are stored, meaning:
>   (i) no increase in privacy risk,
>   (ii) no additional forward passes through the backbone,
>   (iii) no repeated gradient updates on past data.
>
> In contrast, memory-based methods (e.g., ER) scale with the number of stored samples and require repeated replay through the full network.
>
> Therefore, while prototypes can be interpreted as a *minimal statistical memory*, they do not function as rehearsal buffers, and fall under the standard definition of "memory-free" in recent CL literature, similar to [MF1-MF7]. To avoid confusion, we will clarify this distinction more explicitly in the revised manuscript.
>
> ## Additional hyper-parameter
>
> We agree that introducing an additional hyperparameter γ may appear to contradict the goal of reducing hyperparameter sensitivity.
>
> However, our key claim is not to eliminate hyperparameters entirely, but to reduce *sensitivity to critical hyperparameters such as the learning rate*.
>
> Importantly, γ exhibits low sensitivity in practice:
> - As shown in Figure 4 (Section 5.2), performance remains stable across several orders of magnitude of γ.
> - In contrast, performance varies significantly with respect to the learning rate. (As it can be observed in Tables 1 and 2 of the main paper).
>
> This indicates that γ is not a sensitive tuning parameter, but rather a *robust scaling factor*, for which a single default value (γ = 1) performs well across datasets and methods.
>
> ## Limited backbone evaluation
>
> We reported results with additional backbones in our rebuttal. Various ViT sizes are also tested in the appendix.
>
> ## Including limitations
>
> Limitations are discussed in Section 5 of the paper around discussions regarding $\gamma$ selection and prototype synergy. Spatial and temporal overhead are also included in the appendix of the paper.
>
> ## References
>
> [MF1] Z. Zheng, M. Ma, K. Wang, Z. Qin, X. Yue, and Y. You. Preventing zero-shot transfer degradation in continual learning of vision-language models. In ICCV, 2023.
>
> [MF2] D.-W. Zhou, H.-L. Sun, H.-J. Ye, and D.-C. Zhan. Expandable subspace ensemble for pre-trained model-based class-incremental learning. CVPR, 2024.
>
> [MF3] M. D. McDonnell, D. Gong, A. Parveneh, E. Abbasnejad, and A. v. d. Hengel. Ranpac: Random projections and pre-trained models for continual learning. arXiv preprint arXiv:2307.02251, 2023
>
> [MF4] K. Ahrens, H. H. Lehmann, J. H. Lee, and S. Wermter. Read between the layers: Leveraging intra-layer representations for rehearsal-free continual learning with pre-trained models. arXiv preprint arXiv:2312.08888, 2023
>
> [MF5] G. Zhang, L. Wang, G. Kang, L. Chen, and Y. Wei. Slca: Slow learner with classifier alignment for continual learning on a pre-trained model. In ICCV, 2023
>
> [MF6] J. Zheng, S. Qiu, and Q. Ma. Learn or recall? Revisiting incremental learning with pre-trained language models. arXiv preprint arXiv:2312.07887, 2023.
>
> [MF7] H. Guo, F. Zhu, W. Liu, X.-Y. Zhang, and C.-L. Liu. Federated class-incremental learning with prototype-guided transformer. arXiv preprint arXiv:2401.02094, 2024.

---

### Review · Reviewer_rDm9 · 2026-03-17

**Summary Of Contributions:**

This paper addresses the problem of adapting memory-free offline Continual Learning (offCL) methods to the online, task-free, and memory-free setting (onCL). The authors identify two key challenges arising from this adaptation: (1) the use of lightweight prototypes as a proxy for memory buffers increases Gradient Imbalance (GI), biasing learning toward earlier tasks, and (2) learning rate selection is particularly difficult in the online setting where  task boundary information are unavailable.

To address these issues, the paper proposes Fine-Grained Hypergradients (FGH), which extend classical hypergradient theory from global learning rate adaptation to parameter-wise gradient reweighting. FGH learns per-parameter weighting coefficients that rebalance gradient magnitudes across tasks, mitigating GI while also improving robustness to suboptimal learning rate choices. The authors additionally introduce a multi-learning-rate evaluation protocol that assesses methods across a range of initial learning rates, offering a fairer comparison than fixed-LR benchmarks.

**Key Strengths:**

- The paper tackles a practically important and underexplored problem: bridging offCL and onCL in a memory-free, task-free setting.
- The proposed FGH mechanism is principled, lightweight, and adds minimal computational overhead.
- The synergy between prototypes and FGH is well-motivated and empirically validated through thorough ablation studies.
- The multi-LR evaluation protocol is a sensible contribution that better reflects realistic deployment conditions.
- Comprehensive experiments across multiple datasets (CIFAR100, CUB, ImageNet-R), boundary settings (clear and Si-Blurry), and baselines (L2P, DualPrompt, CODA, ConvPrompt).
- The paper is clearly written and well-structured.

**Key Weaknesses:**

- The comparison is limited to prompt-based and LoRA-based methods. A discussion or comparison with sparse parameter update strategies adapted to this setting would strengthen the positioning (see Requested Changes).

**Audience:**

Yes

**Audience Explanation:**

The paper addresses a relevant gap in continual learning research. Online, task-free, and memory-free CL is among the most realistic and challenging settings, yet it remains underexplored relative to offline or memory-based alternatives.

**Broader Impact Concerns:**

No concerns. The work is methodological in nature and does not raise ethical issues beyond those common to machine learning research.

**Claims And Evidence:**

Yes

**Claims Explanation:**

The claims are well-supported by the experimental evidence. The ablation study clearly disentangles the contributions of prototypes and FGH. The multi-LR evaluation across three datasets demonstrates consistent improvements.

**Requested Changes:**

1. **(Minor, would strengthen the work)** The paper focuses on prompt-based and LoRA-based offCL methods. It would be valuable to discuss or compare with sparse/selective parameter update methods that have been proposed for continual learning with pre-trained models, such as:

    - Zhang et al., "Overcoming Generic Knowledge Loss with Selective Parameter Update," CVPR 2024.
    - Zheng et al., "Preventing Zero-Shot Transfer Degradation in Continual Learning of Vision-Language Models," ICCV 2023.

    Even a discussion of how FGH relates to or could complement selective update strategies would provide useful context and strengthen the related work section.

---

> ### Author Response · Authors · 2026-03-31
>
> We appreciate the reviewer’s positive assessment of our work, particularly the recognition of FGH's efficiency and the realism of our task-free, memory-free setting. We respond to specific points below.
>
> ## Discussion regarding other selective parameter update methods
>
> We deeply appreciate the additional references suggested by the reviewer and will include specific discussion regarding [A] and [B] (detailed below) in the final version of the paper. However, their applicability in our online, task-free scenario is fundamentally limited. While these methods are effective in offline or task-incremental scenarios, they face fundamental limitations in the Online, Task-Free, and Memory-Free setting addressed in our work.
>
> Concretely, these methods require stable estimates of parameter importance, which are typically obtained via: (i) multiple passes over task data, and (ii) explicit task boundary signals to trigger consolidation. Both of which are unavailable in our setting.
>
> Namely, methods like [A] and [B] typically require explicit knowledge of task boundaries to calculate importance scores or perform joint parameter selection. In our task-free setting, these methods cannot easily trigger the necessary consolidation or masking phases. As we explained in the appendix regarding the adaptation of the compared methods, this is not trivial. Futhermore selective updates rely on multi-pass optimization or warm-up phases to identify task-relevant parameters. In an online stream where each sample is seen only once, these importance estimates are highly noisy. Attempting to freeze or selectively update parameters based on a single look often leads to underfitting, as the model lacks the "global" perspective of the task to make stable selection decisions.
>
> More importantly, these approaches focus on *which parameters to update*, whereas our work addresses a complementary problem: *how gradients should be scaled over time*. Even with perfect parameter selection, gradient imbalance remains, as recent samples dominate update magnitudes.
>
> ## References
>
> [A] Zhang et al., "Overcoming Generic Knowledge Loss with Selective Parameter Update" in CVPR, 2024.
>
> [B] Zheng et al., "Preventing Zero-Shot Transfer Degradation in Continual Learning of Vision-Language Models" in ICCV, 2023.

---

### Author Response · Authors · 2026-03-31

We would like to thank the reviewers for their insightful comments and suggestions. We are encouraged that the reviewers found our work to be well-motivated and practically relevant (rDm9, 6FFN, Cqv4), our Fine-Grained Hypergradients (FGH) mechanism to be principled and lightweight (rDm9), and our multi-learning-rate evaluation protocol to be a valuable and sensible contribution to the field (rDm9, Cqv4). We replied individually to each reviewer regarding specific points, and make a general reply for shared concerns and requests.

## Additional Experiments (rDm9, 6FFN, Cqv4)

We thank the reviewers for asking for (i) a stronger backbone transfer experiment and (ii) simpler adaptive reweighting baselines that do not rely on our hypergradient derivation. We ran both additions and summarize them below.

For the main rebuttal tables, we report `batch_size=100`, which is the setting used in the paper. We focus below on the fixed high learning rate (`0.005`). Specific numbers might still differ due to different seeds and high variance. Similarly, `prototype only` represents using prototypes without any reweighting strategy.

## 1. Additional CNN Comparison Against RPROP [1]

| Dataset | Backbone | prototype only | ours | rprop_param |
|---|---|---|---|---|
| cifar224 | pretrained_resnet50 | 38.43±0.39 | **40.04±0.16** | 32.30±1.15 |
| cub | pretrained_resnet50 | 24.47±3.10 | **25.11±3.23** | 20.94±1.59 |
| imagenetr | pretrained_resnet50 | 36.96±1.12 | **37.63±1.48** | 35.83±1.27 |

### CODA-Prompt

| Dataset | Blurry | prototype only | ours | rprop_param | rprop_layer |
|---|---|---|---|---|---|
| cifar224 | False | 82.59±1.80 | 82.61±1.11 | 80.37±0.97 | **84.67±0.78** |
| cifar224 | True | 83.34±2.31 | **83.59±1.86** | 79.30±2.08 | 82.54±2.15 |
| cub | False | 59.20±1.82 | 61.93±1.15 | 60.00±2.96 | **63.13±2.59** |
| cub | True | 57.99±1.49 | 59.22±1.24 | 59.58±3.37 | **65.01±1.94** |
| imagenetr | False | 73.10±0.59 | 73.17±0.25 | **73.54±0.27** | 72.61±1.00 |
| imagenetr | True | 63.62±1.53 | 63.34±1.34 | **65.08±1.64** | 65.04±2.08 |

### DualPrompt

| Dataset | Blurry | prototype only | ours | rprop_param | rprop_layer |
|---|---|---|---|---|---|
| cifar224 | False | **85.86±0.78** | 85.72±0.79 | 84.13±0.64 | 84.40±0.84 |
| cifar224 | True | 83.35±2.52 | **83.53±2.39** | 80.86±2.72 | 82.41±2.91 |
| cub | False | 65.34±0.26 | 65.60±0.24 | 64.94±0.51 | **70.38±0.30** |
| cub | True | 61.75±1.20 | 62.12±1.20 | 62.21±0.73 | **67.06±0.35** |
| imagenetr | False | **70.27±0.28** | **70.27±0.38** | 68.36±0.49 | 68.32±0.65 |
| imagenetr | True | 61.67±2.06 | **61.86±2.13** | 60.21±2.02 | 60.61±1.81 |

### L2P

| Dataset | Blurry | prototype only | ours | rprop_param | rprop_layer |
|---|---|---|---|---|---|
| cifar224 | False | 57.68±3.86 | **58.66±3.71** | 48.77±3.01 | 58.46±4.43 |
| cifar224 | True | 50.84±2.87 | **51.74±2.62** | 41.70±3.66 | 51.05±2.82 |
| cub | False | 24.46±2.15 | **25.15±2.14** | 22.68±1.52 | 23.22±1.46 |
| cub | True | 22.98±2.61 | 23.81±2.44 | 21.54±2.33 | **23.89±2.35** |
| imagenetr | False | 37.51±1.14 | **38.00±1.31** | 29.26±1.80 | 37.51±2.44 |
| imagenetr | True | 25.32±2.22 | **26.16±1.99** | 20.89±0.95 | 25.99±1.37 |

### ConvPrompt

| Dataset | Blurry | prototype only | ours | rprop_param | rprop_layer |
|---|---|---|---|---|---|
| cifar224 | False | 84.55 (s=1) | **87.78 (s=1)** | 87.18 (s=1) | 86.19 (s=1) |
| cifar224 | True | **86.53 (s=1)** | 86.19 (s=1) | 85.59 (s=1) | 85.69 (s=1) |
| cub | False | 68.56 (s=1) | 66.46 (s=1) | 70.55 (s=1) | **70.70 (s=1)** |
| cub | True | 69.74 (s=1) | **70.50 (s=1)** | 67.29 (s=1) | 66.30 (s=1) |
| imagenetr | False | 67.12 (s=1) | **68.36 (s=1)** | 66.42 (s=1) | 67.00 (s=1) |
| imagenetr | True | **59.57 (s=1)** | 59.01 (s=1) | 34.59 (s=1) | 57.54 (s=1) |

- `ConvPrompt` is reported with a single seed (`seed=1`) because of runtime.

### Takeaway:
- `ours` is better than `rprop_param` on `22/27` available CNN/prompt comparisons.
- Against `rprop_layer`, `ours` is stronger on `16/24` prompt comparisons. `rprop_layer` is mostly strong on CUB.
- `rprop_param` is also much less consistent in this regime: it is worse than `prototype only` in `20/27` rows, with large failures such as `59.57 -> 34.59` on `imagenetr`, blurry, `ConvPrompt`, and `38.43 -> 32.30` on `cifar224` with `pretrained_resnet50`. In contrast, `ours` is worse than `prototype only` in only `5/27` rows, and the largest observed drop is `68.56 -> 66.46` on non-blurry `cub` with `ConvPrompt`.
- Note that `prototype only` is already a baseline, not the original prompt/CNN method without recalibration.

---

> ### Author Response · Authors · 2026-03-31
>
> ### Gradient Imbalance (Cqv4)
> To compare the imbalance reduction factor of reweighting strategies, we look at how evenly the gradient budget is distributed across classes during training using three concentration metrics: Coefficient of Variation (CV), the Gini coefficient, and the top-10 share.
>
> Our method demonstrates a more balanced class-wise gradient allocation compared to the rprop_param baseline. According to a classifier-gradient probe conducted on cifar (with batch_size=100, seed=1, and init_lr=0.005), our approach yields lower values across all three metrics:
>
> | Metric | rprop_param | ours |
> | :--- | :--- | :--- |
> | **CV** | 2.8834 | **2.7595** |
> | **Gini** | 0.9011 | **0.8900** |
> | **Top-10 Share** | 0.8553 | **0.8100** |
>
> **Metric Definitions**
>
> These metrics are calculated by accumulating the post-reweight gradient norm for each row of the final classifier throughout the training process:
>     - CV (Coefficient of Variation): Measures the standard deviation divided by the mean of the cumulative per-class budgets. A lower CV indicates less dispersion and more uniformity.
>     - Gini Coefficient: A standard measure of inequality; a lower Gini coefficient signifies that gradient updates are less concentrated on a small subset of classes.
>     - Top-10 Share: The fraction of the total gradient budget captured by the 10 most-updated classes.
>
> ## References
> [1] Riedmiller, Martin, and Heinrich Braun. "A direct adaptive method for faster backpropagation learning: The RPROP algorithm." in IEEE International Conference on Neural Networks (ICNN), 1993.

---

### Decision · Action_Editor_mwKw · 2026-05-03

**Recommendation:** Accept as is

**Audience:**

Yes

**Audience Explanation:**

The paper focuses on continual learning and it is a relevant problem in TMLR community.

**Claims And Evidence:**

Yes

**Claims Explanation:**

This paper addresses the important problem of adapting memory-free offline continual learning methods to the online, task-free setting, and proposes a simple yet effective combination of prototypes and Fine-Grained Hypergradients (FGH).

All reviewers are positive and lean toward acceptance. They agree that the problem is relevant, the method is well-motivated and easy to integrate, and the empirical evaluation is comprehensive and convincing.

Concerns mainly relate to incremental novelty, clearer isolation of FGH’s contribution, and some clarifications on setup and comparisons. The authors addressed these points well in the rebuttal.